# Measurement-Consistent Langevin Corrector for Stabilizing Latent Diffusion Inverse Problem Solvers

Lee Hyoseok [1]  Sohwi Lim [1]  Eunju Cha [†2]  Tae-Hyun Oh [†1]

## Abstract

While latent diffusion models (LDMs) have emerged as powerful priors for inverse problems, existing LDM-based solvers frequently suffer from instability. In this work, we first identify the instability as a discrepancy between the solver dynamics and stable reverse diffusion dynamics learned by the diffusion model, and show that reducing this gap stabilizes the solver. Building on this, we introduce *Measurement-Consistent Langevin Corrector (MCLC)*, a theoretically grounded plug-and-play stabilization module that remedies the LDM-based inverse problem solvers through measurement-consistent Langevin updates. Compared to prior approaches that rely on linear manifold assumptions, which often fail to hold in latent space, MCLC provides a principled stabilization mechanism, leading to more stable and reliable behavior in latent space.

## 1. Introduction

In many scientific and engineering problems, we have access only to limited observations obtained through a forward system; thus, recovering the underlying signal from these measurements is a longstanding challenge, known as the *inverse problem* (Groetsch, 1993). As formalized by Hadamard (1902), inverse problems from partial and noisy measurements are ill-posed, necessitating prior knowledge of the signal domain. While hand-crafted priors (Romano et al., 2017; Ulyanov et al., 2018) impose manually designed regularization, their limited expressiveness has motivated studies on data-driven priors. With advances in generative modeling, diffusion models serve as strong learned priors and show remarkable performance in inverse problems (Chung et al., 2023; Song et al., 2021; Kawar et al., 2022).

†Equal correspondence. [1]KAIST [2]Sookmyung Women's University. Correspondence to: Eunju Cha <eunju.cha@sookmyung.ac.kr>, Tae-Hyun Oh <taehyun.oh@kaist.ac.kr>.

*Proceedings of the 43rd International Conference on Machine Learning*, Seoul, South Korea. PMLR 306, 2026. Copyright 2026 by the author(s).

Due to the ability of latent diffusion models (LDMs) to efficiently learn priors from large-scale datasets and their strong generalization capability, LDM-based inverse problem solvers have been studied. Nevertheless, existing LDM-based solvers fall short due to the solver instability, which leads to artifacts and degraded reconstruction quality (Rout et al., 2023; 2024; Song et al., 2024; Zhang et al., 2025), as shown in the first row of Fig. 1. To address this instability, prior works (Chung et al., 2022; 2023; Zirvi et al., 2025; He et al., 2024) have interpreted the instability as off-manifold behavior through the lens of the manifold hypothesis. In order to formalize and mitigate off-manifold behavior, they have relied on the strong linear manifold assumption and studied manifold-preserving approaches under this assumption. Despite these efforts, such approaches still suffer from instability, as the assumption fails to hold in latent space (Song et al., 2024), particularly due to highly nonlinear decoder (Chung et al., 2024; Raphaeli et al., 2025).

In this work, we take a different perspective on the instability of LDM-based inverse problem solvers. Rather than interpreting instability through geometric manifold assumptions, we characterize it as a discrepancy between the solver-induced dynamics and the stable reverse diffusion dynamics defined by the learned time-marginal distributions. This perspective provides a concrete and explicit notion of instability, grounded in the stable reverse dynamics specified by the diffusion training objective. By introducing this explicit notion, we can directly stabilize LDM-based inverse solvers by reducing the discrepancy which is a principled stabilization mechanism for inverse solvers operating in latent space, without relying on manifold assumptions that often fail to hold in latent space.

To this end, we propose **M**easurement-**C**onsistent **L**angevin **C**orrector (MCLC), a theoretically justified plug-and-play method that stabilizes existing solvers by reducing the characterized gap without compromising measurement fidelity, the core objective of inverse problems. MCLC achieves this by constraining Langevin updates to the orthogonal complement of the measurement-consistent gradient. As a result, MCLC leads to more stable and reliable solutions, as demonstrated by overall performance improvement for base and competing methods across a range of inverse problems.

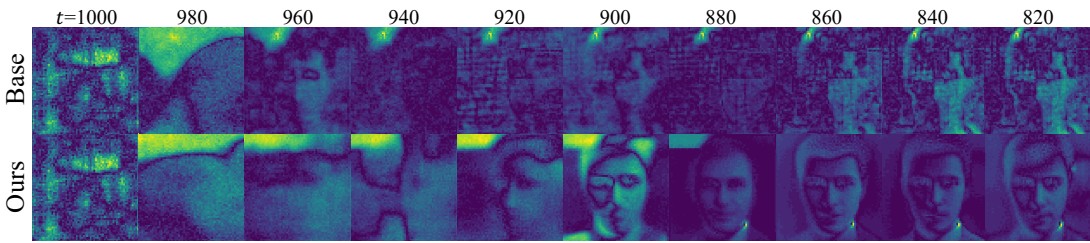

*Figure 1.* **Reverse dynamics of LDM-based inverse problem solver, PSLD.** We visualize the reverse sampling trajectory of $z_{0|t}$ for the latent diffusion inverse solver, PSLD (Rout et al., 2023). The naive dynamics of the solver exhibit undesirable artifacts (first row), whereas the solver corrected with MCLC yields cleaner and more structured latents (second row). For clarity, only the fourth channel is visualized.

Our contributions are summarized as follows:

- We introduce a new perspective on the instability of LDM-based inverse problem solvers by characterizing it as a discrepancy from the target reverse diffusion dynamics

- We propose Measurement-Consistent Langevin Corrector (MCLC), a theoretically grounded and plug-and-play stabilization module that directly reduces the discrepancy while preserving measurement fidelity.

## 2. Background and Related Work

### 2.1. Diffusion Models

Diffusion models learn data distribution $p(\boldsymbol{x})$ by modeling score vector field, *i.e.*, $\nabla_{\boldsymbol{x}} \log p(\boldsymbol{x})$ (Song et al., 2021). Since the true score of the data distribution is intractable, a forward diffusion process is introduced that gradually perturbs the data into a Gaussian distribution (Ho et al., 2020). This process is formulated as stochastic differential equations (SDE), defining a family of marginal distributions at each timestep $\{q_t\}_{t \in [0,1]}$. The training objective, denoising score matching, is defined such that the reverse process matches the family of distributions induced by the forward process. Formally, the model is trained to minimize the expected KL divergence across timesteps: $\mathbb{E}_{t \sim \mathcal{U}[0,1]}[\mathcal{D}_{\mathrm{KL}}(q_t \| p_t)]$. With the learned score network, sampling is performed by solving the reverse-time SDE (or probability flow ODE) (Ho et al., 2020).

### 2.2. Diffusion Inverse Solvers

Inverse problems aim to estimate the underlying signal $\boldsymbol{x}$ from measurements $\boldsymbol{y}$, formulated as: $\boldsymbol{y} = \mathcal{A}(\boldsymbol{x}) + \boldsymbol{n}$, where $\mathcal{A}$ is a measurement operator and $\boldsymbol{n}$ denotes measurement noise. From a Bayesian perspective, the goal is to characterize the posterior distribution $p(\boldsymbol{x}|\boldsymbol{y}) \propto p(\boldsymbol{y}|\boldsymbol{x}) \, p(\boldsymbol{x})$ where the $p(\boldsymbol{y}|\boldsymbol{x})$ represents measurement consistency and $p(\boldsymbol{x})$ is prior knowledge of the signal. Posterior inference can be performed via the gradient of the log-posterior, which decomposes into likelihood and prior terms:

$$\nabla_{\boldsymbol{x}} \log p(\boldsymbol{x} \mid \boldsymbol{y}) = \nabla_{\boldsymbol{x}} \log p(\boldsymbol{y} \mid \boldsymbol{x}) + \nabla_{\boldsymbol{x}} \log p(\boldsymbol{x}). \quad (1)$$

In diffusion-based inverse solvers, the prior gradient is provided by a pre-trained diffusion model, and the likelihood gradient is derived from the measurement model. By incorporating the likelihood gradient into the diffusion sampling process, diffusion inverse problem solvers (Kawar et al., 2022; Wang et al., 2023) achieve measurement-consistent posterior sampling and have been widely adopted.

**Instability of Diffusion Inverse Solvers.** While diffusion-based inverse problem solvers have shown remarkable performance, they often exhibit unstable behaviors (Chung et al., 2022). The measurement-consistency steps can push the sampling path off the data manifold, and such off-manifold drift induces instability, leading to reduced fidelity. To mitigate this issue, prior studies (Chung et al., 2022; 2023; He et al., 2024; Zirvi et al., 2025) adopt a linear manifold assumption, under which the diffusion manifold is locally approximated as linear, and propose manifold-preserving solvers that aim to preserve on the diffusion manifold. He et al. (2024) use a pretrained autoencoder for manifold projection, and Zirvi et al. (2025) project the gradient, both seeking to prevent off-manifold updates.

### 2.3. Latent Diffusion Inverse Solvers

Since scaling diffusion models to large datasets in the raw signal domain (*e.g.*, pixel) is computationally prohibitive, Latent Diffusion Models (LDMs) (Rombach et al., 2022) perform in the latent space of a pretrained autoencoder (Kingma & Welling, 2014). In latent space, the signal is represented as $\boldsymbol{x} = D(\boldsymbol{z})$, where $\boldsymbol{z}$ denotes the latent variable and $D$ is a decoder. In this context, a line of works has extended diffusion-based inverse solvers to LDMs (Rout et al., 2023; 2024; Song et al., 2024; Zhang et al., 2025; Kim et al., 2025b). While these approaches broaden the applicability and generalizability of inverse solvers, instability remains an inherent challenge, leading to artifacts and degraded quality (Chung et al., 2024; Raphaeli et al., 2025).

A few notable approaches (Zirvi et al., 2025; He et al., 2024) have addressed this challenge by extending manifold-preserving methods to LDMs under the linear manifold assumption. While these methods alleviate instability to some extent, LDM-based inverse problem solvers still suffer

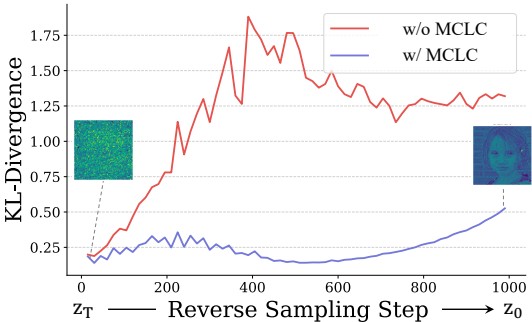

*Figure 2.* **Instability characterized via KL divergence.** We plot KL divergence between the time-evolving distribution of an LDM-based inverse problem solver and the time-marginal distribution of the pretrained diffusion model. The clear gap (red line) supports our assumption, and MCLC effectively narrows it (purple line). Detailed experimental settings are provided in the Appendix. D.1.

from instability (Zirvi et al., 2025; Raphaeli et al., 2025). We attribute this limitation to the failure of the linear manifold assumption in latent space. Specifically, even if the data manifold in raw signal (*e.g.*, pixel) space holds the linear manifold assumption, this property does not carry over to the latent space due to the highly nonlinear decoder (see Sec. D.2 of Song et al. (2024)).

## 3. Measurement-Consistent Langevin Corrector (MCLC)

Firstly, to clarify the instability observed in prior works, we examine the reverse diffusion dynamics of latent diffusion inverse solvers, where the sequence of time-marginal distributions is defined as the reverse diffusion dynamics. While prior works often describe instability as off-manifold behavior under the linear manifold assumption, we instead characterize it concretely as deviations from the learned time-marginal distributions of the diffusion model. Since diffusion models are trained to match these time-marginals at each timestep, they provide a concrete and measurable reference for stable reverse dynamics. To quantify this discrepancy, we measure the Kullback–Leibler (KL) divergence between the solver-induced dynamics and the learned time-marginal distributions (*i.e.*, stable reverse dynamics). Figure 2 demonstrates that the naive solver dynamics exhibit a significant gap, indicating clear divergence across timesteps. To formalize this observed discrepancy, we introduce the following assumption.

**Assumption 3.1** (Deviation from $p_t$). The measurement-guided, time-evolving distribution $q_t^{\#}$ deviates from $p_t$ at timestep $t$. Formally,

$$\mathcal{D}_{\mathrm{KL}}(q_t^{\#}\|p_t) \;\geq\; \gamma_t, \quad \text{for some } \gamma_t > 0, \qquad (2)$$

where $p_t$ denotes the time-marginal distribution induced by the reverse dynamics of the pretrained diffusion model.

As mentioned in Sec. 2.2, the measurement consistency step is essential for solving inverse problems, but it may induce unstable solver dynamics (Assumption 3.1). To explicitly reduce this undesirable discrepancy, we introduce a post-update Langevin step, referred to as a corrector. Since Langevin dynamics are guaranteed to converge to a target stationary distribution when driven by the gradient of its log-density, the following proposition establishes that applying Langevin dynamics with the score estimated from the diffusion model, after the measurement-consistency step, drives the solver dynamics toward the time-marginal distribution $p_t$ (Vempala & Wibisono, 2019). This convergence stabilizes the dynamics of latent diffusion inverse solvers, which improves reconstruction fidelity. The proof of Proposition 3.2 can be found in Appendix. A.

**Proposition 3.2** (Langevin Corrector). *Fix a timestep $t$ and let $p_t$ be a target distribution. Consider the continuous corrector process $\{Z_t^c\}_{c\geq 0}$ initialized with $Z_t^0 \sim q_t^{\#}$. The process evolves according to the Langevin dynamics with frozen target $p_t$: $dZ_t^c = \nabla \log p_t(Z_t^c)dc + \sqrt{2}dW_c$. Let $q_t^c$ denote the distribution of $Z_c$. Then, the KL divergence monotonically decreases along the process, unless $q_t^c = p_t$, in which case equality holds:*

$$\mathcal{D}_{\mathrm{KL}}(q_t^c\|p_t) \leq \mathcal{D}_{\mathrm{KL}}(q_t^{\#}\|p_t), \qquad \forall c \geq 0. \qquad (3)$$

Importantly, the role of corrector is to reduce the discrepancy arising from the measurement-consistency step, rather than numerical errors from discretizing the diffusion dynamics. As prior studies (Dalalyan, 2017; Durmus & Moulines, 2019) have shown, discretization of the Langevin process preserves the property of decreasing KL divergence up to discretization error, provided the step size is sufficiently small. In this work, we implement the corrector using the Euler–Maruyama discretization of Langevin SDE:

$$z_t^c \leftarrow z_t^{\#} + \eta_t \, \nabla \log p_t(z_t^{\#}) + \sqrt{2\eta_t} \, \epsilon, \qquad (4)$$

where $\epsilon \sim \mathcal{N}(0, I)$, and $z_t^{\#}$ denotes the latent variable after applying the measurement-consistency step.

*Remark* 3.3 (Vanilla corrector may disturb measurement consistency). While the Langevin corrector is effective in reducing the discrepancy, the vanilla Langevin update may disturb measurement consistency $r(z_t) := L(z_t, y)$ imposed by the LDM inverse solver. A first-order Taylor expansion of $r$ after the Langevin update is given by:

$$r(z_t + \Delta z_t) \approx r(z_t) + \nabla_{z_t} r(z_t) \, \Delta z_t, \qquad (5)$$

where $\Delta z_t = \eta_t \nabla \log p_t(z_t^{\#}) + \sqrt{2\eta_t} \epsilon$ represents the Langevin corrector step. Even when higher-order terms are neglected, the measurement consistency is perturbed since the first-order term $\nabla_{z_t} r(z_t) \Delta z_t \neq 0$ in general, that is, $\mathbb{E}[r(z_t + \Delta z_t)] \neq r(z_t)$.

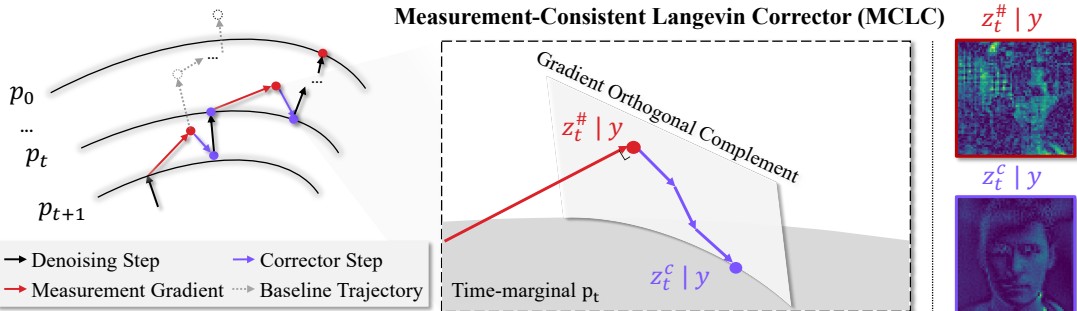

*Figure 3.* **Illustration of MCLC.** After the measurement consistency step, MCLC is applied to mitigate the off-stationarity. MCLC performs Langevin updates on the subspace orthogonal to the measurement gradient, thereby preserving measurement consistency during the correction process. Our proposed corrector effectively alleviates the problematic latent updates.

Although the instability of LDM-based inverse solvers remains a persistent challenge to be addressed, ensuring measurement fidelity is essential for faithful signal reconstruction in inverse problems. However, as noted in Remark 1, even neglecting higher-order terms, the vanilla Langevin update generally perturbs measurement consistency. This motivates us to propose *Measurement-Consistent Langevin Corrector* (MCLC), which applies an orthogonal projection at each Langevin update step onto the current measurement-consistent subspace. The MCLC update takes the form:

$$z_t^c \leftarrow z_t^\# + \eta_t \cdot P_{\perp g_t} s_\theta(z_t^\#, t) + \sqrt{2\eta_t} \cdot P_{\perp g_t}(\epsilon), \quad (6)$$

where $g_t := \dfrac{\nabla_{z_t} r(z_t)}{\|\nabla_{z_t} r(z_t)\|}$ and $P_{\perp g} = (I - gg^T)$ denotes projection of $v$ onto the orthogonal complement of $g$.

MCLC restricts the correction step to the orthogonal complement of the measurement-consistent gradient. Consequently, it preserves measurement consistency up to the first-order Taylor expansion (*i.e.*, $\nabla_{z_t} r(z_t)\Delta z_t = 0$). Even if higher-order terms are taken into account, MCLC still guarantees that the perturbation $\Delta z_t$ after the Langevin update can be bounded in terms of the step size. The MCLC is illustrated in Fig. 3 and the algorithm is detailed in Algorithm 2.

**Theorem 3.4.** *The projected Langevin update onto the orthogonal complement of the measurement gradient decreases the KL divergence while preserving measurement consistency up to a controlled bound. If the update satisfies*

$$\mathbb{E}[\|\Delta z_t\|^2] \le k < 1, \quad (7)$$

*the expected measurement consistency perturbation follows:*

$$\mathbb{E}[\Delta r] \le Ck + O(k), \quad (8)$$

*for some $C > 0$ depending on local smoothness of $r$.*

In particular, as $k$ is controlled by the step size $\eta_t$, $\mathbb{E}[\Delta r]$ can be bounded at each timestep by selecting $\eta_t$ appropriately, thereby preserving measurement consistency while reducing the KL divergence. Theorem 3.4 suggests that latents

deviated from $p_t$ can be pushed back toward the stationary distribution in terms of the KL divergence, while preserving measurement consistency within a controlled error. Detailed proofs are given in Appendix. A. Based on the theoretical derivations, we obtain a dimension-dependent upper bound on the step size, which enables stable correction and provides a theoretically grounded setting. In practice, the step size and the number of correction steps are tuned to balance data fidelity, stability, and computational overhead. Such tuning is inherent to inverse problems due to varying degradation levels and measurement characteristics across tasks and instances, and is not specific to our method (Daubechies et al., 2004; Beck & Teboulle, 2009; Hu & Jacob, 2012; Chen et al., 2020; Hurault et al., 2023; Fabian et al., 2024; Pandey et al., 2024; Shen et al., 2024). Despite this, we provide guidelines for hyperparameters and show that a single hyperparameter setting generalizes well across tasks, measurement operators, and solvers. (See Appendix. B.7)

In summary, prior approaches interpret instability through the lens of the manifold hypothesis and rely on strong linear manifold assumptions to formalize and mitigate off-manifold behavior. However, such assumptions do not generally hold in latent spaces (Song et al., 2024). Instead, we characterize instability more concretely as a discrepancy between the solver-induced dynamics and the stable reverse diffusion dynamics defined by the sequence of time-marginal distributions. From this perspective, we propose MCLC, a principled correction scheme that reduces this discrepancy while preserving measurement fidelity, and can be integrated into existing LDM-based solvers in a plug-and-play manner. Here, plug-and-play refers to the modular integration of the correction mechanism, rather than the complete absence of hyperparameter tuning.

## 4. Experiments

**Experimental setup.** We mainly evaluate our method by plugging it into existing LDM-based inverse solvers, including Latent DPS (LDPS) (Chung et al., 2023), PSLD (Rout

*Table 1.* **Quantitative comparison for linear and nonlinear tasks on FFHQ and ImageNet.** MCLC improves overall performance across diverse methods, demonstrating compatibility while achieving impressive performance compared to each baseline and DiffStateGrad.

| Task | Base | Method | FFHQ | | | | ImageNet | | | |
|---|---|---|---|---|---|---|---|---|---|---|
| | | | PSNR ($\uparrow$) | LPIPS ($\downarrow$) | FID ($\downarrow$) | P-FID ($\downarrow$) | PSNR ($\uparrow$) | LPIPS ($\downarrow$) | FID ($\downarrow$) | P-FID ($\downarrow$) |
| Gaussian Deblur | LDPS | Base | 27.61 | 0.349 | 100.10 | 93.55 | 25.04 | 0.407 | 120.79 | 108.52 |
| | | DiffState | 27.59 | 0.348 | 100.82 | 94.14 | 25.00 | 0.409 | 122.14 | 106.84 |
| | | Ours | **28.14** | **0.303** | **80.83** | **54.74** | **25.84** | **0.395** | **103.87** | **93.28** |
| | PSLD | Base | 27.84 | 0.314 | 89.18 | 90.54 | 25.52 | **0.371** | 104.86 | 108.76 |
| | | DiffState | 27.89 | 0.311 | 86.73 | 87.90 | 25.47 | 0.377 | 106.90 | 109.92 |
| | | Ours | **27.97** | **0.286** | **66.28** | **59.13** | **25.89** | 0.380 | **92.74** | **95.01** |
| | ReSample | Base | 26.44 | 0.368 | 75.17 | 148.11 | 24.15 | 0.404 | 83.90 | 135.07 |
| | | DiffState | 26.05 | 0.396 | **74.03** | 140.76 | 24.12 | 0.417 | **79.57** | 133.22 |
| | | Ours | **27.25** | **0.353** | 78.38 | **106.16** | **25.19** | **0.378** | 81.71 | **123.00** |
| | LatentDAPS | Base | 27.51 | **0.348** | **99.53** | **120.56** | 25.41 | 0.375 | **112.54** | 111.22 |
| | | DiffState | **27.52** | 0.349 | 106.04 | 122.03 | **25.47** | **0.374** | 113.57 | **110.81** |
| | | Ours | 27.42 | 0.349 | 100.58 | 123.06 | 25.42 | 0.376 | 116.35 | 111.04 |
| Motion Deblur | LDPS | Base | 26.54 | 0.390 | 118.77 | 112.74 | 23.93 | 0.451 | 154.30 | 121.79 |
| | | DiffState | 26.56 | 0.387 | 118.71 | 110.60 | 24.08 | 0.447 | 154.48 | 122.01 |
| | | Ours | **27.45** | **0.318** | **82.94** | **55.55** | **24.79** | **0.424** | **119.65** | **97.68** |
| | PSLD | Base | 26.87 | 0.343 | 106.34 | 102.60 | 24.54 | 0.407 | 141.67 | 121.41 |
| | | DiffState | **26.88** | 0.340 | 107.95 | 102.28 | 24.60 | 0.401 | 138.91 | 121.98 |
| | | Ours | 26.86 | **0.308** | **74.64** | **60.05** | **24.94** | **0.387** | **99.21** | **93.49** |
| | ReSample | Base | 22.45 | 0.635 | 108.14 | 174.52 | 21.42 | **0.589** | 156.25 | 158.90 |
| | | DiffState | 23.16 | 0.623 | 104.42 | 133.68 | 21.58 | 0.633 | **101.84** | 129.92 |
| | | Ours | **24.24** | **0.588** | **102.02** | **118.87** | **22.33** | 0.616 | 102.81 | **118.81** |
| | LatentDAPS | Base | **24.83** | **0.491** | **160.50** | **198.21** | 23.06 | 0.503 | 183.22 | 142.50 |
| | | DiffState | 24.60 | 0.494 | 163.28 | 199.27 | 23.11 | **0.500** | 184.03 | **141.35** |
| | | Ours | 24.65 | 0.496 | 165.19 | 199.74 | **23.31** | 0.502 | **176.77** | 141.66 |
| Super Resolution (4×) | LDPS | Base | 28.47 | 0.301 | 78.08 | 69.66 | 26.22 | 0.401 | 118.33 | 107.19 |
| | | DiffState | **28.58** | 0.299 | 77.40 | 68.14 | **26.25** | 0.401 | 115.18 | 106.66 |
| | | Ours | 28.34 | **0.283** | **74.78** | **58.55** | 26.20 | **0.388** | **114.96** | **101.17** |
| | PSLD | Base | 27.69 | 0.265 | 63.95 | 63.47 | 25.21 | 0.373 | **88.92** | 95.21 |
| | | DiffState | **27.71** | **0.262** | 64.23 | 63.08 | **25.37** | **0.363** | 91.60 | **94.75** |
| | | Ours | 27.33 | 0.267 | **62.12** | **58.13** | 25.02 | 0.384 | 94.95 | 96.22 |
| | ReSample | Base | 26.40 | 0.347 | 70.16 | 133.15 | 23.37 | 0.435 | 72.22 | 134.39 |
| | | DiffState | 25.22 | 0.464 | 79.94 | 135.19 | 22.51 | 0.531 | 76.10 | 133.53 |
| | | Ours | **28.32** | **0.236** | **53.85** | **78.08** | **25.76** | **0.318** | **68.21** | **104.82** |
| | LatentDAPS | Base | **28.61** | **0.269** | 72.70 | **73.37** | 26.21 | **0.287** | **67.46** | **89.38** |
| | | DiffState | 28.58 | 0.270 | **72.08** | 73.92 | 26.22 | 0.288 | 71.16 | 90.32 |
| | | Ours | 28.50 | 0.270 | 75.25 | 73.79 | **26.25** | 0.290 | 70.86 | 89.78 |
| Inpainting (Random) | LDPS | Base | 31.22 | 0.171 | 48.88 | 83.30 | 28.78 | 0.238 | 57.12 | 119.25 |
| | | DiffState | 31.23 | 0.170 | 48.45 | 83.81 | 29.03 | **0.230** | 54.31 | 94.99 |
| | | Ours | **31.28** | **0.169** | **48.05** | **81.68** | **29.18** | 0.231 | **52.70** | **93.97** |
| | PSLD | Base | 30.14 | 0.222 | 58.84 | 79.59 | 27.85 | 0.300 | 70.51 | 96.30 |
| | | DiffState | 29.93 | 0.231 | 66.06 | 88.54 | 27.87 | 0.298 | 68.78 | 96.86 |
| | | Ours | **30.73** | **0.185** | **49.80** | **72.69** | **29.18** | **0.230** | **52.70** | **94.35** |
| | ReSample | Base | 27.27 | 0.374 | 103.17 | 133.80 | 24.84 | 0.489 | 132.87 | 151.96 |
| | | DiffState | 27.33 | 0.390 | 102.81 | 132.11 | 24.91 | 0.507 | 131.61 | 144.68 |
| | | Ours | **29.35** | **0.235** | **75.65** | **108.27** | **26.50** | **0.388** | **99.71** | **123.27** |
| | LatentDAPS | Base | 28.26 | **0.224** | 68.43 | **73.09** | 26.43 | 0.254 | 69.80 | 92.72 |
| | | DiffState | 28.27 | 0.225 | 65.99 | 73.10 | 26.35 | 0.256 | 69.89 | 92.17 |
| | | Ours | **28.29** | 0.225 | **64.49** | 73.70 | **26.51** | **0.253** | **68.84** | **90.99** |
| HDR | ReSample | Base | 25.75 | 0.197 | 80.44 | 85.46 | 24.91 | 0.218 | 74.43 | 91.78 |
| | | DiffState | **25.90** | 0.214 | 83.85 | 91.36 | **24.93** | **0.215** | 72.41 | 92.55 |
| | | Ours | 25.55 | **0.196** | **77.64** | **82.18** | 24.79 | 0.217 | **71.67** | **87.39** |
| | LatentDAPS | Base | 24.13 | 0.294 | **91.02** | 101.29 | **23.32** | 0.306 | 107.01 | 112.89 |
| | | DiffState | 24.12 | 0.295 | 91.57 | 100.64 | 23.31 | 0.306 | **104.67** | 112.96 |
| | | Ours | **24.52** | **0.293** | 91.12 | **99.03** | 23.25 | **0.305** | 106.91 | **112.76** |
| Nonlinear Deblur | ReSample | Base | 24.65 | **0.431** | 151.79 | **153.50** | 23.01 | 0.423 | 195.66 | 135.51 |
| | | DiffState | 24.61 | 0.432 | **142.54** | 154.07 | **23.10** | 0.424 | **182.52** | **135.05** |
| | | Ours | **24.84** | 0.449 | 148.42 | 159.76 | 22.96 | **0.421** | 185.63 | 136.69 |
| | LatentDAPS | Base | 24.48 | 0.481 | 152.40 | 152.80 | 22.58 | 0.515 | 186.81 | 148.47 |
| | | DiffState | **24.58** | 0.480 | 149.67 | 150.81 | **22.65** | 0.511 | **179.25** | **147.14** |
| | | Ours | 24.43 | **0.475** | **147.81** | **148.72** | 22.38 | **0.508** | 186.76 | 150.35 |

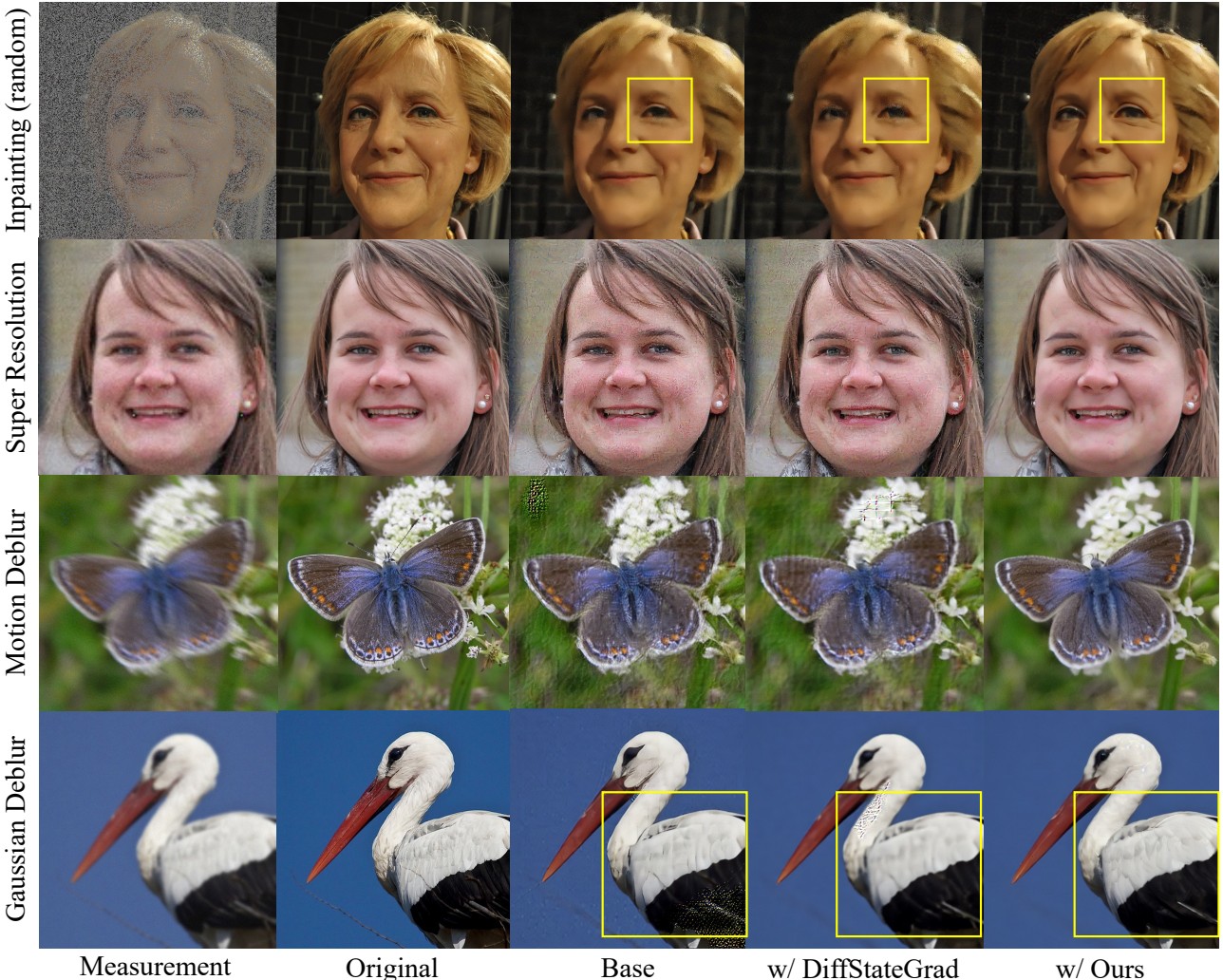

*Figure 4.* **Qualitative comparison of base latent diffusion inverse solvers and their plug-in versions with DiffStateGrad (Zirvi et al., 2025) and MCLC (ours)**. The proposed MCLC yields more stable and reliable solutions by effectively alleviating artifacts and enhancing reconstruction quality. The baseline used for visualization is ReSample (top two rows) and PSLD (bottom two rows).

et al., 2023), ReSample (Song et al., 2024), and Latent-DAPS (Zhang et al., 2025). This evaluation protocol allows us to directly assess the stabilization effect of MCLC across diverse solvers without modifying their core algorithms. We further compare MCLC with DiffStateGrad (Zirvi et al., 2025), a recent plug-and-play method that relies on a linear manifold assumption. The experiments adopt Stable Diffusion v1.5 (SD v1.5) as the underlying latent diffusion model. For reproducibility, further details—including solver-MCLC integration algorithms, hyperparameters of MCLC, as well as the configurations of solvers, samplers, and other settings—are provided in the Appendix. D.

We benchmark the method across both linear and nonlinear inverse problems using two image datasets, FFHQ (Karras et al., 2019) and ImageNet (Deng et al., 2009). Following the data protocol (Zhang et al., 2025; Zirvi et al., 2025), we selected 100 validation images from each dataset (*i.e.*, the first 100 images of each validation set). For linear tasks, we consider (1) Super Resolution with a downscaling factor of 4 using a bicubic resizer, (2) Gaussian Deblur with a $121 \times 121$ kernel and standard deviation $\sigma = 3.0$, (3) Motion Deblur with a $121 \times 121$ kernel and standard deviation $\sigma = 0.5$, and (4) inpainting with 70% random pixel dropout. For nonlinear tasks, we evaluate (5) High Dynamic Range (HDR) reconstruction with oversampling rate 2.0 and (6) Nonlinear Deblur with $64 \times 64$ kernel. All experiments are conducted at a resolution of $512 \times 512$ with a Gaussian noise scale fixed to $\sigma = 0.03$, except for nonlinear deblurring, where the blur kernel is generated for $256 \times 256$. For the evaluation metric, we adopt PSNR, LPIPS, and FID following previous works. In addition, we introduce Patch-FID (P-FID) to more effectively quantify regional artifacts by comparing patch-wise statistics. To implement P-FID, we split each image into $3 \times 3$ patches and treat them as individual images to calculate the FID score.

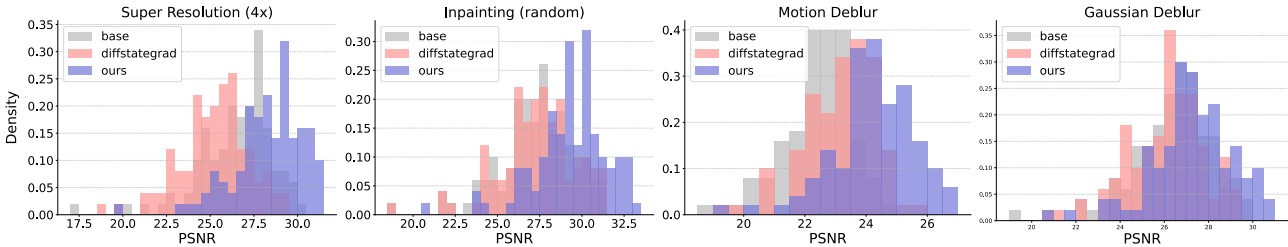

*Figure 5.* **PSNR histograms highlighting stabilization effect.** Compared to manifold-preserving comparison DiffStateGrad, MCLC exhibits a stronger stabilization effect, shifting the distributions toward higher PSNR values and reducing low-PSNR failures across tasks. In this figure, ReSample (Song et al., 2024) is used as the base solver.

*Table 2.* **Quantitative comparison with non-pluggable stabilization approaches.** MCLC, a fully pluggable method, achieves competitive overall performance while offering broad applicability.

| Method | Super Resolution (4×) | | | Gaussian Deblur | | |
|---|---|---|---|---|---|---|
| | PSNR↑ | LPIPS↓ | FID↓ | PSNR↑ | LPIPS↓ | FID↓ |
| MPGD | 27.25 | 0.280 | 77.90 | 28.69 | 0.260 | 75.52 |
| SILO | 25.91 | 0.251 | 70.11 | 25.79 | 0.276 | 76.18 |
| SITCOM | 26.77 | 0.397 | 103.95 | 26.45 | 0.426 | 109.06 |
| PSLD w/ Ours | 27.33 | 0.267 | 62.12 | 27.97 | 0.286 | 66.28 |
| ReSample w/ Ours | 28.32 | 0.236 | 53.85 | 27.25 | 0.353 | 78.38 |

*Table 3.* **Quantitative results under more severe degradation settings.** We evaluate MCLC on 92% random inpainting on FFHQ and ×12 super-resolution on ImageNet, which are more severe than the main settings of 70% random inpainting and ×4 super-resolution. ReSample is used as the base solver.

| | Random Inpainting 92% (FFHQ) | | | Super-Resolution 12× (ImageNet) | | |
|---|---|---|---|---|---|---|
| | PSNR↑ | LPIPS↓ | FID↓ | PSNR↑ | LPIPS↓ | FID↓ |
| Base | 21.51 | 0.758 | 177.81 | 20.66 | 0.616 | 265.82 |
| Ours | 25.01 | 0.524 | 128.18 | 22.76 | 0.606 | 222.05 |

*Table 4.* **Quantitative results under increasing measurement noise.** We evaluate the robustness of MCLC under 92% random inpainting on FFHQ across varying measurement noise levels. ReSample is used as the base solver.

| Noise level | Method | PSNR (↑) | LPIPS (↓) | FID (↓) | P-FID (↓) |
|---|---|---|---|---|---|
| $\sigma = 0.01$ | Base | 21.69 | 0.741 | 173.10 | 219.09 |
| | Ours | **25.15** | **0.509** | **124.52** | **150.35** |
| $\sigma = 0.03$ | Base | 21.51 | 0.758 | 177.10 | 225.62 |
| | Ours | **25.01** | **0.524** | **128.18** | **152.86** |
| $\sigma = 0.05$ | Base | 21.26 | 0.783 | 181.01 | 227.52 |
| | Ours | **24.80** | **0.547** | **128.79** | **154.53** |

*Table 5.* **Quantitative results on AFHQ-val 1K using TReg** (Kim et al., 2025b). To further validate the applicability of MCLC with recent advanced solvers, we report comparisons with and without MCLC using TReg, a recent latent diffusion inverse solver.

| | Gaussian Deblur | | | Super Resolution (16×) | | |
|---|---|---|---|---|---|---|
| | PSNR↑ | LPIPS↓ | FID↓ | PSNR↑ | LPIPS↓ | FID↓ |
| TReg | 20.84 | 0.476 | 37.12 | 18.39 | **0.633** | 44.91 |
| TReg w/ Ours | **21.33** | **0.456** | **27.62** | **19.15** | 0.646 | **33.86** |

**Experimental results.** We quantitatively demonstrate the benefits of integrating MCLC into existing LDM-based inverse solvers in Table 1. By stabilizing solver dynamics, MCLC consistently improves overall performance without any modification to the original solver designs. Compared to DiffStateGrad, a manifold-preserving plug-and-play approach based on linear manifold assumptions, MCLC shows clear advantages for inverse solvers operating in latent space. Perceptual b that reflect sample plausibility and stability, such as LPIPS and FID, show substantial gains. At the same time, since MCLC improves the stability of existing solvers without sacrificing measurement fidelity, PSNR shows little to no degradation and is improved depending on the stabilization effect. Additionally, Fig. 4 shows that MCLC produces faithful reconstructions and yields more reliable solutions, with further qualitative results provided in the Appendix. C.1. This stabilization effect is further demonstrated in the PSNR histograms in Fig. 5, where the distributions consistently shift toward higher values across tasks and significantly reduce severe low-PSNR failure cases, indicating improved stability and overall effectiveness.

We also compare our method against non-pluggable approaches for LDM-based inverse problem solvers, including

MPGD (He et al., 2024), which performs autoencoder-based manifold projection; SILO (Raphaeli et al., 2025), which employs a trained degradation operator to avoid decoder backpropagation; and SITCOM (Alkhouri et al., 2025), a recent stabilization method. Table 2 demonstrates the effectiveness of our method, even though it is employed as a plug-and-play module. In contrast, these approaches often leave artifacts and degraded quality or compromise the measurement fidelity. Further discussions are in Appendix. C.2.

**More Severe Degradation Settings.** To further assess MCLC under highly ill-posed scenarios, we evaluate it on more severe degradations, including 92% random inpainting on FFHQ and ×12 super-resolution on ImageNet. These experiments allow us to more thoroughly examine the robustness of MCLC under challenging settings. As shown in Table 3, MCLC improves the stability and reconstruction quality of the base solver, demonstrating its effectiveness under severe degradations. We further note that all experiments use a general-purpose prior, such as SD v1.5, rather than a domain-specific prior (*e.g.*, FFHQ-LDM). Since general-purpose priors induce a broader solution space, these results further support the robustness of MCLC in challenging and general inverse problem settings.

*Table 6.* **Quantitative results on FFHQ using FlowChef** (Patel et al., 2025). We report drop-in performance improvements on FlowChef, a recent latent flow-based inverse solver, demonstrating the applicability of MCLC to latent flow–based inverse solvers.

| | Motion Deblur | | | Gaussian Deblur | | | SR 12× (Bicubic) | | | SR 12× (Avgpool) | | |
|---|---|---|---|---|---|---|---|---|---|---|---|---|
| Method | PSNR↑ | LPIPS↓ | FID↓ | PSNR↑ | LPIPS↓ | FID↓ | PSNR↑ | LPIPS↓ | FID↓ | PSNR↑ | LPIPS↓ | FID↓ |
| FlowChef | 22.58 | 0.519 | 185.44 | 23.76 | 0.364 | 106.76 | **25.30** | 0.501 | 174.51 | **25.26** | 0.480 | 181.15 |
| FlowChef w/ Ours | **26.01** | **0.353** | **100.40** | **28.52** | **0.288** | **77.36** | 24.85 | **0.424** | **130.70** | 24.81 | **0.393** | **125.57** |

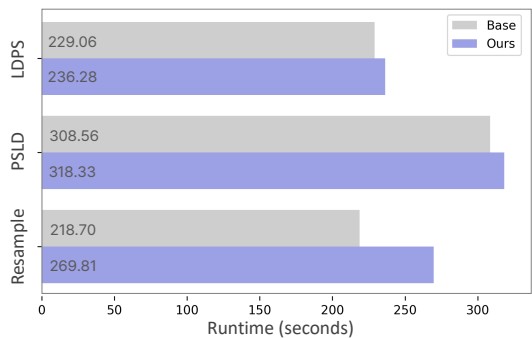

*Figure 6.* **Runtime overhead of MCLC.** The additional computational cost introduced by MCLC is within a reasonable range.

**What happens if the measurement gradient is poor?**
We further examine the robustness of MCLC when the measurement-consistency gradient becomes less reliable. To simulate this setting, we evaluate MCLC under sparse measurements with increasing measurement noise. Specifically, we use 92% random inpainting on FFHQ, which makes the inverse problem highly ill-posed due to limited observations, and vary the measurement noise level. As shown in Table 4, MCLC remains effective even as the measurement noise increases, demonstrating its robustness under sparse and noisy measurement settings. This robustness is supported by Theorem 3.4: regardless of the quality of the measurement gradient, MCLC preserves measurement-consistency up to the first-order term at the current state, with higher-order perturbations controlled by the step size, while its KL-reduction property still holds within the projected subspace. However, in regimes where the measurement gradient is severely inaccurate, the underlying solver may not be reliable, which can limit effectiveness of MCLC.

**Broader applicability.** We show that MCLC can be applied not only to recently proposed LDM-based solvers (Kim et al., 2025b) with more specialized formulations but also to latent flow–based solvers (Patel et al., 2025). MCLC is designed to stabilize solver dynamics that evolve according to reverse-time dynamics; therefore, it is applicable to most diffusion- and flow-based solvers that commonly follow this formulation. In particular, since latent flow models are also trained to match the time-marginal distributions, MCLC is naturally applicable to stabilizing their solver dynamics. As shown in Table 5 and 6, MCLC demonstrates clear and consistent gains. In these experiments, we follow the original implementations and experimental protocols for each solver and task. The underlying More detailed settings and discus-

*Table 7.* **Ablation study on measurement-consistent projection** "w/ MC" denotes MCLC and "w/o MC" indicates vanilla Langevin correction (LC). Results are on FFHQ 4× SR using LDPS.

| Method | y-PSNR (↑) | PSNR (↑) | LPIPS (↓) | FID (↓) |
|---|---|---|---|---|
| Ours (w/o MC) | 31.59 | 27.50 | **0.277** | 76.78 |
| Ours (w/ MC) | **33.23** | **28.34** | 0.283 | **74.78** |

sions are provided in Appendix. D.2. Notably, despite using a single default hyperparameter configuration for MCLC across all tasks, we observe consistent performance gains.

For LatentDAPS, performance differences are marginal because its specific design breaks the reverse diffusion dynamics by re-initializing each iteration with annealed noise, whereas MCLC is intended to stabilize the reverse dynamics of LDM-based inverse solvers; this does not imply limited applicability. Detailed discussion is in Appendix. B.2.

**Additional computational cost.** We report the additional runtime and memory introduced by MCLC across three solvers (LDPS, PSLD, and Resample). As shown in Fig. 6, the additional wall-clock time introduced by MCLC is modest for LDPS and PSLD (about 3%). For ReSample, the increase is more noticeable because the base solver already performs extensive inner gradient-descent loops for hard data consistency. Even in this case, the overall overhead remains manageable, and MCLC provides substantial improvements in reconstruction quality, as shown in Table 1. This efficiency stems from the fact that MCLC requires only the LDM forward pass and simple algebraic operations, without any backward computation. While backward computation requires several times more computational cost for large prior models, MCLC avoids this backward process by reusing the gradient from the measurement-consistency step. As a result, MCLC incurs no additional memory overhead, as confirmed by the peak memory usage across all solvers. For all experiments, we use a single NVIDIA RTX A6000.

**Ablation study on measurement-consistent projection.**
As discussed in Remark 3.3, we ablate the measurement-consistent projected Langevin by comparing MCLC with vanilla Langevin correction (LC). As shown in Table 7, compared to LC, MCLC yields higher y-PSNR (*i.e.*, PSNR on measurements), indicating better preservation of measurement-consistency while achieving stabilization.

**Ablation study on the number of corrector steps.** The number of corrector steps is an important hyperparameter,

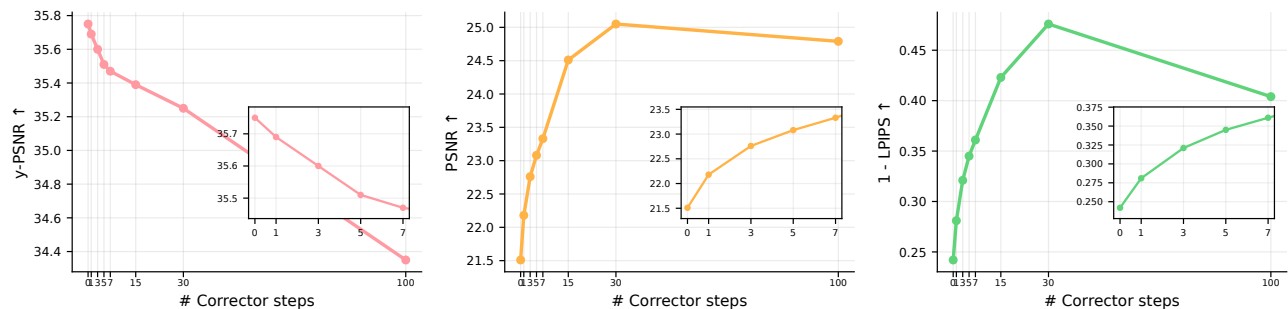

**Figure 7. Effect of the number of corrector steps.** We plot measurement consistency (y-PSNR) and stabilization metrics (PSNR and $1 - \text{LPIPS}$) as the number of corrector steps increases on FFHQ 92% random inpainting using LDPS. While y-PSNR gradually decreases due to accumulated perturbation bound (largely preserving), PSNR and $1 - \text{LPIPS}$ improve, showing the stabilization effect of MCLC.

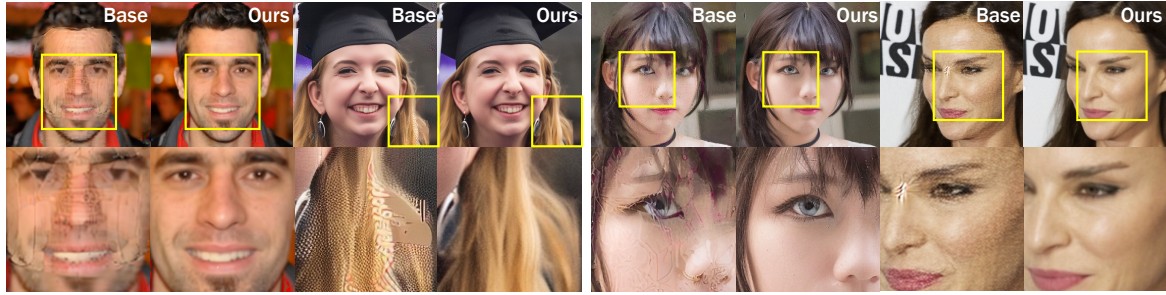

Stable Diffusion v2.1        Realistic Vision v5.1

**Figure 8. Qualitative results under different LDM priors** Gaussian deblurring (left 2×2) and super resolution (right 2×2) results are shown for Stable Diffusion v2.1 and Realistic Vision v5.1 Instability appears across priors, and MCLC mitigates it.

*Table 8.* **Quantitative results on FFHQ with other LDMs.** Across different LDM priors, including Realistic Vision (RV) v5.1 and Stable Diffusion (SD) v2.1, instability can be alleviated by MCLC. In this table, PSLD is used as the base solver.

| Prior | Method | Gaussian Deblur | | | | Super Resolution (4×) | | | |
|---|---|---|---|---|---|---|---|---|---|
| | | PSNR (↑) | LPIPS (↓) | FID (↓) | P-FID (↓) | PSNR (↑) | LPIPS (↓) | FID (↓) | P-FID (↓) |
| SD v2.1 | Base | 26.22 | 0.403 | 116.49 | 120.94 | **28.70** | **0.244** | 60.34 | 49.99 |
| | Ours | **26.69** | **0.335** | **91.13** | **52.76** | 28.64 | 0.246 | **60.33** | **48.30** |
| RV v5.1 | Base | 26.69 | 0.380 | 110.38 | 78.25 | **27.96** | 0.305 | 71.55 | 54.91 |
| | Ours | **26.75** | **0.334** | **95.46** | **56.66** | 27.82 | **0.295** | **68.20** | **50.10** |

together with the step size, as increasing it can further reduce the discrepancy and improve stabilization. However, as shown in Theorem 3.4, each correction step preserves measurement consistency only up to a controlled perturbation bound, so residual errors can accumulate across repeated steps. As shown in Fig. 7, y-PSNR gradually decreases as the number of corrector steps increases, reflecting this error accumulation. Nevertheless, MCLC provides stabilization benefits over a wide range of corrector steps, with measurement-consistency degradation remaining within an acceptable range even with many correction steps. When the number of steps becomes excessive, however, accumulated perturbations can lead to a larger drop in measurement consistency, which in turn reduces the stabilization effect.

**Generalizability across LDM priors.** We evaluate MCLC on different LDM priors to verify that the stabilization effect generalizes across priors. Results are shown in Table 8 and Fig. 8, using PSLD and the configurations as in Table 1.

## 5. Conclusion

In this work, we present a new perspective on the instability of latent diffusion inverse problem solvers and propose MCLC, a principled stabilization scheme. We characterize solver instability as a discrepancy between solver-induced dynamics and the learned reverse dynamics, offering a concrete characterization without relying on geometric manifold assumptions. Based on this characterization, we address this instability with MCLC, a novel measurement-consistent stabilization scheme that reduces the discrepancy without sacrificing data fidelity. As a plug-and-play module, MCLC can be readily integrated into existing LDM-based inverse solvers and provides more faithful and stable solutions in practice. We believe this work offers a theoretically grounded basis for understanding instability in latent diffusion- and flow-based inverse solvers, provides a principled direction for stabilizing, and inspires further research toward reliable zero-shot inverse problem solvers.

## Acknowledgements

This work was supported by the National Research Foundation of Korea (NRF) grants (Nos. RS-2024-00358135, 5%, Corner Vision: Learning to Look Around the Corner through Multi-modal Signals; and RS-2024-00453301, 20%), and by the Institute of Information & Communications Technology Planning & Evaluation (IITP) grants (Nos. RS-2024-00457882, 25%, National AI Research Lab Project; and RS-2025-25441313, 25%, Professional AI Talent Development Program for Multimodal AI Agents) funded by the Korea government (MSIT). E. Cha was partially supported by the NRF grant funded by the Korea government (MSIT) (No. RS-2024-00357197, 25%).

## Impact statement

This paper presents work whose goal is to advance the field of machine learning. There are many potential societal consequences of our work, none of which we feel must be specifically highlighted here.

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

## Appendix

### A. Proofs

### B. Additional Discussion and Analysis

### C. Additional Results

### D. Implementation Details

## A. Proofs

**Notation.** We summarize the notation used in the proofs:

- $\nabla f$: gradient of a scalar function $f$, i.e., $(\partial_{x_1} f, \ldots, \partial_{x_d} f)^\top$.

- $\nabla \cdot F$: divergence of a vector field $F = (F_1, \ldots, F_d)$, i.e., $\sum_{i=1}^d \partial_{x_i} F_i$.

- $\Delta z_t$: increment (update change) of the variable $z_t$ in algorithmic updates, not to be confused with the Laplacian operator.

- $\mathrm{KL}(q \| p)$: Kullback–Leibler divergence, defined as $\int q(x) \log\left(\frac{q(x)}{p(x)}\right) dx$.

**Proposition** (Langevin Corrector). *Fix a timestep $t$ and let $p_t$ be a target distribution. Consider the continuous corrector process $\{Z_t^c\}_{c \geq 0}$ initialized with $Z_t^0 \sim q_t^\#$. The process evolves according to the Langevin dynamics with frozen target $p_t$: $dZ_t^c = \nabla \log p_t(Z_t^c) dc + \sqrt{2} dW_c$. Let $q_t^c$ denote the distribution of $Z_c$. Then, the KL divergence monotonically decreases along the process, unless $q_t^c = p_t$, in which case equality holds:*

$$\mathcal{D}_{\mathrm{KL}}(q_t^c \| p_t) \leq \mathcal{D}_{\mathrm{KL}}(q_t^\# \| p_t), \qquad \forall c \geq 0. \tag{9}$$

*Proof.* We recall that the Langevin Corrector dynamics is given by

$$dZ_c = \nabla \log p_t(Z_c) \, dc + \sqrt{2} \, dW_c, \qquad c \geq 0, \tag{10}$$

where $p_t$ is a fixed target distribution at timestep $t$. In this corrector process, keeping $t$ fixed, the KL divergence with respect to $p_t$ decreases monotonically (Durmus & Moulines, 2019; Vempala & Wibisono, 2019). For notational simplicity, when time $t$ is fixed in the corrector process, we write $Z_t^c$ as $Z_c$ below.

*Step 1. From SDE to Fokker–Planck PDE.* Let $\{X_t\}_{t\geq 0}$ be a stochastic process. At each time $t$, the random variable $X_t$ evolves according to the stochastic differential equation:

$$dX_t = a(X_t, t)\, dt + b(X_t, t)\, dW_t, \tag{11}$$

where $a(X_t, t)$ denotes the drift term, $b(X_t, t)$ the diffusion coefficient, and $W_t$ a standard Wiener process. The random variable $X_t$ has a distribution, denoted by $q_t$, which evolves over time. The time evolution of this distribution $q_t$ (*i.e.*, the law of $X_t$) is governed by the Fokker–Planck equation:

$$\frac{\partial q(x,t)}{\partial t} = -\nabla \cdot \big(a(x,t)\, q(x,t)\big) + \nabla \cdot \big(D(x,t)\, \nabla q(x,t)\big). \tag{12}$$

where $D(x,t) = \frac{1}{2} b(x,t)\, b(x,t)^\top$.

*Step 2. Langevin Corrector.* From Eq. (11), the Langevin Corrector dynamics can be written as the following SDE:

$$dZ_c = \nabla \log p_t(Z_c)\, dc + \sqrt{2}\, dW_c, \qquad c \geq 0, \tag{13}$$

with drift $a(x) = \nabla \log p_t(x)$ and isotropic diffusion $b(x) = \sqrt{2}\, I$, where $p_t$ is a fixed target distribution at timestep $t$. Let $q_t^c$ denote the distribution of $Z_c$, which evolves along the corrector process with the timestep $t$ fixed. Then, its evolution with respect to the corrector-time variable $c$ is also described by the Fokker–Planck equation:

$$\frac{\partial q_t^c}{\partial c} = \nabla \cdot \Big(q_t^c \nabla \log \frac{q_t^c}{p_t}\Big). \tag{14}$$

*Step 3. Evolution of the KL divergence along the corrector process.* Consider the KL divergence

$$\mathcal{F}[q_t^c] = \mathrm{KL}(q_t^c \,\|\, p_t) = \int q_t^c(x) \log \frac{q_t^c(x)}{p_t(x)}\, dx. \tag{15}$$

Differentiating with respect to $c$ yields,

$$\frac{d}{dc}\mathcal{F}[q_t^c] = \int \Big(1 + \log \frac{q_t^c}{p_t}\Big) \frac{\partial q_t^c}{\partial c}\, dx \tag{16}$$

$$= \int \Big(1 + \log \frac{q_t^c}{p_t}\Big) \nabla \cdot \Big(q_t^c \nabla \log \frac{q_t^c}{p_t}\Big) dx. \quad \text{(by Eq. (14))} \tag{17}$$

$$= -\int q_t^c(x) \Big\|\nabla \log \frac{q_t^c(x)}{p_t(x)}\Big\|^2 dx. \tag{18}$$

Then, following the Langevin Corrector dynamics, $\frac{d}{dc}\mathcal{F}[q_t^c]$ is written as Eq. (18). Since $q_t^c(x) \geq 0$ and $\big\|\nabla \log \frac{q_t^c(x)}{p_t(x)}\big\|^2 \geq 0$, the right-hand side is non-positive. Moreover, it equals zero if and only if $q_t^c = p_t$; otherwise it is strictly negative:

$$\frac{d}{dc}\mathrm{KL}(q_t^c \| p_t) \; \leq \; 0. \tag{19}$$

Therefore, the KL divergence between $q_t^c$ and the fixed target distribution $p_t$ monotonically decreases along the corrector process. In particular, the KL divergence of the corrected distribution is no larger than that of the problematic distribution obtained from the inverse solver update.

$$\mathrm{KL}(q_t^c \| p_t) \leq \mathrm{KL}(q_t^\# \| p_t), \qquad \forall c \geq 0. \tag{20}$$

This proves the proposition. $\qquad\qquad\qquad\qquad\qquad\qquad\qquad\qquad\qquad\qquad\qquad\qquad\qquad\qquad\qquad\qquad\qquad\qquad\qquad\quad\Box$

**Lemma.** *Let $U \sim \mathcal{N}(\mu, \Sigma)$ be a Gaussian random vector in $\mathbb{R}^d$, in the high-dimensional setting where $d$ is large. Then there exists a universal constant $\kappa > 0$ such that*

$$\mathbb{E}[\|U\|^3] \; \leq \; \kappa (\mathbb{E}[\|U\|^2])^{3/2}. \tag{21}$$

*Proof.* Let h: $\mathbb{R}_+ \to \mathbb{R}$ be twice differentiable, and suppose that $h''(x) \leq \Lambda$ for $x$ in the support of a random variable $Y$. Define

$$g(x) = h(x) - \tfrac{1}{2}\Lambda x^2. \tag{22}$$

Since $g''(x) = h''(x) - \Lambda \leq 0$, the function $g$ is concave. By Jensen's inequality,

$$\mathbb{E}[g(Y)] \leq g(\mathbb{E}[Y]). \tag{23}$$

Then,

$$\mathbb{E}[h(Y) - \tfrac{1}{2}\Lambda Y^2] \leq h(\mathbb{E}[Y]) - \tfrac{1}{2}\Lambda\mathbb{E}[Y]^2 \tag{24}$$

$$\mathbb{E}[h(Y)] - h(\mathbb{E}[Y]) \leq \underbrace{\tfrac{1}{2}\Lambda\,\mathbb{E}[Y^2] - \tfrac{1}{2}\Lambda\,\mathbb{E}[Y]^2}_{\tfrac{1}{2}\Lambda\,\mathrm{Var}(Y)}. \tag{25}$$

Let $Y = \|U\|^2$ and $h(x) = x^{3/2}$. Then,

$$h''(x) = \tfrac{3}{4}x^{-1/2}, \quad \text{so } \Lambda = O(x^{-1/2}). \tag{26}$$

Hence, by Eq. (25),

$$\mathbb{E}[h(Y)] - h(\mathbb{E}[Y]) \leq O\left(\frac{\mathrm{Var}(Y)}{\sqrt{\mathbb{E}[Y]}}\right) \tag{27}$$

$$= O\left(\frac{\mathrm{Var}(Y)}{\mathbb{E}[Y]^2}\mathbb{E}[Y]^{3/2}\right) \tag{28}$$

If the $U \in \mathbb{R}^d$ follows a Gaussian distribution in high dimensions, $\|U\|^2$ is concentrated around its mean, so that $\mathrm{Var}(Y)$ grows much more slowly than $\mathbb{E}[Y]^2$ (while $\mathrm{Var}(Y)$ grows only linearly with $d$, $(\mathbb{E}[Y])^2$ grows quadratically). Therefore, we may write

$$\mathbb{E}[h(Y)] - h(\mathbb{E}[Y]) \leq \delta\,\mathbb{E}[Y]^{3/2}, \qquad \text{for some small } \delta > 0, \tag{29}$$

where $\delta$ converges to 0 as the dimension $d \to \infty$.

Consequently,

$$\mathbb{E}[Y^{3/2}] \leq \underbrace{(1 + \delta)}_{:=\kappa}\,\mathbb{E}[Y]^{3/2} \tag{30}$$

Hence, there exists a universal constant $\kappa > 0$ such that

$$\mathbb{E}[\|U\|^3] \leq \kappa(\mathbb{E}[\|U\|^2])^{3/2}. \tag{31}$$

$\square$

**Theorem.** *The projected Langevin update onto the orthogonal complement of the measurement gradient decreases the KL divergence while preserving measurement consistency up to a controlled bound. If the update satisfies*

$$\mathbb{E}[\|\Delta z_t\|^2] \leq k < 1, \tag{32}$$

*the expected measurement consistency perturbation follows:*

$$\mathbb{E}[\Delta r] \leq Ck + O(k), \tag{33}$$

*for some $C > 0$ depending on local smoothness of $r$.*

*Proof.* The proof of Theorem 1 consists of two parts: (i) showing the decrease of the KL divergence, and (ii) establishing the measurement-consistency bound.

*Proof 1. KL divergence decrease.* As proved in Proposition 1, the Measurement-Consistent Langevin Corrector (MCLC) dynamics is written as:

$$dZ_c = P_{\perp \boldsymbol{g}} \nabla \log p_t(Z_c) \, dc + \sqrt{2} \, P_{\perp \boldsymbol{g}} \, dW_c, \tag{34}$$

where $P_{\perp \boldsymbol{g}} = I - \frac{\boldsymbol{g}\boldsymbol{g}^\top}{\|\boldsymbol{g}\|^2}$ denotes the orthogonal projection onto the complement of the measurement gradient direction.

By the Fokker–Planck equation, the evolution of the distribution $q_t^c$ under the MCLC process is

$$\frac{\partial q_t^c}{\partial c} = \nabla \cdot \left( q_t^c \, P_{\perp \boldsymbol{g}} \, \nabla \log \frac{q_t^c}{p_t} \right). \tag{35}$$

Hence, the evolution of the KL divergence along the corrector process is

$$\frac{d}{dc} \mathcal{F}[q_t^c] = \int \left( 1 + \log \frac{q_t^c}{p_t} \right) \frac{\partial q_t^c}{\partial c} \, dx \tag{36}$$

$$= \int \left( 1 + \log \frac{q_t^c}{p_t} \right) \nabla \cdot \left( q_t^c \, P_{\perp \boldsymbol{g}} \nabla \log \frac{q_t^c}{p_t} \right) dx \quad \text{(by Eq. (35))} \tag{37}$$

$$= - \int q_t^c(x) \left\| P_{\perp \boldsymbol{g}} \nabla \log \frac{q_t^c(x)}{p_t(x)} \right\|^2 dx. \tag{38}$$

Then, the MCLC dynamics guarantees that the KL divergence monotonically decreases in the corrector step $c$, whenever the score difference $\nabla \log q_t^c - \nabla \log p_t$ has a non-zero component in the orthogonal complement of the measurement-consistent subspace:

$$\mathrm{KL}(q_t^c \| p_t) \leq \mathrm{KL}(q_t^\# \| p_t), \qquad \forall c \geq 0, \tag{39}$$

where, since $P_{\perp \boldsymbol{g}}$ is projection matrix, $\left\| P_{\perp \boldsymbol{g}} \nabla \log \frac{q_t^c(x)}{p_t(x)} \right\|^2 \geq 0$.

The score difference characterizes how the local geometry of the current corrected distribution $q_t^c$ deviates from that of the target distribution $p_t$. In other words, when the two distributions have divergence in the orthogonal complement of the measurement-consistent gradient, the KL divergence strictly decreases.

*Proof 2. Measurement consistency error bound.* Let $r : \mathbb{R}^d \to \mathbb{R}$ denote the measurement residual (*e.g.*, measurement-consistency loss). Let the MCLC step $\Delta z_t$:

$$\Delta z_{t,\perp} = \eta_t \, P_{\perp \boldsymbol{g}} s_t + \sqrt{2\eta_t} \, P_{\perp \boldsymbol{g}} \, \epsilon, \qquad \text{with } z_t \in \mathbb{R}^d, \ \epsilon \sim \mathcal{N}(0, I), \tag{40}$$

$$\Delta z_{t,\perp} = P_{\perp \boldsymbol{g}} \Big( \underbrace{\eta_t s_t + \sqrt{2\eta_t} \, \epsilon}_{:= \Delta z_t} \Big). \tag{41}$$

For convenience, denote the score as $s_t = \nabla \log p_t(z_t)$.

*Step 1. Bound of residual perturbation along the MCLC step.* By Taylor's expansion, the residual after one MCLC step can be expressed in terms of the residual before the step as:

$$\mathbb{E}[\, r(z_t + \Delta z_t)\,] = \mathbb{E}[\, r(z_t)\,] + \mathbb{E}[\nabla r(z_t)^\top \Delta z_t] + \tfrac{1}{2}\mathbb{E}[\Delta z_t^\top H \Delta z_t] + \mathbb{E}[O(r^3)], \tag{42}$$

where $H$ is the Hessian of $r(z_t)$, and $O(r^3)$ denotes the higher-order terms.

Then, the change in residual after one MCLC step is given by:

$$\underbrace{r(z_t + \Delta z_{t,\perp}) - r(z_t)}_{:= \Delta r(z_t)} = \underbrace{\nabla r(z_t)^\top \Delta z_{t,\perp}}_{=0} + \tfrac{1}{2} \Delta z_{t,\perp}^\top H \Delta z_{t,\perp} + O(r^3). \tag{43}$$

Using $\Delta z_t$, the $\Delta r(z_t)$ can be written as:

$$\Delta r(z_t) = \tfrac{1}{2}\,\Delta z_t^\top \underbrace{P_{g\perp}^\top H P_{g\perp}}_{:=H_{g\perp}} \Delta z_t + O(r^3). \tag{44}$$

$$= \tfrac{1}{2}\,\Delta z_t^\top H_{g\perp}\Delta z_t + O(r^3). \tag{45}$$

By assuming local Hessian Lipschitz continuity, the higher-order terms can be controlled as $O(\|\Delta z_t\|^3)$, that is bounded by Lipschitz bound $\frac{L}{6}\|\Delta z_t\|^3$ on the cubic term. In addition, since $H_{g\perp} = P_{\perp g}^\top H P_{\perp g}$ is symmetric, the second-order term can be bounded via the Rayleigh quotient:

$$\tfrac{1}{2}\lambda_{min,\perp}\|\Delta z_t\|^2 \leq \tfrac{1}{2}\,\Delta z_t^\top H_{g\perp}\Delta z_t \leq \tfrac{1}{2}\lambda_{max,\perp}\|\Delta z_t\|^2, \tag{46}$$

where $\lambda_{min,\perp}$ and $\lambda_{max,\perp}$ denote the minimum and maximum eigenvalues of $H_{g\perp}$, respectively.

Then, $\Delta r(z_t)$ is bounded as follows:

$$\Delta r(z_t) \leq \tfrac{1}{2}\lambda_{max,\perp}\|\Delta z_t\|^2 + O(\|\Delta z_t\|^3). \tag{47}$$

Because the random variable is contained in $\Delta z_t$, we derive the bound in expectation as:

$$\mathbb{E}[\Delta r(z_t)] \leq \tfrac{1}{2}\lambda_{max,\perp}\mathbb{E}[\|\Delta z_t\|^2] + O(\mathbb{E}[\|\Delta z_t\|^3]). \tag{48}$$

In our formulation, the random vector $U \in \mathbb{R}^d$ corresponds to a single step of Langevin dynamics, which yields a Gaussian distribution in the high-dimensional setting. By Lemma A, the cubic term can therefore be controlled in terms of the second moment. Hence,

$$\mathbb{E}[\Delta r(z_t)] \leq \tfrac{1}{2}\lambda_{max,\perp}\mathbb{E}[\|\Delta z_t\|^2] + O\big(\mathbb{E}[\|\Delta z_t\|^2]^{3/2}\big). \tag{49}$$

Suppose that the second moment is controlled as

$$\mathbb{E}[\|\Delta z_t\|^2] \;\leq\; k \;<\; 1. \tag{50}$$

Then, by Eq. (49), the residual perturbation is bounded by

$$\mathbb{E}[\Delta r(z_t)] \leq \tfrac{1}{2}\lambda_{max,\perp}\mathbb{E}[\|\Delta z_t\|^2] + O\big(\mathbb{E}[\|\Delta z_t\|^2]^{3/2}\big) \leq Ck + O(k^{3/2}), \tag{51}$$

where the constant $C = \tfrac{1}{2}\lambda_{max,\perp}$ depends only on the local smoothness of $r$. Since $k \;<\; 1$, the cubic term bound is of order $O(k)$. Moreover, because $k = \mathbb{E}[\|\Delta z_t\|^2]$ is determined by $\eta_t$, we can choose $\eta_t$ that preserves measurement consistency at the current timestep while reducing the KL divergence.

*Step 2. The step size second moment.* We expand the second moment as

$$\mathbb{E}[\|\Delta z_t\|^2] = \mathbb{E}[\|\Delta z_t\|^2] \tag{52}$$

$$= \mathbb{E}[\|\eta_t s_t + \sqrt{2\eta_t}\,\epsilon\|^2] \tag{53}$$

$$= \mathbb{E}[\eta_t^2\|s_t\|^2 + 2\eta_t\|\epsilon\|^2] + \underbrace{\mathbb{E}[2\eta_t\sqrt{2\eta_t}s_t^\top\epsilon]}_{=0} \tag{54}$$

$$= \mathbb{E}[\eta_t^2\|s_t\|^2 + 2\eta_t\|\epsilon\|^2], \tag{55}$$

where $\mathbb{E}[e] = 0$.

Let the adaptive step size be parameterized as

$$\eta_t = \frac{\|\epsilon\|^2}{\|s_t\|^2}\lambda, \qquad \lambda > 0, \tag{56}$$

where $\lambda$ is a constant hyperparameter balancing the drift and diffusion terms as proposed in (Song et al., 2021). Under this parameterization, the second moment becomes:

$$\mathbb{E}[\|\Delta z_t\|^2] = \lambda^2 \, \mathbb{E}\!\left[\frac{\|\epsilon\|^4}{\|s_t\|^2}\right] + 2\lambda \, \mathbb{E}\!\left[\frac{\|\epsilon\|^4}{\|s_t\|^2}\right]. \tag{57}$$

Since $\epsilon \sim \mathcal{N}(0, I)$, we have $\mathbb{E}[\|\epsilon\|^4] = d\,(d+2)$. Moreover, by the concentration of measure in high dimension (Chung et al., 2023), the squared norm $\|\epsilon_\theta\|^2$ concentrates sharply around its mean $d$. For a well-trained diffusion model, the network prediction $\epsilon_\theta$ approximates $\epsilon$ in distribution. Therefore, we can assume $\mathbb{E}[\|\epsilon_\theta\|^2] = d$, with high probability due to the concentration effect.

Since the Stein score is defined as $s_t = -\frac{\epsilon_\theta}{\sigma_t}$, it follows that $\mathbb{E}[\|s_t\|^2] = \frac{d}{\sigma_t^2}$, where $\sigma_t$ is the variance schedule of the diffusion model. Hence,

$$\mathbb{E}\!\left[\frac{\|\epsilon\|^4}{\|s_t\|^2}\right] \approx \frac{\mathbb{E}\!\left[\|\epsilon\|^4\right]}{\|s_t\|^2} = (d+2)\sigma_t^2, \tag{58}$$

where the approximation holds with high probability by the concentration effect in high dimensions. Therefore, we obtain the compact form of the second moment:

$$\mathbb{E}[\|\Delta z_t\|^2] = (\lambda^2 + 2\lambda)\,(d+2)\,\sigma_t^2. \tag{59}$$

We aim to control the second moment such that

$$\mathbb{E}[\|\Delta z_t\|^2] \;\leq\; k \;<\; 1. \tag{60}$$

From the Eq. (59), this requirements holds if

$$\lambda^2 + 2\lambda \leq \frac{k}{(d+2)\sigma_t^2} \tag{61}$$

$$(\lambda + 1)^2 \leq 1 + \frac{k}{(d+2)\sigma_t^2} \tag{62}$$

$$\therefore \lambda \leq \sqrt{1 + \frac{k}{(d+2)\sigma_t^2}} - 1 \tag{63}$$

Since for any $v \geq 0$, it holds that $\sqrt{1+v} - 1 \leq \sqrt{v}$, a sufficient condition is

$$\lambda \leq \frac{1}{\sigma_t}\sqrt{\frac{k}{(d+2)}}. \tag{64}$$

Therefore, the sufficient condition $\lambda \leq \sqrt{\frac{k}{(d+2)}}$ guarantees that $\mathbb{E}[\|\Delta z_t\|^2] \;\leq\; k \;<\; 1$. $\qquad\square$

# B. Additional Discussion and Analysis

## B.1. Limitation

Although MCLC provides meaningful progress, hyperparameter tuning remains non-trivial, as is common in inverse problem solving. Moreover, while MCLC is broadly applicable to LDM-based inverse solvers, empirical validation in domains such as physics or medical imaging remains limited due to the lack of publicly available domain-specific LDM priors. We believe that developing practical ways to approximate the reverse-dynamics gap, designing adaptive strategies to reduce this gap, and validating MCLC in broader scientific inverse problem domains are promising directions for future work.

## B.2. Compatibility with LatentDAPS

As discussed in the main paper (Sec. 4), we provide a more detailed explanation of why LatentDAPS (Zhang et al., 2025) is less compatible with our proposed MCLC. Briefly, this could be attributed to its solving procedure which does not inherit reverse time dynamics; instead, each iteration is initialized from a newly predicted $z_0$ with annealed noise, decoupling consecutive updates. As a result, the stabilizing effect of our corrector appears limited in this case.

To clarify, we first review how LatentDAPS operates. LatentDAPS decouples consecutive steps in the reverse sampling trajectory. Rather than performing standard reverse sampling, it directly predicts $z_{0|t}$ with ODE solver and updates it using the log-posterior gradient $\nabla_{z_0} \log p(z_{0|t}|y)$, after which annealed noise is added to proceed to the next step. In this way, the dependency between successive steps is broken, allowing the method to explore a larger solution space.

Our proposed MCLC is designed to reduce the gap between the stable reverse diffusion dynamics and the dynamics of the latent diffusion inverse solver, thereby making latent diffusion solvers more stable. However, since LatentDAPS does not follow reverse dynamics by decoupling consecutive steps and repeatedly reinitializing with annealed noise, the stabilizing effect of MCLC accumulates less effectively in this setting. For this reason, Table 1 shows that MCLC is less effective with LatentDAPS compared to other solvers, where it delivers substantial performance gains. Although the powerful recent solver LatentDAPS is less compatible with MCLC, the value of our approach remains clear: it closes the gap toward the reverse process of diffusion model without relying on the linear manifold assumption, and most solvers are still built on reverse diffusion sampling combined with a measurement-consistency step.

## B.3. Clarification and Justification of Measurement-Consistent Correction Scheme

**Distinction from Predictor–corrector (PC) sampling.** PC sampling (Song et al., 2021) was originally introduced to correct discretization and score approximation errors during diffusion sampling. Subsequent PC-based samplers (Zhao et al., 2023; 2024; Lezama et al., 2023) have been studied in the context of mitigating numerical error and accelerating sampling efficiency, e.g., by reducing the number of function evaluations (NFEs). In contrast, MCLC is designed to correct a mismatch between the solver dynamics and the desired measurement-consistent distribution that arises in diffusion-based inverse problem solver's measurement-consistency step, especially LDM-based solvers. Rather than addressing sampling numerical error, MCLC targets the gap induced by the measurement-consistent step and reduces this gap in a principled manner aligned with the objective of inverse reconstruction.

Importantly, our contribution is not limited to a technical improvement. By identifying instability as a discrepancy between the solver dynamics and the stable reverse dynamics, our perspective provides a theoretically grounded tool for handling and formalizing instabilities in diffusion-based inverse problems, which can inspire future work. Moreover, to the best of our knowledge, this work is the first to introduce and theoretically justify a measurement-consistent correction scheme, elevating it from an implementation heuristic to a principled and novel methodological contribution.

**Need for measurement-consistent scheme.** Our theoretical analysis reveals that instability can arise in the reverse dynamics of LDM-based inverse solvers, from which we propose Langevin correction as a tool for mitigating such instability. However, applying it directly can lead to suboptimal solutions: the update pulls the dynamics toward high-probability regions of the prior while disturbing measurement consistency, producing plausible but data-inconsistent solutions. Since measurement consistency is the primary objective in inverse problems, such behavior is fundamentally misaligned with the goal of inverse problem solving. As shown in Table 7, MCLC overcomes this issue by maintaining the measurement consistency while stabilizing the dynamics.

## B.4. Discussion on the Design of MCLC

**Related Work.** MCLC is conceptually related to constrained Langevin dynamics, a broad class of methods that impose feasibility or geometric constraints on Langevin-type stochastic dynamics. Constrained Langevin methods (Bubeck et al., 2018; Lamperski, 2021) often enforce global constraints, such as explicit feasible sets, for constrained sampling or optimization, where the latter can be viewed through sampling from low-temperature Gibbs distributions. Motivated by this constraint-preserving view, MCLC applies a local constraint to the Langevin correction in latent diffusion inverse problems: the update is restricted to the orthogonal complement of the measurement-consistency gradient. As a result, the correction is approximately tangent to the measurement-consistency level set, moving samples toward the diffusion time marginal while preserving measurement consistency up to first order.

**Clarification on state-dependency of the MCLC projection.** MCLC computes the projection onto the orthogonal complement of the measurement-consistency gradient once at the initial state of each corrector step and keeps it fixed during the inner corrector sub-iterations. This state-independent projection (during a sub-iteration) is consistent with our theoretical derivation and is important for preserving the KL monotonicity guarantee.

One may consider recomputing the projection at every corrector sub-iteration, since this would more directly reflect the current measurement-consistency geometry as the latent state evolves. However, such a state-dependent projection does not ensure the KL decreases. In particular, it introduces an additional term on the right-hand side of Eq. (38):

$$\int \left(1 + \log\frac{q_t^c}{p_t}\right) \nabla \cdot \left(q_t^c \nabla \cdot P_{\perp g(x)}\right) dx. \tag{65}$$

Since this term has an indefinite sign, the KL monotonicity guarantee may no longer hold. In addition, recomputing the measurement-consistency gradient and the corresponding projection at every sub-iteration requires additional backpropagation, increasing the computational cost.

A remaining concern is whether fixing the projection at the initial state can be problematic when the measurement-consistency geometry changes significantly during the corrector sub-iterations, especially for highly nonlinear inverse problems. However, this does not make the measurement-consistency perturbation uncontrolled. As shown in Theorem 3.4, the perturbation bound depends on the local smoothness of the measurement operator and the corrector step size. For highly nonlinear inverse problems, the smoothness-related constant may become larger, which can widen the bound. Nevertheless, the perturbation remains governed by the step size and can therefore be controlled by choosing an appropriate corrector step size. Thus, the fixed projection used in MCLC provides a principled trade-off: it preserves the KL monotonicity guarantee while keeping the measurement-consistency perturbation controlled and the algorithm efficient.

## B.5. Patch FID and Patch Granularity

Patch FID has been adopted in several prior works (Chai et al., 2022; Poirier-Ginter & Lalonde, 2023; Wang et al., 2025) as a complementary metric to standard FID for evaluating localized artifacts and fine-grained image degradations that may not be well captured by global image statistics. In particular, Patch FID computes FID over spatial patches extracted from images, and is used to assess spatially varying distortions in inverse problems and restoration tasks.

A potential concern is that using a $3 \times 3$ patch grid may be too coarse to provide meaningful localization. To address this, we additionally evaluate Patch FID with finer patch granularities, including $5 \times 5$ patches (patch size $103 \times 103$) and $8 \times 8$ patches (patch size $64 \times 64$). Across all patch grid resolutions, MCLC consistently achieves lower Patch FID compared to the baseline methods, indicating that the observed improvements are not sensitive to the choice of patch granularity. Consequently, the main conclusions and implications of our method are preserved regardless of patch size, supporting the robustness of MCLC in improving stability.

*Table 9.* Patch-FID comparison under different patch granularities for Gaussian and Motion Deblur. The base solver is PSLD, and evaluations are conducted on the FFHQ dataset.

| Method | Gaussian Deblur | | | Motion Deblur | | |
|---|---|---|---|---|---|---|
| | 3×3 | 5×5 | 8×8 | 3×3 | 5×5 | 8×8 |
| Base | 90.54 | 85.20 | 77.17 | 102.60 | 85.20 | 82.15 |
| Ours | **59.13** | **50.37** | **58.41** | **60.05** | **50.37** | **59.88** |

## B.6. Justification of the Evaluation Protocol

As stated in the main paper, we strictly follow the evaluation protocol introduced by (Zhang et al., 2025). Specifically, we use the identical 100-image validation subset provided in the official DAPS implementation, corresponding to the first 100 images of the validation split (e.g., `FFHQ 49000--49099.png`), without any additional filtering or preferential selection. This protocol has been widely adopted in recent diffusion-based inverse problem literature (Zirvi et al., 2025; Song et al., 2024; Zhang et al., 2025; Wu et al., 2024) for extensive comparisons across datasets and tasks, enabling fair and reproducible evaluation.

To further validate that our findings generalize beyond the 100-image evaluation protocol, we additionally report results on the full 1k-image validation set for the $4\times$ super-resolution task using ReSample. As shown in Table 10, MCLC consistently improves PSNR, LPIPS, FID, and Patch FID over the baseline on the full validation set. These results demonstrate that our main empirical trends remain stable at a larger evaluation scale.

*Table 10.* Evaluation on the full 1k-FFHQ validation set for $4\times$ super-resolution using ReSample.

| Method | PSNR ↑ | LPIPS ↓ | FID ↓ | P-FID ↓ |
|---|---|---|---|---|
| ReSample | 26.26 | 0.348 | 36.62 | 93.82 |
| ReSample w/ MCLC | **27.74** | **0.270** | **29.39** | **42.79** |

## B.7. Reasonable Improvements with a Single Hyperparameter Set

In inverse problem solving, it is commonly expected that solver hyperparameters require tuning when the task or the severity of degradation changes. Our proposed MCLC also requires some tuning to achieve optimal performance. Nevertheless, Table 11 shows that MCLC achieves reasonable and consistent performance improvements even when a single default hyperparameter set is applied across all tasks. Specifically, for PSLD and LDPS, we use a unified configuration of $k = 5$, $N_c = 1$, and $\lambda = 0.1$. For Resample, we adopt the inpainting-style configuration with $k = 5$, $N_c = 3$, $\lambda = 0.15$, $N_c^{\mathrm{DPS}} = 1$, and $\lambda^{\mathrm{DPS}} = 0.05$. This unified choice is supported by our theoretical insight in Eq. (64), which indicates that an appropriate step size and a sufficient number of correction iterations allow MCLC to operate effectively without task-specific tuning. While additional tuning can further improve the trade-off between efficiency and performance under more severe degradations, these results indicate that MCLC does not rely on extensive hyperparameter tuning to obtain meaningful gains.

**Guide for MCLC Hyperparameter Choice.** The step size and the number of MCLC iterations can be selected in a simple and interpretable manner. If the degradation becomes more severe, one may increase the corrector step size or apply the correction more frequently. According to Eq. (64), this setting may introduce a slightly larger deviation from exact measurement consistency; however, under stronger degradations, prioritizing the stabilization of the reverse dynamics becomes more beneficial.

## B.8. Baseline Solver Parameter Search

We examine whether solver parameter choices could substantially improve the baseline performance beyond the configurations adopted in prior work. Focusing on the gradient step size, one of the most influential solver parameters, we consider a range of values $\{0.1\times, 0.5\times, 1\times, 2\times, 4\times\}$ centered around the setting used in the original papers (denoted as $1\times$). Figure 9 shows the PSNR and FID curves across the swept step-size parameters. While the $1\times$ setting used in the main paper is not the exact optimum, it is generally close and remains a reasonably well-tuned choice across tasks. Smaller step sizes (e.g., $0.1\times$) tend to weaken data-fidelity updates, while larger step sizes often introduce additional artifacts. Based on these observations, we treat the original solver configurations as reasonably well-tuned baselines in our experiments.

In this figure, PSNR (red, higher is better) indicates data fidelity, and FID (blue, lower is better) indicates stability without artifacts and degraded results. This figure demonstrates that some configurations come close to optimal, but the curves reveal a practical ceiling, meaning that it is difficult to find a single configuration that fully satisfies both reconstruction fidelity (PSNR) and perceptual quality (FID). Notably, applying MCLC to the default $1\times$ setting pushes both PSNR and FID beyond this apparent ceiling. This demonstrates that MCLC provides substantial gains beyond what can be achieved through solver hyperparameter tuning.

*Table 11.* **Quantitative results using a single default hyperparameter set.** Even with a unified hyperparameter configuration across tasks, MCLC provides consistent and meaningful performance improvements.

| Task | Base | Method | FFHQ | | | |
|---|---|---|---|---|---|---|
| | | | PSNR (↑) | LPIPS (↓) | FID (↓) | P-FID (↓) |
| Gaussian Deblur | LDPS | Base | 27.61 | 0.349 | 100.10 | 93.55 |
| | | Ours (single default) | **27.99** | **0.334** | **85.84** | **71.62** |
| | | Ours (tuned) | 28.14 | 0.303 | 80.83 | 54.74 |
| | PSLD | Base | 27.84 | **0.314** | 89.18 | 90.54 |
| | | Ours (single default) | **27.94** | 0.317 | **79.81** | **69.39** |
| | | Ours (tuned) | 27.97 | 0.286 | 66.28 | 59.13 |
| | ReSample | Base | 26.44 | 0.368 | **75.17** | 148.11 |
| | | Ours (single default) | **27.33** | **0.355** | 80.90 | **96.17** |
| | | Ours (tuned) | 27.25 | 0.353 | 78.38 | 106.16 |
| Motion Deblur | LDPS | Base | 26.54 | 0.390 | 118.77 | 112.74 |
| | | Ours (single default) | **27.03** | **0.363** | **99.71** | **81.82** |
| | | Ours (tuned) | 27.45 | 0.318 | 82.94 | 55.55 |
| | PSLD | Base | 26.87 | **0.343** | 106.34 | 102.60 |
| | | Ours (single default) | **26.92** | 0.348 | **90.92** | **72.30** |
| | | Ours (tuned) | 26.86 | 0.308 | 74.64 | 60.05 |
| | ReSample | Base | 22.45 | 0.635 | 108.14 | 174.52 |
| | | Ours (single default) | **24.19** | **0.599** | **103.70** | **114.85** |
| | | Ours (tuned) | 24.24 | 0.588 | 102.02 | 118.87 |
| Super Resolution (4×) | LDPS | Base | **28.47** | **0.301** | 78.08 | 69.66 |
| | | Ours (single default) | 28.19 | 0.307 | **74.33** | **57.48** |
| | | Ours (tuned) | 28.34 | 0.283 | 74.78 | 58.55 |
| | PSLD | Base | **27.69** | 0.265 | 63.95 | 63.47 |
| | | Ours (single default) | 27.44 | **0.261** | **61.09** | **52.43** |
| | | Ours (tuned) | 27.33 | 0.267 | 62.12 | 58.13 |
| | ReSample | Base | 26.40 | 0.347 | 70.16 | 133.15 |
| | | Ours (single default) | **27.73** | **0.264** | **55.38** | **68.55** |
| | | Ours (tuned) | 28.32 | 0.236 | 53.85 | 78.08 |
| Inpainting (Random) | LDPS | Base | 31.22 | 0.171 | 48.88 | 83.30 |
| | | Ours (single default) | **31.31** | **0.167** | **47.76** | **79.23** |
| | | Ours (tuned) | 31.28 | 0.169 | 48.05 | 81.68 |
| | PSLD | Base | 30.14 | 0.222 | 58.84 | **79.59** |
| | | Ours (single default) | **31.30** | **0.167** | **48.04** | 79.74 |
| | | Ours (tuned) | 30.73 | 0.185 | 49.80 | 72.69 |
| | ReSample | Base | 27.27 | 0.374 | 103.17 | 133.80 |
| | | Ours (single default) | **28.75** | **0.296** | **85.90** | **103.25** |
| | | Ours (tuned) | 29.35 | 0.235 | 75.65 | 108.27 |

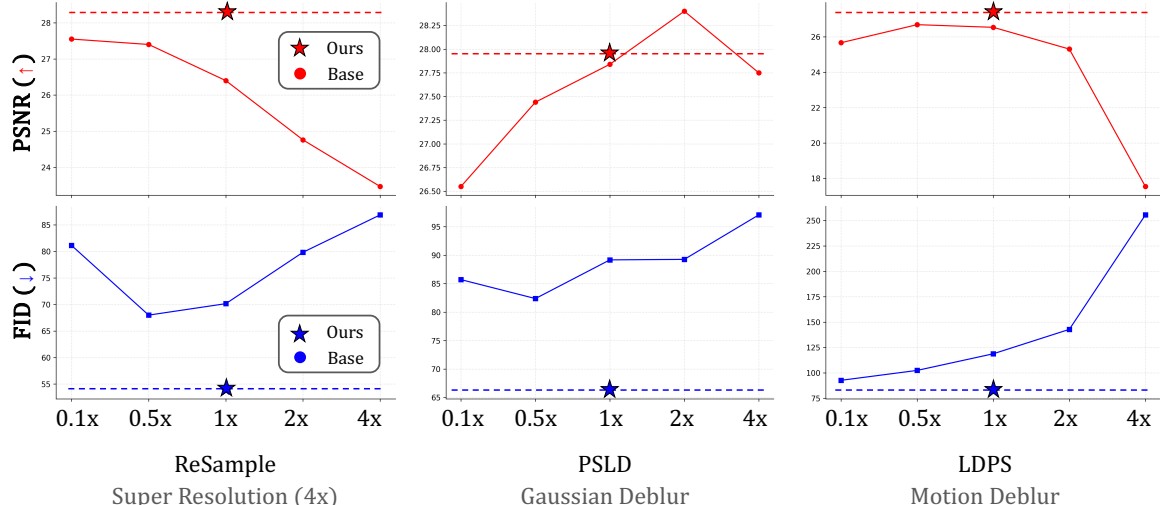

*Figure 9.* **PSNR and FID analyses across swept step-sizes.** The red and blue stars indicate the performance of MCLC applied to the default $1\times$ setting. MCLC pushes both metrics beyond the practical ceiling inferred by the sweep, demonstrating improvements that cannot be achieved through step-size tuning alone.

### B.9. Step-size Condition Verification

Our paper is developed under assumptions that are reasonable in theory. To further validate that these assumptions hold in practice, we verify that the step-size condition derived in our proof (Theorem 3.4, Eq. (60)–(64)) is empirically satisfied during real runs. Specifically, the condition guarantees $\mathbb{E}[\|\Delta z_t\|] < 1$ ensuring controlled perturbation to measurement consistency. For our experimental setting ($d = 64 \times 64 \times 4$ where resolution is 512 and compression ratio is 8), the derived condition yields $\lambda \lesssim 0.005$, and we confirm that this bound is consistently satisfied across real runs, as shown in Table 12

As this is a sufficient condition, mild violations do not necessarily lead to failure. In practice, larger step sizes ($\lambda \in [0.05, 0.15]$) rather improve stability and reconstruction performance while largely preserving measurement consistency, whereas overly large step sizes (*e.g.*, $\lambda = 0.5$) may lead to failure. These results demonstrate that the theoretical conditions are well satisfied in real runs, and that MCLC is not overly sensitive to mild violations, instead yielding improved stability and performance rather than abrupt failure.

*Table 12.* **Quantitative results on the step-size condition.** We evaluate MCLC across varying $\lambda$ under 92% random inpainting on FFHQ. LDPS is used as the base solver.

| $\lambda$ | $E[\|\Delta z_t\|]$ | y-PSNR ($\uparrow$) | PSNR ($\uparrow$) | LPIPS ($\downarrow$) |
|---|---|---|---|---|
| 0 (base) | 0 | 35.34 | 25.04 | 0.421 |
| 0.005 | 0.89 | 35.28 | 25.14 | 0.405 |
| 0.05 | 8.93 | 35.18 | **27.00** | **0.304** |
| 0.15 | 26.17 | 34.73 | 26.27 | 0.389 |
| 0.5 | 89.97 | 27.48 | 16.57 | 0.742 |

### B.10. Further Analysis of Artifacts in Latent Space

In Sec. 3 and 4, we demonstrate that MCLC effectively mitigates most types of artifacts by considering them as instances of instability. Nevertheless, certain artifacts occur even when advanced solvers are employed. The artifacts, known as blob artifacts (Raphaeli et al., 2025), are characterized by localized distortions in the reconstructed results, as shown in Fig. 10. In this section, we analyze the special case and discuss possible ways to address them.

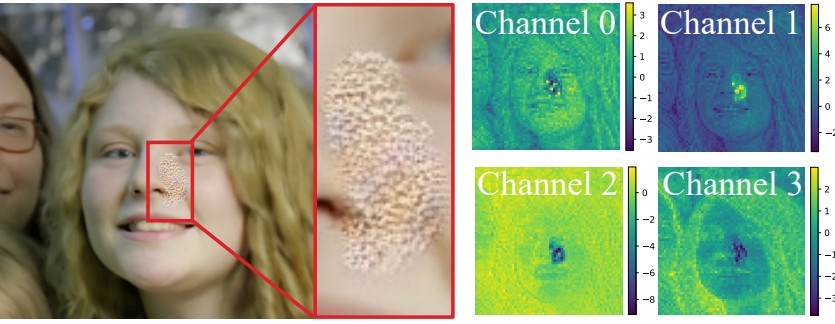

*Figure 10.* **Analysis on blob artifacts.** Blob artifacts in the decoded image arise when scaled-outliers exist in the latent.

**Where do these blob artifacts originate from?** Previous works (Song et al., 2024; Raphaeli et al., 2025) have mentioned that the artifact is caused by the decoder, which makes the gradient problematic. To clearly investigate the origin of these artifacts, we perform a more detailed analysis. As a first step, we examine whether the artifacts indeed originate from backpropagation through the decoder by analyzing the gradients $\frac{\partial L}{\partial x_{0|t}}$ and $\frac{\partial L}{\partial z_{0|t}} = J_{\mathcal{D}}^{\top} \frac{\partial L}{\partial x_{0|t}}$, where $J_{\mathcal{D}}$ denotes the Jacobian of the decoder. Then, we observe that backpropagation through the decoder makes the signal that is unrelated to the measurement gradient $\frac{\partial L}{\partial x_{0|t}}$. This observation indicates that artifacts can indeed arise from decoder backpropagation. In addition, the blob artifacts tend to occur when *scaled-outliers* are present in the latent (see Fig. 10). We define the scaled-outlier as a localized latent region whose values are substantially higher or lower than its surroundings, *i.e.*, deviations outside the typical latent range. This shows that the blob artifacts result from scaled-outliers.

**Clarification of Setups.** The latent *before* the measurement update is denoted by $z_{0|t}$ and the latent *after* applying the measurement gradient through the decoder's Jacobian $J_{\mathcal{D}}$ is denoted by $z_{0|t}(y)$. In the following, we analyze relationship between scaled-outliers and the decoder's Jacobian in detail.

**Why do scaled-outliers emerge?** Since scaled-outliers consistently appear in specific regions, we hypothesize that the decoder's Jacobian $J_{\mathcal{D}}$ selectively amplifies certain latent directions. To examine this, we analyze the principal eigenvector of $J_{\mathcal{D}} J_{\mathcal{D}}^{\top}$. Figure 11 shows that scaled-outlier regions in the updated latent $z_{0|t}(y)$ are strongly correlated with the regions amplified by the decoder's Jacobian $J_{\mathcal{D}}$. This reveals that scaled-outliers arise from Jacobian amplification.

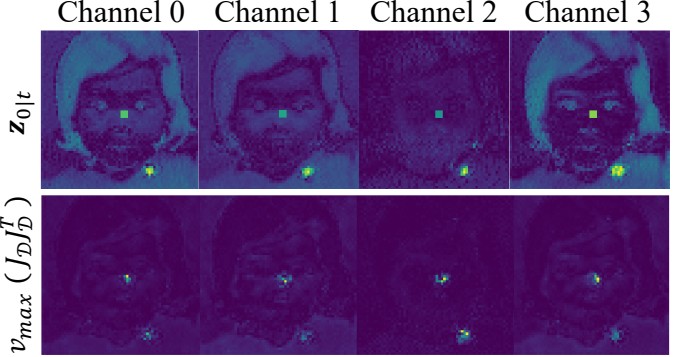

$$v_{max}\left(J_{\mathcal{D}} J_{\mathcal{D}}^{T}\right) \qquad z_{0|t}(y) \qquad x_0(y)$$

*Figure 11.* **Analysis on scaled-outliers.** Scaled-outlier regions in $z_{0|t}(y)$ are aligned with regions amplified by the decoder's Jacobian $J_{\mathcal{D}}$.

**Why do such artifacts remain?** In principle, such artifacts should be heavily penalized by the loss function and thus eliminated, yet they persist. We find that when the latent $z_{0|t}$ already contains scaled-outlier regions before the update, the decoder's Jacobian amplifies the gradient in the surrounding area. To verify this effect, we artificially inject a $3 \times 3$ scaled-outlier latent patch into the center of the input latent $z_{0|t}$. As shown in Fig. 12, the decoder's Jacobian $J_{\mathcal{D}}$ exhibits strong amplification around the injected center region. In summary, when the input latent $z_{0|t}$ contains scaled-outliers, the decoder's Jacobian amplifies these, which then reappear in the updated latent $z_{0|t}(y)$. During the reverse sampling process, scaled-outliers are further magnified, which prevents their elimination and eventually manifests as blob artifacts.

| Channel 0 | Channel 1 | Channel 2 | Channel 3 |
|---|---|---|---|

*Figure 12.* **Relation between scaled-outliers and $J_{\mathcal{D}}$.** By artificially injecting a scaled-outlier patch into $z_{0|t}$, we confirm that $J_{\mathcal{D}}$ amplifies such regions when the outliers are present in the input.

**How can we handle it?** According to the analyses above, if the input latent contains no scaled-outliers, amplification does not occur. Interestingly, we find that the latent space of SD v1.5, which is widely used in latent diffusion inverse solvers, inherently contains scaled-outlier regions as confirmed by the encoding of original images (see Fig. 14). Although low-magnitude outliers do not manifest as visible artifacts when decoded, they can be amplified during the reverse sampling process of latent diffusion inverse solvers and eventually appear as visible artifacts. As shown in the last two rows of Fig. 4 and Sec. C.2, our proposed method suppresses this amplification, thereby removing the blob artifacts. However, since these outliers are inherent to the target latent distribution itself, they cannot be fully eliminated. A straightforward alternative is to adopt latent spaces that are free from such outliers, for example, those of SDXL (see Fig. 14). Another possible approach is to incorporate pixel-level optimization to avoid blob artifacts, as demonstrated in P2L (Chung et al., 2024).

**Problematic gradient.** As noted above, we confirm that the decoder itself produces problematic gradients. To investigate this, we decompose the gradient $\frac{\partial L}{\partial z_{0|t}}$ into two components $J_{\mathcal{D}}^{\top} \frac{\partial L}{\partial x_{0|t}}$ and compare their characteristics to examine the effect of the decoder Jacobian. Specifically, we analyze the gradients $\frac{\partial L}{\partial x_{0|t}}$ in pixel space and $\frac{\partial L}{\partial z_{0|t}}$ in latent space. As shown in Fig. 13, the decoder Jacobian introduces signals in directions redundant to the measurement-consistent pixel-level gradients, thereby distorting the latent gradients.

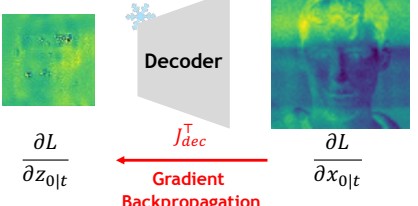

*Figure 13.* Redundant gradient signals

**Scaled outliers in latent space.** We analyze the scaled-outlier in the latent space of VAE. We present a visualization of the encoded latents from multiple RGB images in Fig. 14. The result indicates that the encoded latents from Stable Diffusion v1.5 already contain scaled-outlier regions across channels and images, whereas those from Stable Diffusion XL do not. This finding supports that blob artifacts arise not only from the VAE decoder, but also from the pre-trained VAE itself. Therefore, the blob artifacts could be mitigated by replacing the base diffusion model with an enhanced diffusion model that can reduce scaled-outliers.

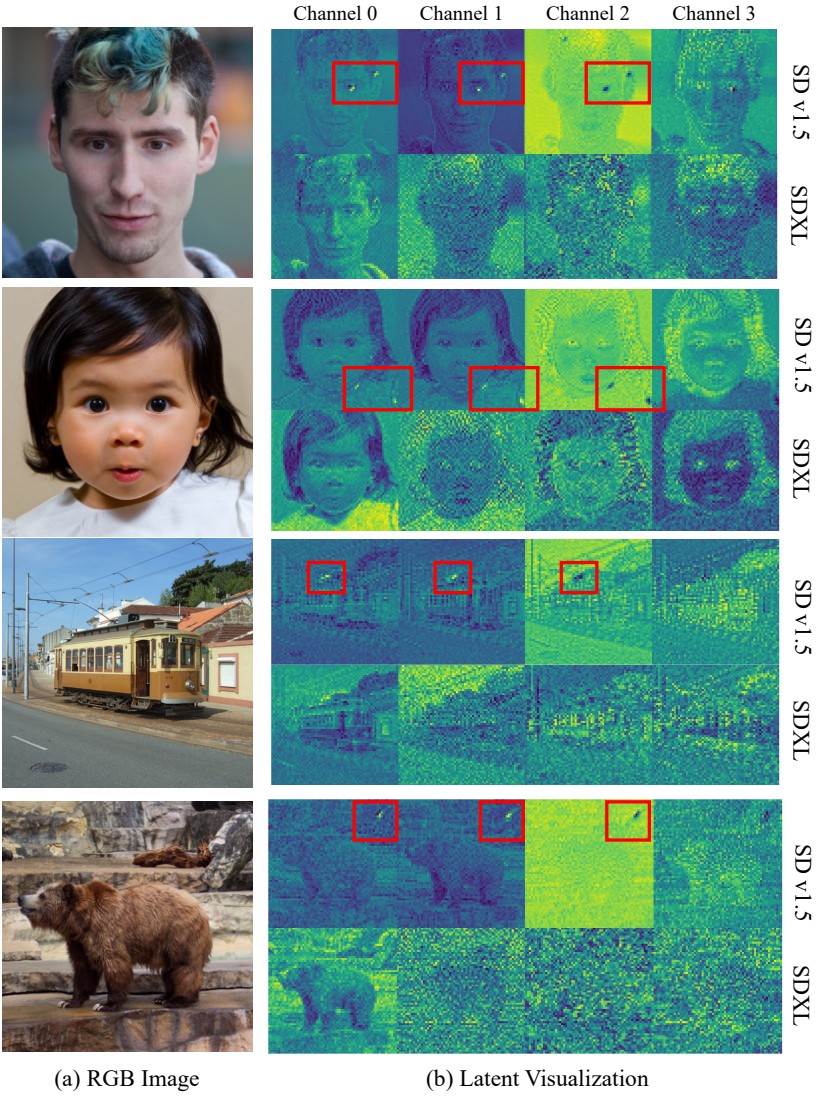

(a) RGB Image          (b) Latent Visualization

*Figure 14.* Visualization of latent spaces in SD v1.5 and SDXL. SD v1.5, the commonly used latent diffusion model for inverse problems, exhibits scaled outliers in its latent space. The scaled-outliers are amplified through the decoder, which may result in undesirable artifacts. Unlike SD v1.5, the latent space of SDXL shows no such outliers, displaying only a slightly noisy appearance.

**Blob artifcats mitigation.** As noted earlier, blob artifacts arise when latent values become excessively amplified, that is, when they are pushed far outside the feasible latent range. While MCLC suppresses this artifact by pulling the latent values back toward stable regions, this corrective influence can be weaker than the amplification, which explains why certain artifacts may not be fully removed. Nevertheless, MCLC sufficiently suppresses the blob artifacts. To demonstrate how effectively MCLC mitigates this phenomenon, we provide additional qualitative results in Fig. 15. These examples show that MCLC significantly reduces the magnitude of out-of-range latent values and noticeably suppresses the resulting blob artifacts in the decoded images.

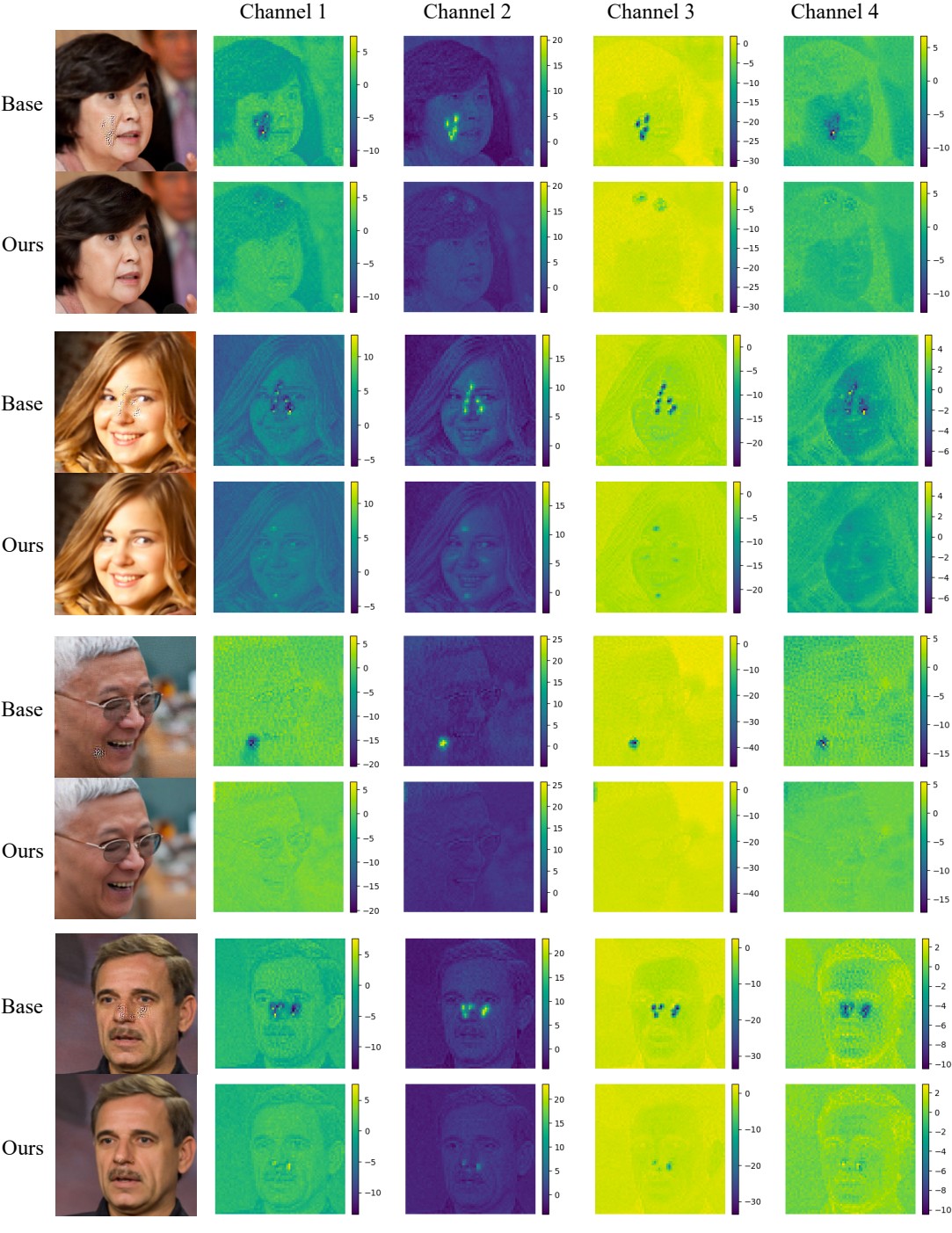

*Figure 15.* **Visualization of MCLC's suppression of blob artifacts.** The base solver is LDPS and the task is Gaussian deblurring.

# C. Additional Results

## C.1. Qualitative Results: Drop-in Improvement

We provide additional qualitative comparisons against reconstructions obtained with the naive latent diffusion solver (base), as well as the same base solver plugged in with DiffStateGrad or MCLC (ours) for each task. As shown in Fig. 19, 20, 21, 22, 23, and 24, MCLC substantially improves the performance of existing latent diffusion solvers and effectively stabilizes solutions.

## C.2. Qualitative Results: Comparison with Non-Pluggable Approaches

As noted in the main paper (Sec. 4), we further validate the effectiveness of our method by comparing it with non-pluggable artifact-removal approaches, MPGD (He et al., 2024) and SILO (Raphaeli et al., 2025). As shown in Fig. 16, MCLC achieves notable improvements in both measurement consistency and perceptual quality, even when compared to non-pluggable approaches designed for stabilization.

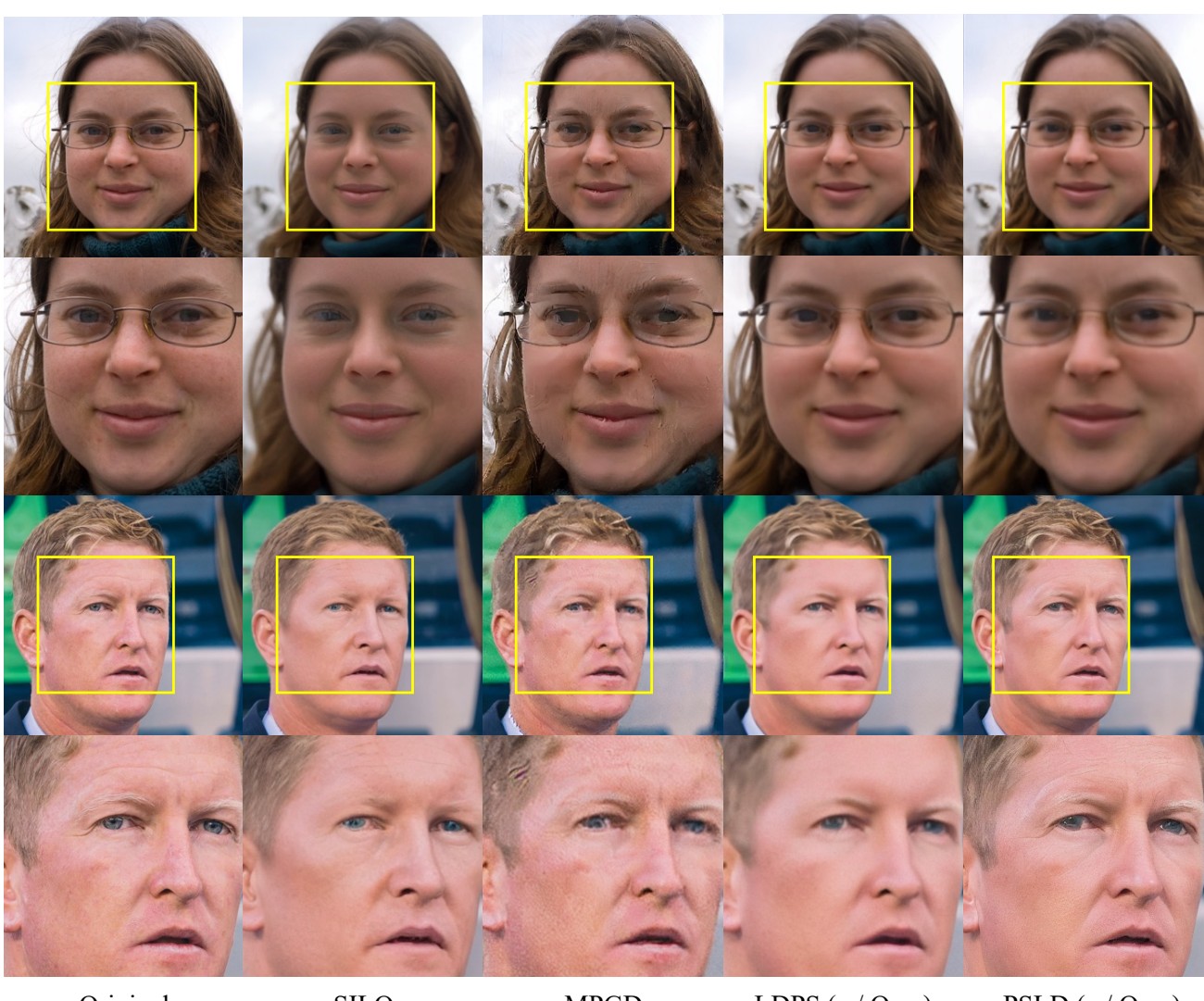

| Original | SILO | MPGD | LDPS (w/ Ours) | PSLD (w/ Ours) |

*Figure 16.* **Qualitative comparison with non-pluggable approaches.** MCLC mitigates artifacts and improves quality in a plug-and-play manner, surpassing non-pluggable approaches such as MPGD (He et al., 2024) and SILO (Raphaeli et al., 2025). SILO fails to reconstruct the glasses in the first row and often alters identity. MPGD exhibits noticeable overall artifacts.

MPGD provides slightly stronger measurement consistency; however, this improvement does not consistently translate into stability or overall quality, and noticeable artifacts remain. SILO achieves high perceptual quality by effectively removing artifacts; however, encoding measurements into the latent space introduces information loss, which leads to low measurement consistency. This limitation may undermine the fundamental goal of inverse problems, reconstructing the original signal consistent with observations. Moreover, since SILO trains its latent degradation operator in a domain-specific manner, it is difficult to generalize in a domain-agnostic setting, which restricts its applicability.

In contrast, MCLC not only adapts readily across domains in a plug-and-play manner without specialized designs but also ensures measurement consistency. This enables existing LDM-based inverse solvers to realize their potential by enhancing stability and quality without sacrificing fidelity or generalizability. Notably, MCLC yields significant gains even when combined with basic solvers such as LDPS and PSLD, highlighting its effectiveness. Considering its easily pluggable nature, MCLC can be combined with various baselines, leaving further room for performance improvement.

### C.3. Quantitative and Qualitative Results: Pixel Diffusion Model (PDM)

Since MCLC can be adaptable not only to Latent Diffusion Models(LDMs) but also to Pixel Diffusion Models(PDMs), we report the experimental results on DPS with and without MCLC. As shown in Table 13 and Fig. 17, our method achieves performance gain, with particularly notable improvements on the motion deblurring task. In this work, we focus on Latent Diffusion Models (LDMs) that provide generic priors. Unlike domain-specialized priors, which tend to produce fewer artifacts, LDMs such as Stable Diffusion suffer more severely from the gap to true reverse diffusion dynamics, often resulting in artifacts and degraded quality. This explains why the performance gain appears more substantial in the generic LDMs.

Additionally, MCLC performs well when combined with a recent diffusion inverse solver SITCOM (Alkhouri et al., 2025), which enforces step-wise triple backward consistency for stabilization, as shown in Table 13.

|  | Motion Deblur | | | Gaussian Deblur | | | Nonlinear Deblur | | |
|---|---|---|---|---|---|---|---|---|---|
|  | PSNR (↑) | LPIPS (↓) | FID (↓) | PSNR (↑) | LPIPS (↓) | FID (↓) | PSNR (↑) | LPIPS (↓) | FID (↓) |
| DPS | 24.31 | 0.282 | 81.14 | 25.06 | 0.257 | 75.73 | 22.97 | 0.369 | 108.97 |
| DPS w/ Ours | **26.11** | **0.248** | **70.60** | **25.08** | **0.256** | **74.77** | **23.11** | **0.362** | **104.83** |
| SITCOM | 27.51 | 0.178 | 91.67 | 26.79 | 0.242 | 109.13 | 26.96 | 0.107 | 56.21 |
| SITCOM w/ Ours | **28.94** | **0.147** | **82.36** | **26.79** | **0.242** | **103.88** | **26.99** | **0.106** | **55.28** |

*Table 13.* Comparison on DPS (Chung et al., 2023) and SITCOM (Alkhouri et al., 2025) with and without MCLC on FFHQ.

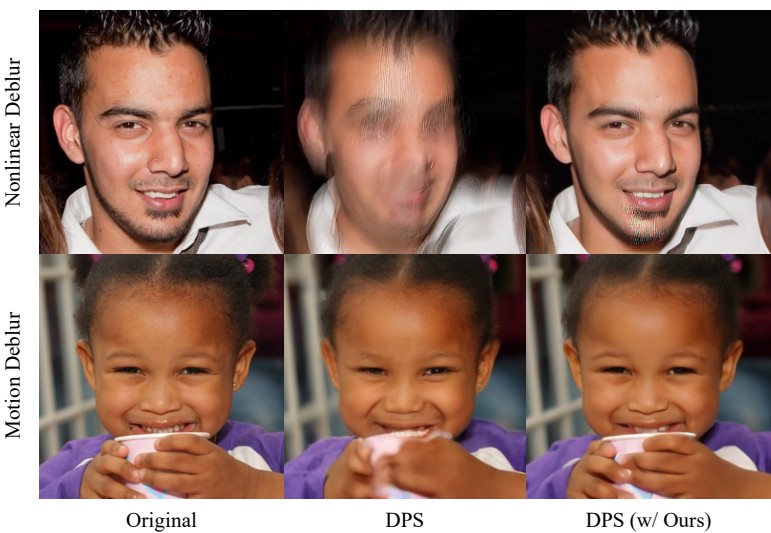

Original          DPS          DPS (w/ Ours)

*Figure 17.* Qualitative result on pixel diffusion inverse solver (DPS (Chung et al., 2023)), with and without our method MCLC.

## C.4. Application beyond inverse problem

**Prompt:** walker hound, Walker foxhound on snow

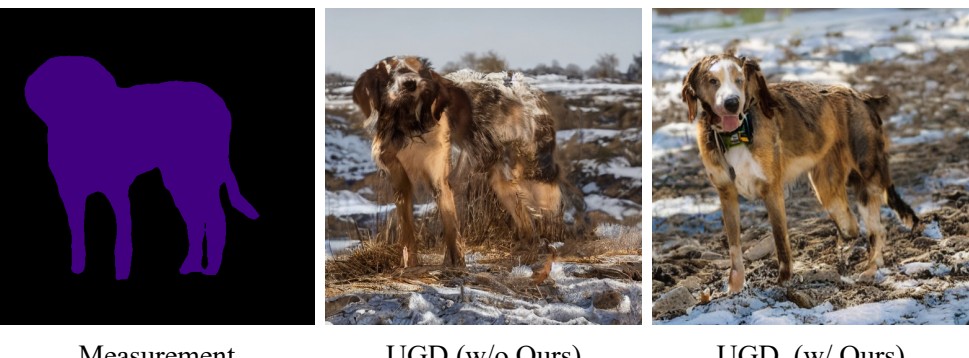

Measurement        UGD (w/o Ours)        UGD (w/ Ours)

*Figure 18.* Text-conditioned image generation results guided by segmentation masks using UGD (Bansal et al., 2024).

To further investigate the applicability of MCLC, we provide guided-sampling results with and without MCLC. Universal Guided Diffusion (UGD) (Bansal et al., 2024) proposes a universal guided diffusion sampling framework applicable to various guidance signals, where the guidance can be any off-the-shelf model or guiding function. We present text-conditioned image generation results guided by a segmentation mask. Figure 18 shows the extensibility of our proposed correcting mechanism to guided sampling. In this experiment, we set the inner iteration of UGD to 3.

# D. Implementation Details

In this section, we provide details of our experimental implementation. In Sec. D.1, we describe how we measure the KL divergence with respect to the true reverse diffusion dynamics, corresponding to the experiments shown in Fig. 2. In Sec. D.3, we present an overview of our proposed method, explaining how it can be plugged into existing solvers along with the detailed algorithm. In the following sections (Sec. D.4, D.5, D.6, and D.7), we sequentially report the hyperparameter settings of the baseline latent diffusion solvers, the integration of our method into these solvers, and the configuration of our correctors, including their hyperparameter choices.

## D.1. Experimental Details: Measuring KL Divergence

Figure 2 reports the KL divergence at each timestep (sampled every 15 steps) between the true reverse diffusion dynamics and those produced by the LDPS solver on the Gaussian deblurring task with the FFHQ dataset. Results are shown for both the baseline solver and the solver corrected with our MCLC. To compute each KL divergence, we first collect the intermediate states of each dynamics across the dataset: $z_t^{\#} \sim q_t^{\#}$, denoting the time-evolving distribution of the latent diffusion inverse solver, and $z_t^c \sim q_t^c$, denoting that of the corrected solver. To approximate the true reverse diffusion dynamics, we employ DDIM inversion to obtain the corresponding intermediate states $z_t \sim p_t$ at each timestep. For each intermediate latent tensor, we treat every spatial 4-dimensional latent code ($z_{t,l} \in \mathbb{R}^4$ as one sample, where $l$ indexes the $64 \times 64$ latent grid corresponding to a $512 \times 512$ resolution image. We then fit Gaussian Mixture Models (GMMs) with 32 Gaussian components to these sets of samples and compute the KL divergence between the resulting GMM distributions, i.e., $\mathcal{D}_{KL}(q_t^{\#}\|p_t)$ (red line) and $\mathcal{D}_{KL}(q_t^c\|p_t)$ (purple line). Since the KL divergence between two GMMs is not generally available in closed form, we approximate $\mathcal{D}_{KL}$ using Monte Carlo sampling (Hershey & Olsen, 2007). Specifically, to compute $\mathcal{D}_{KL}(q\|p)$, we sample $z_{t,l}^q \sim q_{\text{GMM}}$, evaluate the closed-form GMM log-densities $\log q_{\text{GMM}}(z_{t,l}^q)$ and $\log p_{\text{GMM}}(z_{t,l}^q)$, and approximate $\mathbb{E}[\log q(z_{t,l}^q) - \log p(z_{t,l}^q)]$ by the empirical sample average.

**Clarification on the role of GMM-based KL estimation and potential bias.** There is a potential bias in GMM-based KL estimation. We emphasize that this estimator is used solely for analysis and visualization in Fig. 2, and is not involved in the algorithm or theoretical derivation of MCLC. The KL-decrease property of MCLC is established independently by Proposition 3.2 and Theorem 3.4. Thus, the estimated KL values should be interpreted as approximate diagnostics rather than exact high-dimensional divergences. In other words, even in the presence of potential estimator bias, the relative reduction observed under the same estimator provides a useful diagnostic for examining how MCLC reduces the discrepancy to the time-marginal distribution in practice.

---

**Algorithm 1** GMM-based KL Approximation

---

**Require:** Latent sample sets $\mathcal{Z}_t^p = \{z_{t,i,l}^p\}_{i=1,\ldots,N \,;\, l\in[64]\times[64]}$ and $\mathcal{Z}_t^q = \{z_{t,i,l}^q\}_{i=1,\ldots,N \,;\, l\in[64]\times[64]}$ at timestep $t$; number of GMM components $K$; number of Monte Carlo samples $M$
1: Fit $p_{t,\text{GMM}} \leftarrow \text{GMM}(\mathcal{Z}_t^p; K)$
2: Fit $q_{t,\text{GMM}} \leftarrow \text{GMM}(\mathcal{Z}_t^q; K)$
3: Draw samples $\{\tilde{z}_t^{(s)}\}_{s=1}^M \sim q_{t,\text{GMM}}$
4: **for** $s = 1,\ldots,M$ **do**
5: $\quad \log q_{t,\text{GMM}}^{(s)} \leftarrow \log q_{t,\text{GMM}}(\tilde{z}_t^{(s)})$
6: $\quad \log p_{t,\text{GMM}}^{(s)} \leftarrow \log p_{t,\text{GMM}}(\tilde{z}_t^{(s)})$
7: **end for**
8: **return** $\widehat{\mathcal{D}}_{\text{KL}}(q_t\|p_t) \leftarrow \frac{1}{M} \sum_{s=1}^M \left( \log q_{t,\text{GMM}}^{(s)} - \log p_{t,\text{GMM}}^{(s)} \right)$

---

## D.2. Experimental Details: Applicability to Recent Inverse Solvers

**TReg experiments (Table 5).** Table 5 shows the compatibility of our MCLC with the recent advanced latent diffusion inverse solver, TReg (Kim et al., 2025b). For TReg, we strictly follow the experimental settings described in the paper (*e.g.*, task setup). Since several configuration details are not explicitly noted in the paper, we use the default AFHQ settings provided in the authors' official code (https://github.com/TReg-inverse/TReg) for the unspecified ones. Our reproduced performance is slightly worse than the reported numbers but remains close, and the relative trend is consistent. Within this reproduced setup, adding our MCLC module provides consistent improvements over the TReg baseline. In this

experiment, we evaluate on the AFHQ-val 1K dataset (Choi et al., 2020). For all tasks, the measurement noise level is set to $\sigma_y = 0.01$. Super-resolution is performed with a 12× bicubic downsampling operator, and Gaussian deblurring uses a kernel size of 61 with intensity 5.0. We use the same MCLC hyperparameters (every 3 sampling steps, 3 correction iterations, and $\lambda = 0.2$) across all tasks.

**FlowChef experiments (Table 6).** Table 6 show further applicability of our MCLC with the recent flow-based generative model. In this experiment, we use Stable Diffusion v3 as the prior model and FlowChef (Patel et al., 2025) as the base solver. Across diverse tasks, MCLC consistently provides noticeable improvements. Although we demonstrate the applicability of MCLC to a flow-based model, we note that flow-based generative models differ from diffusion models in that they parameterize a velocity field rather than a score function. Accordingly, we estimate the score following the approach in (Kim et al., 2025a) to enable our stabilization scheme. A more tailored variant of MCLC for flow-based methods would be an interesting direction for future improvement. We use the FlowChef implementation from a subsequent work(`https://github.com/FlowDPS-Inverse/FlowDPS`) along with the recommended configurations. However, we find that the step size of 200 used for super-resolution in the paper (Kim et al., 2025a) is unsuitable, so we tune it to 20 for all super-resolution tasks.

### D.3. Algorithms

In this section, we present the algorithm of our proposed MCLC and provide an overview of how it can be plugged into existing latent diffusion inverse solvers. More generally, latent diffusion inverse solvers follow a framework where the reverse sampling process is interleaved with a measurement-consistency step. Our MCLC step is inserted immediately after this measurement-consistency step. An overview of the pluggable algorithm is given in Algorithm 2, and the detailed procedure of MCLC is described in Algorithm 3. For efficiency, MCLC is executed every $k$ steps (e.g., $k = 3$) instead of being applied at each step.

---

**Algorithm 2** Latent diffusion inverse solver with MCLC correction

---

**Require:** Pretrained LDM $s_\theta$, VAE decoder $\mathcal{D}_\phi$, measurement $\boldsymbol{y}$ variance schedule $\{\beta(t)\}_{t=1}^T$, corrector step size hyperparameter $\lambda$

    **Init:** $\boldsymbol{z}_T \sim \mathcal{N}(\boldsymbol{0}, \mathbf{I})$

    **for** $t = T, \dots, 1$ **do**

        $\hat{\boldsymbol{z}}_{0|t} \leftarrow \text{ApproxPosterior}\left(\boldsymbol{z}_t, \boldsymbol{s}_\theta(\boldsymbol{z}_t, t)\right)$                                           $\triangleright$ e.g., Tweedie's formula

        **// Measurement Consistency Step**

        $\hat{\boldsymbol{x}}_{0|t} \leftarrow \mathcal{D}_\phi(\hat{\boldsymbol{z}}_{0|t})$                                                    $\triangleright$ Decode latent

        $(\boldsymbol{z}_{t|\boldsymbol{y}}^{\#}, \boldsymbol{g}_t) \leftarrow \text{LatentUpdate}\left(\hat{\boldsymbol{x}}_{0|t}, \boldsymbol{y}, \mathcal{A}\right)$                   $\triangleright$ Return measurement-consistent gradient $\boldsymbol{g}_t$

        $\boldsymbol{z}_{t|\boldsymbol{y}}^0 \leftarrow \boldsymbol{z}_{t|\boldsymbol{y}}^{\#}$

        **// Correction Step**

        **for** $c = 1, \dots, N_c$ **do**                      $\triangleright$ Langevin update within the orthogonal complement of $\boldsymbol{g}_t$

            $\mathbf{z}_{t|y}^c \leftarrow \texttt{MCLC}(\mathbf{z}_{t|y}^{c-1}, \mathbf{s}_\theta, t, \boldsymbol{g}_t)$

        $\boldsymbol{z}_{t-1} \leftarrow \text{ReverseSampling}(\boldsymbol{z}_{t|\boldsymbol{y}}^{N_c}, \boldsymbol{s}_\theta, \beta(t))$

    **end for**

    **return** $\boldsymbol{x}_0 = \mathcal{D}_\phi(\boldsymbol{z}_0)$                                                         $\triangleright$ final reconstruction

---

---

**Algorithm 3** Measurement-Consistent Langevin Corrector (MCLC)

---

**Parameters:** Corrector step size $\{\eta_t\}_{t=1}^T$

**Inputs:** Pre-corrected latent $z_t^\#$, score network $s_\theta$, time $t$, measurement-consistent gradient $g_t$

**Outputs:** Corrected latent $z_t^c$

**Function** MCLC($z_t^\#, s_\theta, t, g_t$) :

    $g \leftarrow g_t/\|g_t\|$                                          ▷ normalize measurement gradient $g_t$

    $z_t^c \leftarrow z_t^\# + \eta_t \cdot \Pi_{\perp g}\big(s_\theta(z_t^\#, t)\big) + \sqrt{2\eta_t} \cdot \Pi_{\perp g}(\epsilon)$            ▷ $\epsilon \sim \mathcal{N}(\mathbf{0}, \mathbf{I})$

    **return** $z_t^c$

---

## D.4. Algorithmic Details: LDPS (Latent DPS)

We plug MCLC into Latent DPS (LDPS) and present the resulting algorithm in Algorithm 4. We used the LDPS is an extension of DPS (Chung et al., 2023) to latent diffusion models. LDPS applies a measurement-consistency step at every sampling iteration. For LDPS, we used the original PSLD implementation, with the only modifications being the removal of the PSLD regularization term and the addition of MCLC. For our experiments, we use the DDIM sampling procedure with 1000 timesteps, and apply the MCLC step at every $k$-th step during sampling. At each corrector step, we perform $N_c$ corrector iterations with step size hyperparameter $\lambda$, as defined in Eq. (56). The detailed experimental settings, corresponding to those reported in Table 1, are summarized in Table 14 across each task.

---

**Algorithm 4** MCLC-LDPS

---

**Require:** $T, y, \zeta, \{\alpha_t\}_{t=1}^T, \{\bar{\alpha}_t\}_{t=1}^T, \{\tilde{\sigma}_t\}_{t=1}^T$

**Require:** $\mathcal{E}, \mathcal{D}, \mathcal{A}, s_\theta, N_c, \lambda,$

  $z_T \sim \mathcal{N}(\mathbf{0}, \mathbf{I})$

  **for** $t = T$ **to** 1 **do**

    $\hat{s} \leftarrow s_\theta(z_t, t)$

    $\hat{z}_0 \leftarrow \frac{1}{\sqrt{\bar{\alpha}_t}}(z_t + (1 - \bar{\alpha}_t)\hat{s})$

    $\epsilon \sim \mathcal{N}(\mathbf{0}, \mathbf{I})$

    $z_{t-1} \leftarrow \frac{\sqrt{\alpha_t}(1-\bar{\alpha}_{t-1})}{1-\bar{\alpha}_t}z_t + \frac{\sqrt{\bar{\alpha}_{t-1}}(1-\alpha_t)}{1-\bar{\alpha}_t}\hat{z}_0 + \tilde{\sigma}_t\epsilon$

    $g_t \leftarrow \zeta\nabla_{z_t}\|y - \mathcal{A}(\mathcal{D}(\hat{z}_0))\|_2^2$

    $z'_{t-1} \leftarrow z_{t-1} - g_t$

    **// MCLC Correction Step**

    **if** $(t \mod k) = 0$ **then**

        **for** $c = 1, \ldots, N_c$ **do**

            $z'_{t-1} \leftarrow$ MCLC($z'_{t-1}, s_\theta, t - 1, g_t$)            ▷ Corrector step size hyperparameter $\lambda$

  **end for**

  **return** $\mathcal{D}(\hat{z}_0)$

---

|  | Inpainting (random) | Super Resolution | Gaussian Deblur | Motion Deblur |
|---|---|---|---|---|
| $k$ | 15 | 15 | 10 | 10 |
| $N_c$ | 3 | 3 | 3 | 3 |
| $\lambda$ | 0.07 | 0.15 | 0.27 | 0.27 |

*Table 14.* Experiment configurations for Latent DPS (LDPS).

## D.5. Algorithmic Details: PSLD

We plug MCLC into PSLD (Rout et al., 2023) and present the resulting algorithm in Algorithm 5. PSLD introduces an additional regularization term (gluing) and applies a measurement-consistency step with the regularized loss at every sampling iteration. We used the original PSLD implementation without any modification, except for applying MCLC. For our experiments, we use the DDIM sampling procedure with 1000 timesteps, and apply the MCLC step at every $k$-th step during sampling. At each corrector step, we perform $N_c$ corrector iterations with step size hyperparameter $\lambda$, as defined in Eq. (56). The detailed experimental settings, corresponding to those reported in Table 1, are summarized in Table 15 across each task.

---

**Algorithm 5** MCLC-PSLD

---

**Require:** $T, \boldsymbol{y}, \gamma, \zeta, \{\alpha_t\}_{t=1}^T, \{\bar{\alpha}_t\}_{t=1}^T, \{\tilde{\sigma}_t\}_{t=1}^T$

**Require:** $\mathcal{E}, \mathcal{D}, \mathcal{A}, \mathcal{A}\boldsymbol{x}_0^*, \boldsymbol{s}_\theta, N_c, \lambda,$

   $\boldsymbol{z}_T \sim \mathcal{N}(\boldsymbol{0}, \boldsymbol{I})$

   **for** $t = T$ **to** 1 **do**

      $\hat{\boldsymbol{s}} \leftarrow \boldsymbol{s}_\theta(\boldsymbol{z}_t, t)$

      $\hat{\boldsymbol{z}}_0 \leftarrow \frac{1}{\sqrt{\bar{\alpha}_t}}(\boldsymbol{z}_t + (1 - \bar{\alpha}_t)\hat{\boldsymbol{s}})$

      $\boldsymbol{\epsilon} \sim \mathcal{N}(\boldsymbol{0}, \boldsymbol{I})$

      $\boldsymbol{z}_{t-1} \leftarrow \frac{\sqrt{\alpha_t}(1-\bar{\alpha}_{t-1})}{1-\bar{\alpha}_t}\boldsymbol{z}_t + \frac{\sqrt{\bar{\alpha}_{t-1}}(1-\alpha_t)}{1-\bar{\alpha}_t}\hat{\boldsymbol{z}}_0 + \tilde{\sigma}_t\boldsymbol{\epsilon}$

      $\boldsymbol{g}_t \leftarrow \zeta\nabla_{\boldsymbol{z}_t}\|\boldsymbol{y} - \mathcal{A}(\mathcal{D}(\hat{\boldsymbol{z}}_0))\|_2^2 + \gamma\nabla_{\boldsymbol{z}_t}\|\hat{\boldsymbol{z}}_0 - \mathcal{E}(\mathcal{A}^T\mathcal{A}\boldsymbol{x}_0^* + (\boldsymbol{I} - \mathcal{A}^T\mathcal{A})\mathcal{D}(\hat{\boldsymbol{z}}_0))\|_2^2$

      $\boldsymbol{z}_{t-1}' \leftarrow \boldsymbol{z}_{t-1} - \boldsymbol{g}_t$

      // **MCLC Correction Step**

      **if** $(t \mod k) = 0$ **then**

         **for** $c = 1, \ldots, N_c$ **do**

            $\boldsymbol{z}_{t-1}' \leftarrow \texttt{MCLC}(\boldsymbol{z}_{t-1}', \mathbf{s}_\theta, t-1, \boldsymbol{g}_t)$          ▷ Corrector step size hyperparameter $\lambda$

   **end for**

   **return** $\mathcal{D}(\hat{z}_0)$

---

| | Inpainting (random) | Super Resolution | Gaussian Deblur | Motion Deblur |
|---|---|---|---|---|
| $k$ | 15 | 15 | 10 | 10 |
| $N_c$ | 3 | 3 | 3 | 3 |
| $\lambda$ | 0.07 | 0.15 | 0.27 | 0.27 |

*Table 15.* Experiment configurations for PSLD.

## D.6. Algorithmic Details: ReSample

We plug MCLC into ReSample and present the resulting algorithm in Algorithm 6. ReSample applies the measurement-consistency step every 10 iterations, with a staged strategy: it skips the first one-third of the reverse process, performs pixel optimization in the middle one-third, and applies latent optimization with hard consistency in the final one-third. In addition, a standard DPS step is included throughout the reverse sampling process. For Resample, we used Diff-State implementation, which extends the original Resample code, with removal of the Diff-State module. To use a better-tuned setting, we adopted their implementation configuration. For our experiments, we adopt DDIM sampling with 50 steps. Following the setting in DiffStateGrad (Zirvi et al., 2025), we insert MCLC into the only latent optimization stage and DPS step. In the pixel optimization stage, we do not perform correction. Specifically, the pixel optimization stage performs 2000 ($N_{\text{pixel}}$) updates, while the latent optimization stage performs 500 ($N_{\text{latent}}$) updates. Within the latent optimization stage, MCLC is applied every $k$ iterations, performing $N_c$ correction steps with step size $\lambda$ as defined in Eq. (56). Furthermore, we also apply MCLC

to the DPS steps, where it is used at every iteration of the 50-step process. In this case, the number of corrector steps and step size are denoted by $N_c^{\text{DPS}}$ and $\lambda^{\text{DPS}}$, respectively. The detailed experimental settings, corresponding to those reported in Table 1, are summarized in Table 16 across each task.

---

**Algorithm 6** MCLC-Resample

---

**Require:** $T, \boldsymbol{y}, \zeta, \{\alpha_t\}_{t=1}^T, \{\bar{\alpha}_t\}_{t=1}^T, \{\tilde{\sigma}_t\}_{t=1}^T$

**Require:** $\mathcal{E}, \mathcal{D}, \mathcal{A}, \boldsymbol{s}_\theta, \gamma, C_{\text{pixel}}, C_{\text{latent}}, N_c, \lambda, N_c^{\text{DPS}}, \lambda^{\text{DPS}}$

  $\boldsymbol{z}_T \sim \mathcal{N}(\boldsymbol{0}, \boldsymbol{I})$                     ▷ Initial noise vector

  **for** $t = T, \ldots, 1$ **do**

    $\hat{\boldsymbol{s}} \leftarrow \boldsymbol{s}_\theta(\boldsymbol{z}_t, t)$

    $\hat{\boldsymbol{z}}_0 \leftarrow \frac{1}{\sqrt{\bar{\alpha}_t}}(\boldsymbol{z}_t + (1 - \bar{\alpha}_t)\hat{\boldsymbol{s}})$

    $\boldsymbol{z}_{t-1} \leftarrow \frac{\sqrt{\alpha_t}(1-\bar{\alpha}_{t-1})}{1-\bar{\alpha}_t}\boldsymbol{z}_t + \frac{\sqrt{\bar{\alpha}_{t-1}}(1-\alpha_t)}{1-\bar{\alpha}_t}\hat{\boldsymbol{z}}_0 + \tilde{\sigma}_t\boldsymbol{\epsilon}$

    $\boldsymbol{g}_t \leftarrow \zeta \nabla_{\boldsymbol{z}_t}\|\boldsymbol{y} - \mathcal{A}(\mathcal{D}(\hat{\boldsymbol{z}}_0))\|_2^2$

    $\boldsymbol{z}'_{t-1} \leftarrow \boldsymbol{z}_{t-1} - \boldsymbol{g}_t$

    **// MCLC Correction Step**

    **for** $c = 1, \ldots, N_c^{\text{DPS}}$ **do**

      $\boldsymbol{z}'_{t-1} \leftarrow \texttt{MCLC}(\boldsymbol{z}'_{t-1}, \mathbf{s}_\theta, t-1, \boldsymbol{g}_t)$        ▷ Corrector step size hyperparameter $\lambda^{\text{DPS}}$

    **if** $t \in C_{\text{pixel}}$ **then**

      **// Pixel Optimization Step**

    **else if** $t \in C_{\text{latent}}$ **then**

      **// Latent Optimization Step**

      **for** $o = 1, \ldots, N_{\text{latent}}$ **do**

        $\boldsymbol{g} \leftarrow \zeta_t \nabla_{\boldsymbol{z}_t}\|\boldsymbol{y} - \mathcal{A}(\mathcal{D}(\hat{\boldsymbol{z}}_0))\|_2^2$

        Update $\boldsymbol{z}_{0|t}(\boldsymbol{y})$ using gradient $\boldsymbol{g}$

        **// MCLC Correction Step**

        **for** $c = 1, \ldots, N_c$ **do**

          $\boldsymbol{z}_{0|t}(\boldsymbol{y}) \leftarrow \texttt{MCLC}(\boldsymbol{z}_{0|t}(\boldsymbol{y}), \mathbf{s}_\theta, 0, \boldsymbol{g})$     ▷ Corrector step size hyperparameter $\lambda$

      **end for**

      $\boldsymbol{z}_{t-1} = \text{StochasticResample}(\hat{z}_0(y), \boldsymbol{z}'_t, \gamma)$

    **else**

      $\boldsymbol{z}_{t-1} = \boldsymbol{z}'_{t-1}$

    **end if**

  **end for**

  $\boldsymbol{x}_0 = \mathcal{D}(\boldsymbol{z}_0)$                     ▷ Output reconstructed image

  **return** $\boldsymbol{x}_0$

---

| | Inpainting (random) | Super Resolution | Gaussian Deblur | Motion Deblur | HDR | Nonlinear Deblur |
|---|---|---|---|---|---|---|
| $k$ | 5 | 5 | 10 | 10 | 5 | 5 |
| $N_c$ | 3 | 3 | 5 | 5 | 3 | 3 |
| $\lambda$ | 0.15 | 0.15 | 0.15 | 0.15 | 0.15 | 0.07 |
| $N_c^{\text{DPS}}$ | 1 | 1 | 1 | 1 | 1 | 1 |
| $\lambda^{\text{DPS}}$ | 0.05 | 0.15 | 0.15 | 0.15 | 0.10 | 0.07 |

*Table 16.* Experiment configurations for ReSample.

### D.7. Algorithmic Details: Latent DAPS

We plug MCLC into LatentDAPS (Zhang et al., 2025) and present the resulting algorithm in Algorithm 7. LatentDAPS, decoupling consecutive steps of diffusion sampling trajectory, does not conduct the reverse sampling process. Following DiffStateGrad (Zirvi et al., 2025), we apply corrections to the log-posterior gradient $\nabla_{\boldsymbol{z}_t} \log p(\boldsymbol{z}_{0|t}|\boldsymbol{y})$. For LatentDAPS, we used the original implementation without any modifications, and followed the configuration described in the paper. For our experiments, we use 50 annealing steps and integrate the ODE with two solver steps. The MCLC step is applied every $k$ iterations during each annealing-based posterior sampling stage. At each corrector step, we perform $N_c$ updates with step size hyperparameter $\lambda$, as defined in Eq. (56). Additionally, we correct the annealed noisy latent state at every annealing step, using $N_c^{\text{int}}$ corrector updates with step size hyperparameter $\lambda^{\text{int}}$. The detailed experimental settings, corresponding to those reported in Table 1, are summarized in Table 17.

---

**Algorithm 7** MCLC-LatentDAPS

---

**Require:** annealing noise schedule $\sigma_t, \{t_i\}_{i\in\{0,\dots,N_A\}}, \mathcal{E}, \mathcal{D}, \mathcal{A}, \boldsymbol{s}_\theta, \boldsymbol{y}, \gamma_t, N_c, \lambda, N_c^{\text{int}}, \lambda^{\text{int}}$

  Sample $\boldsymbol{z}_T \sim \mathcal{N}(\mathbf{0}, \sigma_T^2 \boldsymbol{I})$

  **for** $i = N_A, \dots, 1$ **do**

    Compute $\hat{\boldsymbol{z}}_0^{(0)} = \hat{\boldsymbol{z}}_0(\boldsymbol{z}_{t_i})$ by solving the probability flow ODE in Eq. (39) with $\boldsymbol{s}_\theta$

    **for** $j = 0, \dots, N-1$ **do**

      $\boldsymbol{g} \leftarrow \nabla_{\hat{\boldsymbol{z}}_0} \log p(\hat{\boldsymbol{z}}_0^{(j)}|\boldsymbol{z}_{t_i}) + \nabla_{\hat{\boldsymbol{z}}_0} \log p(\boldsymbol{z}_{t_i}|\boldsymbol{y})$

      $\hat{\boldsymbol{z}}_0^{(j+1)} \leftarrow \hat{\boldsymbol{z}}_0^{(j)} + \gamma_t \boldsymbol{g}' + \sqrt{2\gamma_t}\,\boldsymbol{\epsilon}_j, \quad \boldsymbol{\epsilon}_j \sim \mathcal{N}(\mathbf{0}, \boldsymbol{I})$

      **// MCLC Correction Step**

      **for** $c = 1, \dots, N_c$ **do**

        $\hat{\boldsymbol{z}}_0^{(j+1)} \leftarrow \texttt{MCLC}(\hat{\boldsymbol{z}}_0^{(j+1)}, \mathbf{s}_\theta, 0, \boldsymbol{g})$                   $\triangleright$ Corrector step size hyperparameter $\lambda$

    **end for**

    Sample $\boldsymbol{z}_{t_{i-1}} \sim \mathcal{N}(\hat{\boldsymbol{z}}_0^{(N)}, \sigma_{t_{i-1}}^2 \boldsymbol{I})$

    **// MCLC Correction Step**

    **for** $c = 1, \dots, N_c^{\text{int}}$ **do**

      $\boldsymbol{z}_{t_{i-1}} \leftarrow \texttt{MCLC}(\boldsymbol{z}_{t_{i-1}}, \mathbf{s}_\theta, t_{i-1}, \boldsymbol{g})$                  $\triangleright$ Corrector step size hyperparameter $\lambda^{\text{int}}$

  **end for**

  **return** $\boldsymbol{z}_0$

---

| | Inpainting (random) | Super Resolution | Gaussian Deblur | Motion Deblur | HDR | Nonlinear Deblur |
|---|---|---|---|---|---|---|
| $k$ | 5 | 5 | 5 | 5 | 5 | 5 |
| $N_c$ | 3 | 3 | 3 | 3 | 3 | 1 |
| $\lambda$ | 0.10 | 0.15 | 0.10 | 0.15 | 0.10 | 0.10 |
| $N_c^{\text{int}}$ | 1 | 0 | 1 | 3 | 1 | 1 |
| $\lambda^{\text{int}}$ | 0.15 | 0 | 0.15 | 0.15 | 0.15 | 0.15 |

*Table 17.* Experiment configurations for LatentDAPS.

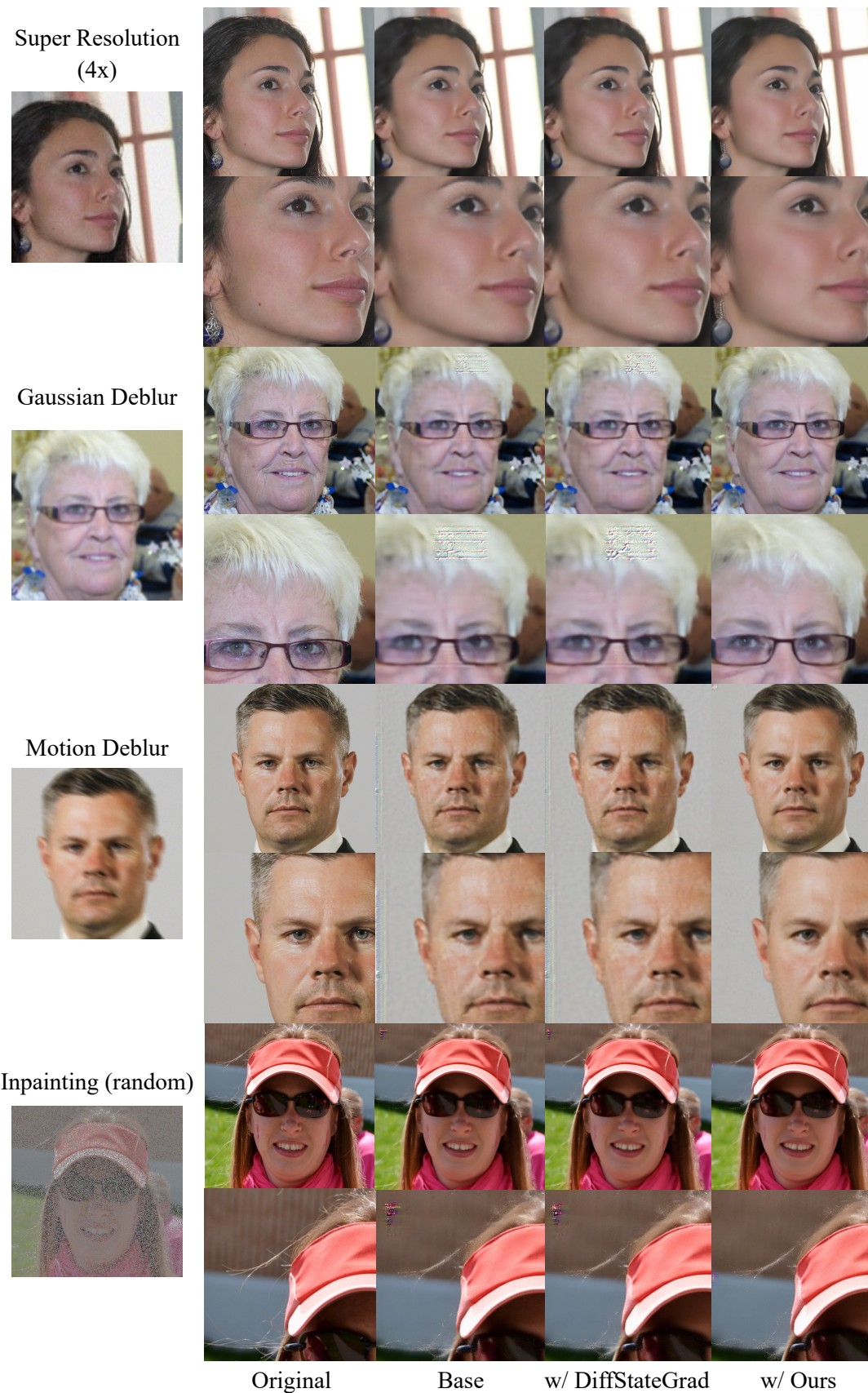

Super Resolution (4x)

Gaussian Deblur

Motion Deblur

Inpainting (random)

Original     Base     w/ DiffStateGrad     w/ Ours

*Figure 19.* Qualitative comparison of LDPS, LDPS-DiffStateGrad, LDPS-MCLC on FFHQ.

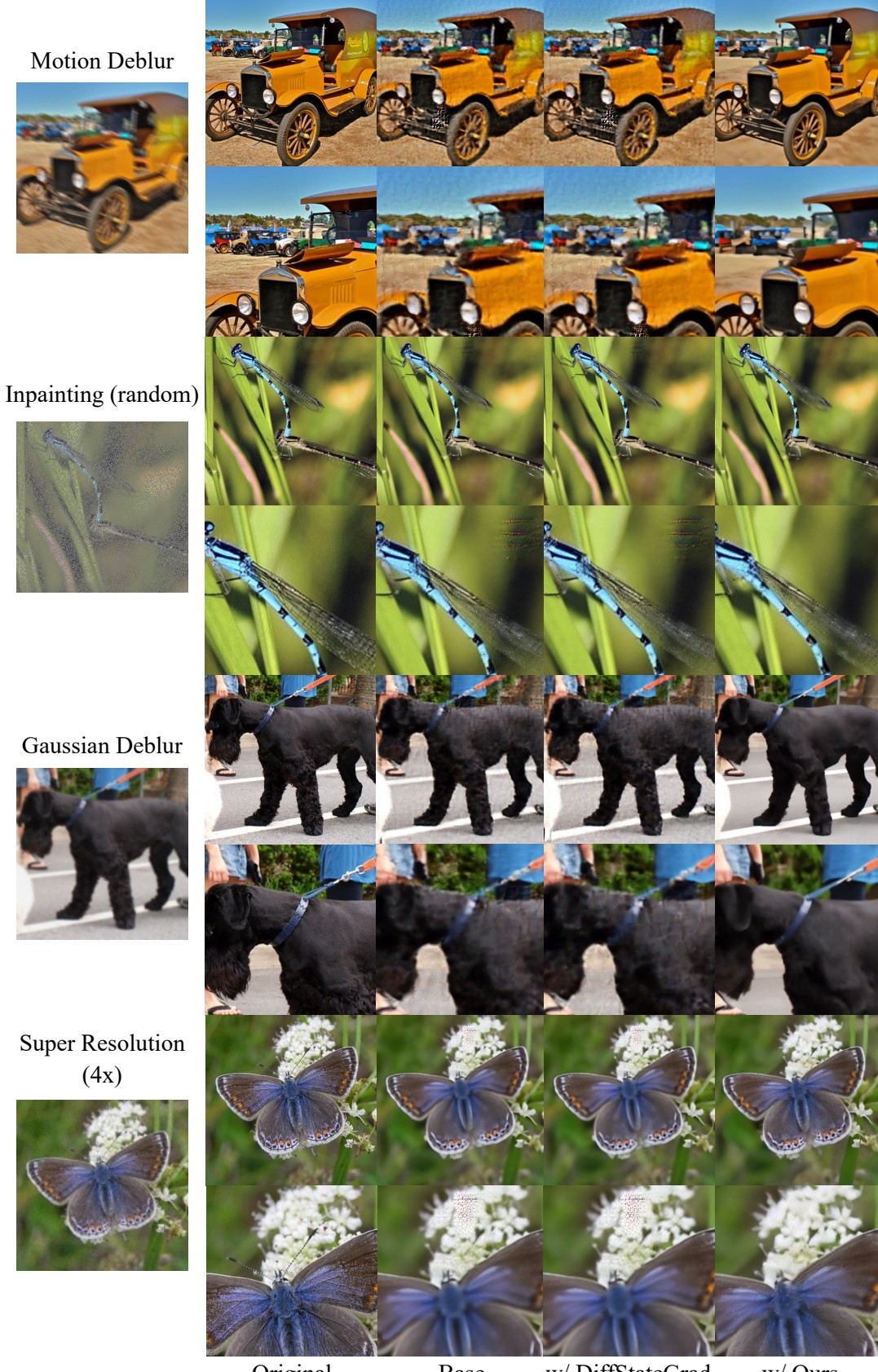

Motion Deblur

Inpainting (random)

Gaussian Deblur

Super Resolution (4x)

Original          Base          w/ DiffStateGrad          w/ Ours

*Figure 20.* Qualitative comparison of LDPS, LDPS-DiffStateGrad, LDPS-MCLC on ImageNet.

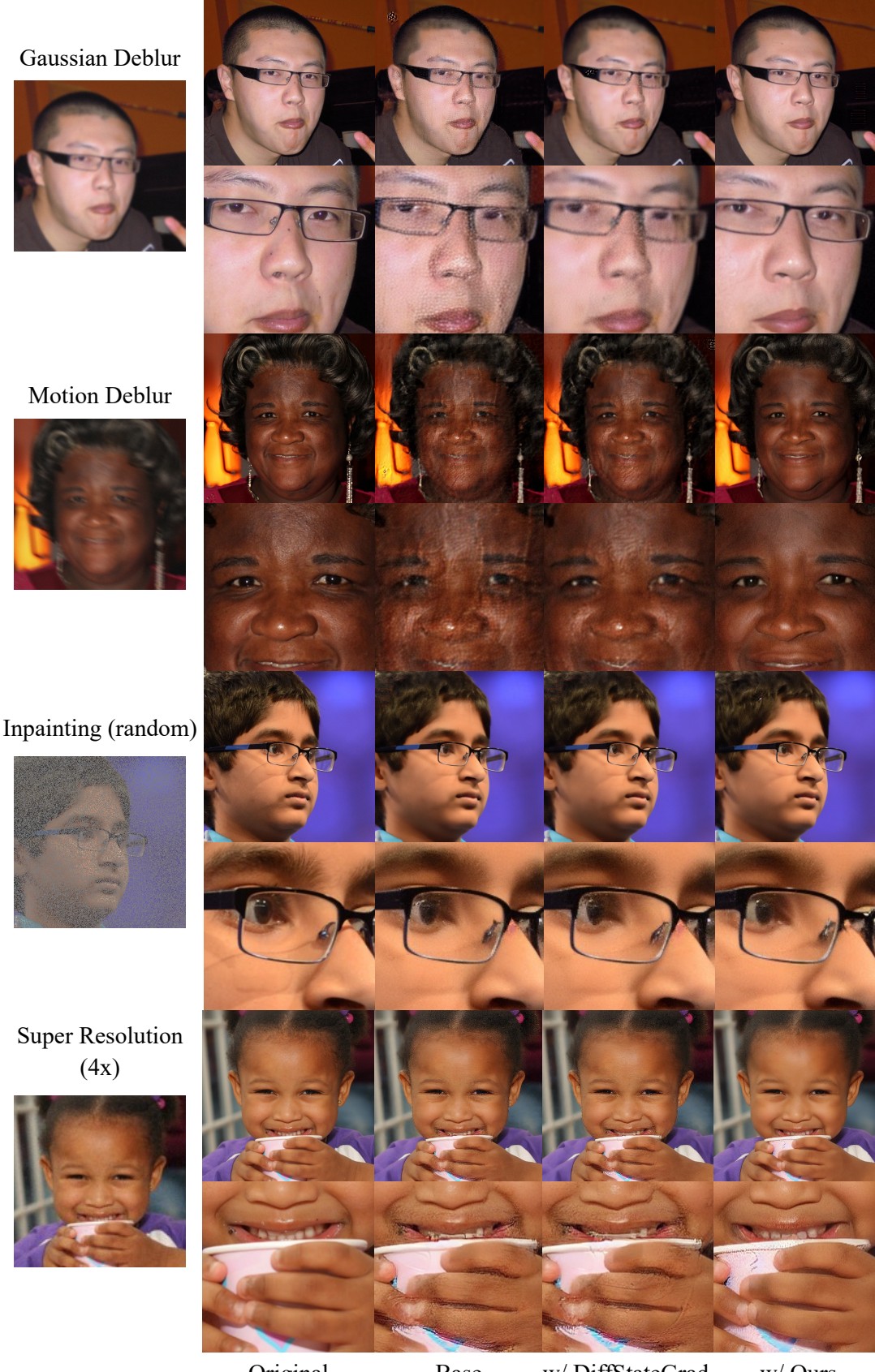

Gaussian Deblur

Motion Deblur

Inpainting (random)

Super Resolution
(4x)

Original          Base          w/ DiffStateGrad          w/ Ours

*Figure 21.* Qualitative comparison of PSLD, PSLD-DiffStateGrad, PSLD-MCLC on FFHQ.

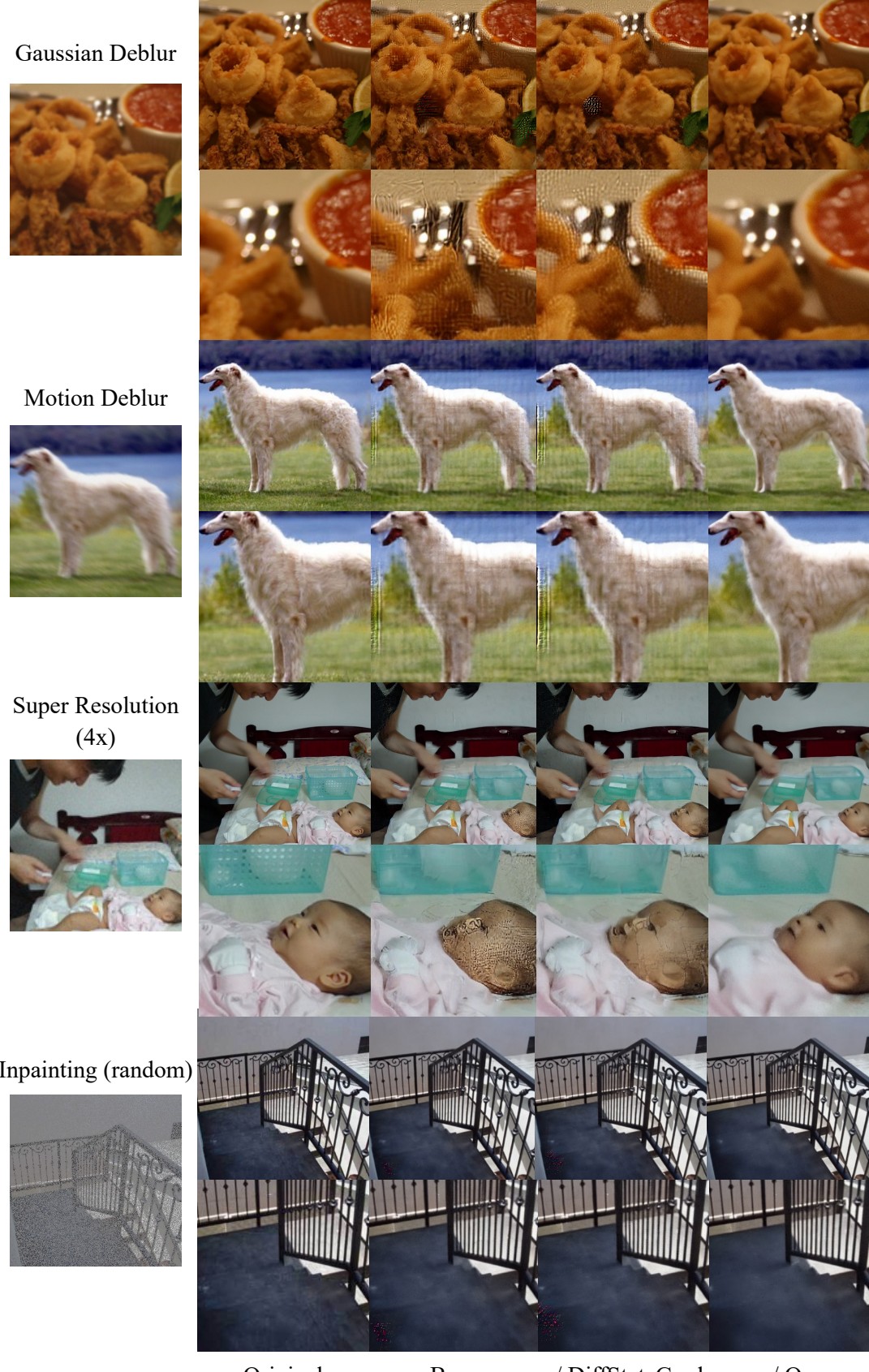

Gaussian Deblur

Motion Deblur

Super Resolution
(4x)

Inpainting (random)

Original          Base          w/ DiffStateGrad          w/ Ours

*Figure 22.* Qualitative comparison of PSLD, PSLD-DiffStateGrad, PSLD-MCLC on ImageNet.

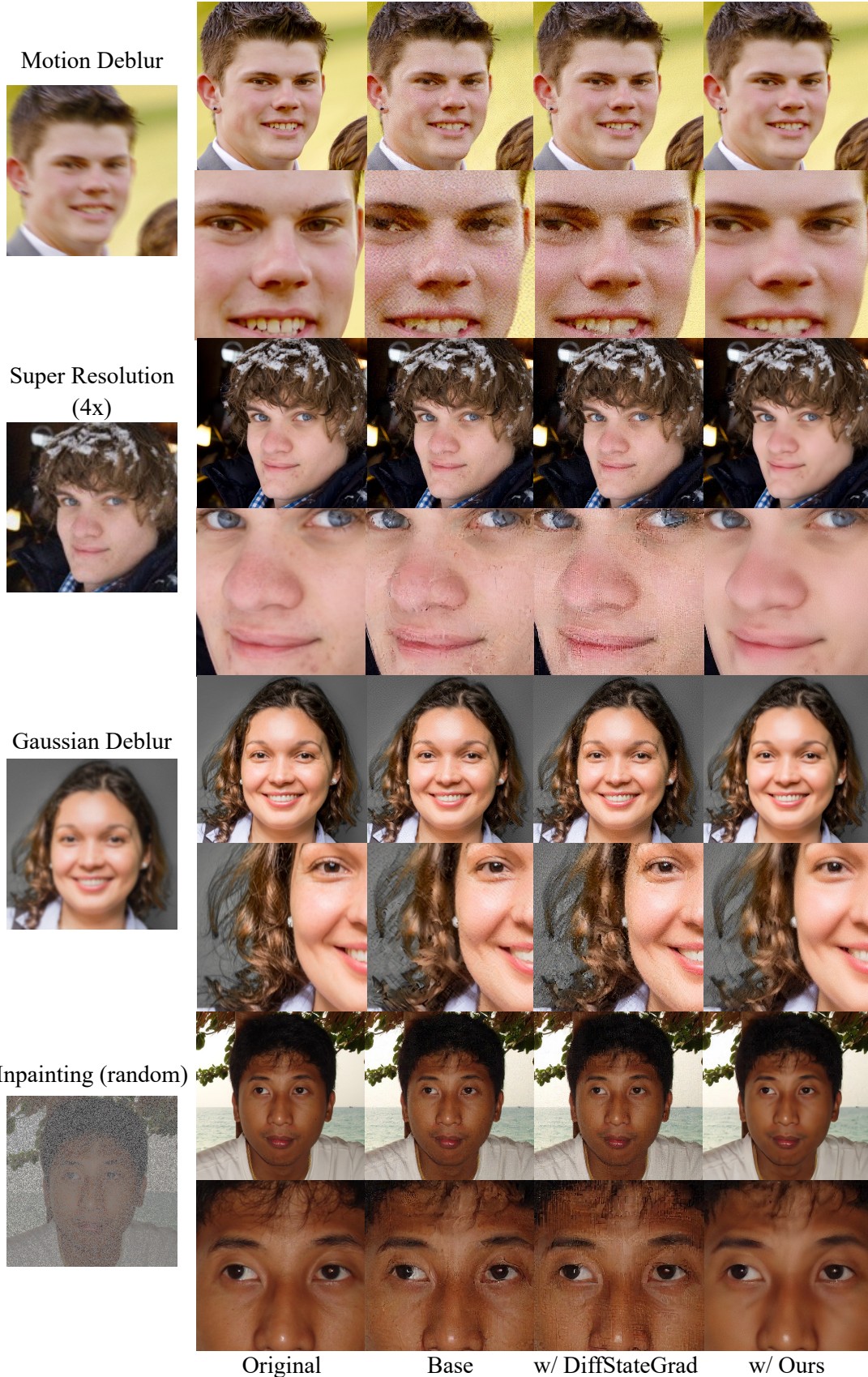

Motion Deblur

Super Resolution (4x)

Gaussian Deblur

Inpainting (random)

Original          Base          w/ DiffStateGrad          w/ Ours

*Figure 23.* Qualitative comparison of Resample, Resample-DiffStateGrad, Resample-MCLC on FFHQ.

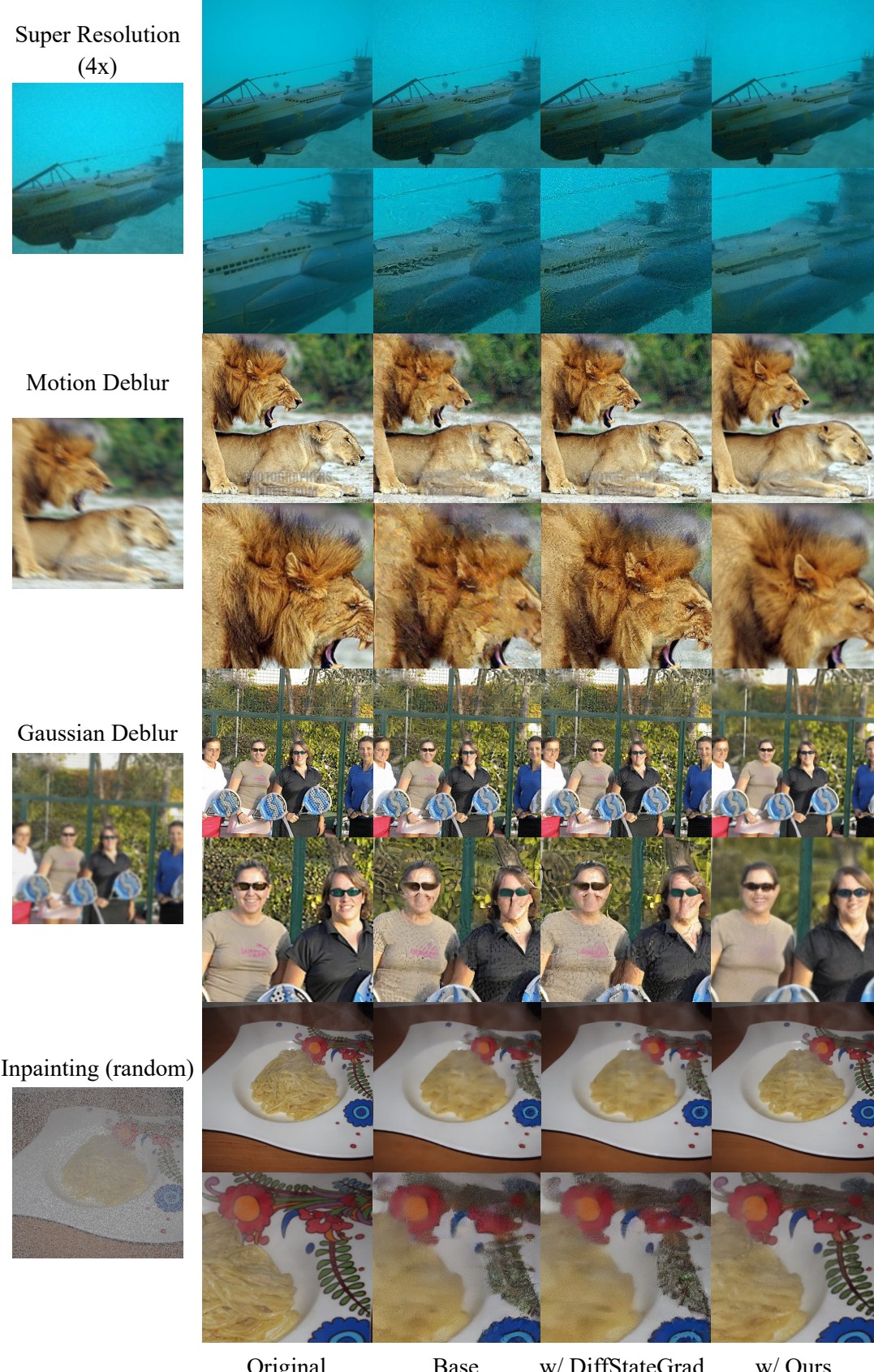

*Figure 24.* Qualitative comparison of Resample, Resample-DiffStateGrad, Resample-MCLC on ImageNet.

