# OpenReview forum: "Measurement-Consistent Langevin Corrector for Stabilizing Latent Diffusion Inverse Problem Solvers"
_ICML.cc/2026/Conference — ICML 2026 regular_

### Official Review · Reviewer_Krky · 2026-03-11

**Soundness:** 4
**Presentation:** 3
**Significance:** 3
**Originality:** 3
**Overall Recommendation:** 5
**Confidence:** 3

**Summary:**

This paper studies instability in LDM-based inverse problem solvers. The authors argue that instability arises from a discrepancy between the solver dynamics and stable reverse diffusion dynamics learned by the diffusion mode. To address this, the paper proposes Measurement-Consistent Langevin Corrector (MCLC), a plug-and-play module that applies a Langevin correction step projected onto the orthogonal complement of the measurement gradient. This reduces the divergence from the diffusion model's time-marginal distribution while preserving measurement consistency. Experiments across different inverse tasks show that integrating MCLC into existing diffusion inverse solvers improves stability and reconstruction results.

**Compliance With Llm Reviewing Policy:**

Affirmed.

**Key Questions For Authors:**

The corrector uses the orthogonal complement of the measurement gradient. What happens if the measurement gradient is noisy or poorly conditioned?

**Limitations:**

Please refer to weakness

**Strengths And Weaknesses:**

### Strengths

1. The paper provides a new interpretation of instability in diffusion inverse solvers: instability arises from the deviation between solver dynamics and the diffusion model's reverse-time dynamics. And provide a preliminary exploration to support the assumption.

2. The proposed method is theoretically grounded: the Langevin corrector monotonically decreases the KL divergence toward the diffusion marginal distribution, and projection onto the orthogonal complement of the measurement gradient preserves measurement consistency up to first order.

3. The proposed MCLC is a modular component that can be inserted into many existing diffusion inverse solvers.

4. Extensive experimental evaluation on multiple inverse tasks and multiple solvers. Results show improvement across many cases.


### Weaknesses

1. Improvements are sometimes modest. In some cases, especially in nonlinear deblurring, improvements are small or inconsistent.

2. Evaluation focuses on image inverse problems, and mostly on linear problems. The applicability to general inverse problems (e.g., physics or medical imaging) is not empirically validated.

---

> ### Author Rebuttal · Authors · 2026-03-31
>
> We sincerely thank `Reviewer Krky` for their time and effort in evaluating our work and for the positive and encouraging feedback. We appreciate the recognition of our new perspective, the theoretical grounding, and the broad applicability of our method. We provide our responses below.
>
> > **Q1. What happens if the measurement gradient is noisy or poorly conditioned?**
>
> We thank the reviewer for this insightful question. We would like to highlight that MCLC remains effective even when the measurement gradient is noisy or poorly conditioned.
>
> **Theoretically (measurement consistency).** Since MCLC projects the update onto the orthogonal complement of the measurement gradient, it ensures first-order measurement consistency at the current state, which holds regardless of gradient accuracy under the assumptions stated in the paper. The residual error in measurement consistency from higher-order terms is bounded by $E[\Delta r] \leq Ck + O(k)$ (`Theorem 3.4`). When the gradient is poorly conditioned, the constant $C$, which depends on the local smoothness of $r$, can increase; thus, the error bound is enlarged. Nevertheless, the error remains controlled by the step size. Thus, measurement consistency is still guaranteed within a bounded range.
>
> **Theoretically (KL reduction).** The KL divergence reduction (`Eq. (39)`) holds regardless of gradient quality, so MCLC continues to guide the solver toward the learned time-marginal distribution.
>
> In extremely poor gradient cases, the base solver itself becomes severely unreliable. In this case, performance may deteriorate, but this limitation is not specific to MCLC.
>
> **Empirically**, we validate this by simulating noisy and poorly conditioned gradients through varying measurement noise ($\sigma \in \{0.01, 0.03, 0.05\}$) and more severely ill-posed degradation settings (`92% random inpainting`). We use Resample as the base solver, and the results are reported below, showing that MCLC remains effective even under such conditions.
>
>
> |  |  | PSNR (↑) | LPIPS (↓) | FID (↓) | P-FID (↓) |
> | --- | --- | --- | --- | --- | --- |
> | $\sigma$=0.01 | base | 21.69 | 0.741 | 173.31 | 219.09 |
> |  | ours | **25.15** | **0.509** | **124.52** | **150.35** |
> | $\sigma$=0.03 | base | 21.51 | 0.758 | 177.81 | 225.62 |
> |  | ours | **25.01** | **0.524** | **128.18** | **152.86** |
> | $\sigma$=0.05 | base | 21.26 | 0.783 | 181.01 | 227.52 |
> |  | ours | **24.80** | **0.547** | **128.79** | **154.53** |
>
> > **W1. Improvements are sometimes modest**
>
> We agree that improvements can appear modest in some cases, particularly under relatively mild degradations where the base solver already performs well. However, under more severe settings (e.g., ×8 super-resolution, 92% random inpainting), MCLC yields clearly more noticeable gains.
>
> |  | Inpainting (92%) |  |  | Super-Resolution (x8) |  |  |
> | --- | --- | --- | --- | --- | --- | --- |
> |  | PSNR (↑) | LPIPS (↓) | FID (↓) | PSNR (↑) | LPIPS (↓) | FID (↓) |
> | Base (Resample) | 21.51 | 0.758 | 177.81 | 23.13 | 0.563 | 153.71 |
> | Ours | **25.01** | **0.524** | **128.18** | **25.19** | **0.443** | **108.90** |
>
> In particular, for nonlinear deblurring, the smaller or inconsistent gains can be attributed to a mismatch between the measurement operator and the prior model. The measurement operator is learned blur kernel that operates on 256x256 resolution inputs, so we perform experiments at 256x256 resolution, while the diffusion prior is trained at 512×512. This resolution mismatch can potentially degrade performance.  We will include a discussion on this in the final version.
>
> > **W2. Applicability to physics or medical imaging**
>
> We thank the reviewer for this insightful and valuable comment, which we believe would further strengthen the paper.
>
> Since MCLC requires only the score function of the LDM prior and the measurement gradient, it is applicable to any LDM-based inverse solver, including those in physics or medical imaging, regardless of domains.
>
> We acknowledge that empirical validation in these domains is currently limited, primarily due to the lack of publicly available LDMs trained on such data. We will clarify this limitation and highlight it as an important direction for future work.

---

> > ### Author Rebuttal · Reviewer_Krky · 2026-04-04
> >
> > The rebuttal addresses most of my concerns. I will keep my current score.

---

> > > ### Author Response · Authors · 2026-04-08
> > >
> > > We sincerely thank Reviewer Krky for taking the time to read our responses to their comments. We appreciate the positive assessment.

---

### Official Review · Reviewer_aWpo · 2026-03-12

**Soundness:** 2
**Presentation:** 3
**Significance:** 3
**Originality:** 3
**Overall Recommendation:** 4
**Confidence:** 3

**Summary:**

To resolve instability in LDM-based inverse problem solvers, this work proposes the Measurement-Consistent Langevin Corrector (MCLC). MCLC functions as a plug-and-play stabilizer that aligns solver dynamics with the diffusion prior via a projected Langevin step. By operating within the orthogonal complement of the measurement gradient, the corrector improves sampling robustness without sacrificing data consistency. The authors validate MCLC with theoretical KL-decrease guarantees and extensive experiments covering linear/nonlinear tasks and latent flow-based methods.

**Compliance With Llm Reviewing Policy:**

Affirmed.

**Final Justification:**

The authors have addressed my primary concerns through a robust rebuttal. They successfully clarified that the theoretical step-size conditions are sufficient guarantees rather than restrictive bounds, and provided empirical evidence showing that measurement consistency remains controlled in practice. Furthermore, the authors clarified that the GMM-based KL estimation is an auxiliary visualization tool and does not affect the algorithm's theoretical soundness. Lastly, they justified the use of fixed projections by highlighting that the method already succeeds in highly non-linear latent diffusion settings. The rebuttal effectively mitigates the my main technical reservations. I thus increase my score to 4.

**Key Questions For Authors:**

above

**Limitations:**

Yes

**Strengths And Weaknesses:**

**Strengths**

1. Introduces a clear, alternative perspective on instability: discrepancy from the prior’s time-marginals, rather than off-manifold geometry under linearization.

2. A Langevin corrector step projected onto the orthogonal complement of the likelihood gradient, which is intuitively well-aligned with preserving data-consistency while re-aligning with the prior.

3. Provides theoretical support that (i) the corrector monotonically reduces KL to the time-marginal and (ii) the projected version preserves measurement consistency up to a controllable bound.

4. Broad evaluation across multiple LDM-based solvers (LDPS, PSLD, ReSample, LatentDAPS), datasets (FFHQ, ImageNet), and tasks (SR, Gaussian and motion blur, inpainting, HDR, nonlinear deblurring).

**Weaknesses**

1. The theoretical results rely on assumptions that are reasonable in a stylized setting but may be fragile in practice (e.g., local properties of the residual Hessian, approximations about score/noise statistics, etc.). The paper could do more to validate how often the step-size conditions are satisfied in real runs, what happens when they are violated, and how sensitive stability is to those assumptions—especially under strongly nonlinear measurements, different decoder geometries, or more realistic degradation/noise models. Additionally, the practical estimation/measurement of the KL “gap” in high-dimensional latent space may be noisy or biased; the paper would benefit from deeper discussion on how this affects the central causal narrative. Besides, the KL-decrease analysis for the projected corrector relies on freezing the projection direction; when the projection depends on state (through ∇r(z)), conditions ensuring the same KL monotonicity are not fully discussed.

2. Guarantees assume access to the true score ∇ log p_t; the impact of score approximation error on the KL-decrease and measurement-consistency bounds is not analyzed.

3. Handling degenerate diffusion (projection to a subspace) can be subtle; the paper’s PDE-based derivation assumes a fixed, x-independent projection in the corrector sub-iteration, which should be made explicit.

4. The method’s core claim is stabilization by reducing discrepancy to time-marginals; the procedure for estimating KL(q_t||p_t) in high dimensions is relegated to the appendix and may be sensitive—more detail and validation of this estimator are needed.

5. Limited ablation on the necessity of the projection: results separating “vanilla corrector” vs “projected corrector” with quantitative fidelity/stability metrics would strengthen the case.

6. Comparisons omit some closely related, contemporary stabilizers and consistency-based samplers (e.g., SITCOM/Step-wise triple-consistency, CDDB, recent flow-based variational solvers like FLAIR); inclusion or discussion would situate MCLC more fully.


7. Constrained/projected Langevin literature and predictor–corrector samplers in diffusion (and works on degenerate diffusion operators) are under-cited; connecting to those threads would clarify where the theory aligns or differs.

8. Additional discussion contrasting MCLC with manifold-preserving methods in latent space (beyond the linear manifold critique) would be helpful to articulate when each approach is preferable.

---

> ### Author Rebuttal · Authors · 2026-03-31
>
> We sincerely thank `Reviewer aWpo` for the time and effort devoted to evaluating our work. We believe the insightful comments will strengthen our paper. Due to space limits, we provide summarized responses below and will further elaborate in the final version.
>
> > **Q1. Theoretical results depend on assumptions but may be fragile in practice**
>
> To validate this, we empirically measure under a severe setting (92% inpainting, measurement noise $\sigma=0.03$). For dimension $d=$ 64x64x4, our theory implies that step size condition (`Theorem 3.4`, `Eq. (7)`) is satisfied when $\lambda \lesssim 0.005$. At this $\lambda$, we verify the condition is consistently satisfied (i.e., $E[||\Delta z_t||]  < 1$) in real runs.
>
> As this is a sufficient condition, mild violations don’t necessarily lead to failure. In practice, larger step sizes (0.05-0.15) rather improve stability and reconstruction performance while largely preserving measurement consistency, whereas overly large step sizes (e.g., $\lambda = 0.5$) may lead to failure.
>
> These results demonstrate that the theoretical conditions are well satisfied in real runs, and that MCLC is not overly sensitive to mild violations, instead yielding improved stability and performance rather than abrupt failure. Similar trends hold across decoders and solvers. We will include these results in the final version.
>
> | $\lambda$ | $E[\lVert\Delta z_t\rVert]$ | y-PSNR (↑) | PSNR (↑) | LPIPS (↓) |
> |-|-|-|-|-|
> |0 (base)|0|35.34|25.04|0.421|
> |0.005 |0.89|35.28|25.14|0.405|
> |0.05|8.93|35.18|27.00|0.304|
> |0.15|26.17|34.73|26.27|0.389|
> |0.5|89.97|27.48|16.57|0.742|
>
> > **Q2. Impact of score approximation error**
>
> We analyze the impact of score approximation error by letting $s_\theta(x)=\nabla \log p_t(x)+e(x)$. For the **measurement consistency**, the satisfaction condition (`Eq. (59)`) is modified under the assumption that error is independent of true score:
>
> $$
> E\left[||\Delta z_t||^2\right] \approx
> (\lambda^2 + 2\lambda)
> \frac{d(d+2)}{d + e}
> \sigma_t^2.
> $$
>
> Then, the measurement consistency perturbation bound remains controlled when the score approximation error is small.
>
> For the **KL decrease**, following [`R1`], the KL evolution of MCLC is derived as:
>
> $$
> \frac{d}{dc}\mathrm{KL}\left(q_t^c \| p_t\right)
> \le
> -\frac{3}{4} \int q_t^c
> \Big\lVert P_{\perp g} \nabla \log \frac{q_t^c}{p_t} \Big\rVert^2 dx
> +
> E_{q_t^c}\left[\left\lVert P_{\perp g} e(x) \right\rVert^2\right].
> $$
>
> Since $||P_{\perp g}e||^2 \leq ||e||^2$, KL decrease remains hold when $E[||e||^2]$ is small, which is standard assumption under score matching objective and supported by our experiments, confirming that the error is well controlled in practice.
>
> - [R1] Yang and Wibisono, Convergence of the Inexact Langevin …, arXiv 22
>
> > **Q3. Clarification on state-dependency of the projection**
>
> We clarify that in our algorithm, the projection is computed only at the initial state and remains fixed during corrector sub-iterations. This is fully aligned with our theoretical derivation, ensuring KL monotonicity. We will clarify in the paper.
>
> We further note that using a state-dependent projection introduces an additional term in the KL evolution (on the right-hand side of `Eq. (38)`), $\int (1+\log \frac{q_t^c}{p_t}) \nabla \cdot (q_t^c\nabla\cdot P_{\perp g(x)})$. Since this has an indefinite sign, KL monotonicity may no longer be guaranteed. It also requires additional backprop at each sub-iteration, leading to practical inefficiency.
>
> > **Q4. Details and validation for the KL estimation procedure.**
>
> We note that measuring divergence in high-dimensional data via Gaussian/GMM modeling has been adopted [`R2-5`]. We also conduct sensitivity analyses on the KL measuring procedure w.r.t. number of GMM modes and random seeds, and observe consistent trends. We will include these results, along with pseudocode and a detailed description of the procedure.
>
> - [R2] Hershey and Olsen, Approximating the Kullback …, ICASSP 07
> - [R3] Luzi et al., Evaluating Generative Networks …, WACV 23
> - [R4] Heusel et al., GANs Trained by…, NIPS 17
> - [R5] Chen et al., Gaussian Mixture Flow..., ICML 25
>
> > **Q5. Ablation on the projection**
>
> We already include this ablation in `Table 6`; please refer to `Sec. B.2` for details.
>
> > **Q6. Additional comparison**
>
> We have added results on SITCOM and will incorporate them in `Table 2`. Due to space limits, additional comparisons will be included in the final version.
>
> | |Super Resolution (4x)| | |Gaussian Deblur| | |
> |-|-|-|-|-|-|-|
> | |PSNR|LPIPS|FID|PSNR|LPIPS|FID|
> |SITCOM|26.77|0.397|103.95|26.45|0.426|109.06|
>
> > **Q7. Under-cited related literature.**
>
> We will include the suggested literature and clarify our positioning in the related work section.
>
> > **Q8. When is manifold-preserving vs. MCLC preferable?**
>
> MCLC is preferable for complex latent space, while manifold-preserving methods are more efficient for smoother structures; the two can also be combined thanks to MCLC’s plug-and-play nature.

---

> > ### Author Rebuttal · Reviewer_aWpo · 2026-04-04
> >
> > I appreciate the authors' efforts in providing additional experiments and theoretical clarifications during the rebuttal. However, several fundamental concerns remain unresolved:
> >
> > * The step-size conditions in Theorem 3.4 appear to be overly conservative and do not align tightly with practical success, suggesting the current theoretical understanding is still preliminary.
> >
> > * The reliance on GMM-based KL estimation in high-dimensional latent spaces remains a major weak point. Without rigorous validation of the estimator's bias, the claim of "empirical KL decrease" is not fully convincing.
> >
> > * The choice to use a fixed projection during corrector steps to maintain KL monotonicity (as clarified in Q3) significantly limits the method's applicability to highly non-linear inverse problems where the measurement geometry changes rapidly.

---

> > > ### Author Response · Authors · 2026-04-08
> > >
> > > We thank the reviewer for the follow-up questions. We address each concern below.
> > >
> > > > **C1. The step-size conditions appear to be overly conservative and do not align tightly with practical success**
> > >
> > > The role of our theorem is to provide rigorous guarantees under minimal assumptions, not to tightly characterize the full range of practical success or optimal hyperparameters. In this sense, our theorem provides a sufficient condition; by definition, it guarantees success (i.e., reduction of KL divergence while preserving measurement consistency) when satisfied. However, this does not imply that success is limited only to cases where the condition is satisfied. Rather, it is natural for practical success (i.e., improved stability with acceptable degradation in measurement consistency) to extend beyond the condition. Therefore, a lack of tight alignment does not indicate that the theory is preliminary, but rather reflects its intended role.
> > >
> > > Furthermore, sufficient conditions are common in theoretical analyses, as tight characterization of optimal behavior is difficult and not well-defined. For example, some applications require strict preservation of measurement consistency (e.g., medical imaging), while others may accept deviations for improved reconstruction quality (e.g., general image restoration).
> > >
> > > Moreover, the condition is not overly conservative: As shown in our response to `Q1`, the condition is satisfied at $\lambda \lesssim 0.005$ (i.e., $E[|\Delta z_t|] < 1$), where measurement consistency is almost perfectly preserved (`y-PSNR`: 35.34 → 35.28). As $\lambda$ increases beyond this range, `y-PSNR` begins to degrade (35.28 → 35.18 → 34.73 ), consistent with our theorem. The fact that measurement consistency remains stable within the bound and degrades once the bound is exceeded indicates that the condition captures a meaningful boundary, not overly conservative one.
> > >
> > > > **C2. Reliance on GMM-based KL estimation**
> > >
> > > We would like to clarify a potential source of confusion regarding the role of GMM-based KL estimation in our work.
> > >
> > > **The GMM-based KL estimator is used solely for analysis and visualization** (`Fig. 2`), and is not involved in the algorithm and theoretical derivation of MCLC.
> > >
> > > **The KL decrease property of MCLC is guaranteed entirely by theoretical derivation, independent of KL estimation procedure**, including potential bias in GMM fitting. As the reviewer noted in `Strength 3`, and as established in Theorem 3.4 and Proposition 3.2, KL decrease guarantee holds regardless of whether or how well KL is estimated empirically for analysis. In other words, our central causal narrative that MCLC stabilizes solver dynamics by reducing discrepancy to time-marginals is not affected by the GMM-based KL estimation procedure. GMM-based visualization in `Fig. 2` serves only to illustrate this theoretically guaranteed behavior in practice.
> > >
> > > That said, we agree that GMM-based KL estimation in high dimensions may introduce bias. Nevertheless, since we compare the KL gap using the same estimator before and after applying MCLC (Fig. 2), the observed relative reduction remains meaningful even in the presence of such bias. Moreover, rigorously characterizing this bias is orthogonal to our work, since our central causal narrative, algorithm, and theoretical guarantees do not rely on GMM-based KL estimation. We will revise to account for potential estimator bias and clarify the estimator’s role.
> > >
> > > > **C3. The fixed projection limits applicability to highly nonlinear inverse problems**
> > >
> > > We respectfully disagree that fixed projection significantly limits applicability, for two reasons.
> > >
> > > **The measurement consistency perturbation bound remains controlled regardless of nonlinearity.** As shown in Theorem 3.4, although the $C$, which depends on the local smoothness of $r$, may increase for highly nonlinear problem, the perturbation bound $E[\Delta r] \leq Ck + O(k)$ is ultimately governed by $k$, which is controlled by step size. Thus, the bound may widen with increasing nonlinearity, but it does not become unbounded. We can manage this via adjusting step size, and such tuning is standard in optimization-based inverse solvers.
> > >
> > > **The LDM setting in our experiments already exhibits high nonlinearity.** Even when the measurement operator $\mathcal{A}$ is linear, composite operator $\mathcal{A} \circ \mathcal{D}(\cdot)$ becomes highly nonlinear due to the decoder [A1-2]. As a result, the problem already reflects a nonlinear measurement geometry. Since we already demonstrated consistent improvements in this setting, fixed projection does not significantly limit applicability.
> > >
> > > Furthermore, the challenge in highly nonlinear inverse problems is a known limitation shared across inverse solvers [A3-4], not specific to MCLC.
> > > We will clarify these points in the paper.
> > >
> > > - [A1] Raphaeli et al., SILO…, ICCV 25
> > > - [A2] Daras et al., A Survey…, arXiv
> > > - [A3] Zhang et al., DAPS…, CVPR 25
> > > - [A4] Zheng et al., InverseBench…, ICLR 25

---

### Official Review · Reviewer_Q2Fy · 2026-03-12

**Soundness:** 3
**Presentation:** 4
**Significance:** 3
**Originality:** 3
**Overall Recommendation:** 5
**Confidence:** 3

**Summary:**

This paper mainly addresses the issue of distribution deviation from the standard diffusion reverse process caused by measurement alignment in LDM-based inverse problem solving. The paper proposes a plug-and-play solution called Measurement-Consistent Langevin Corrector (MCLC), and provides corresponding theoretical explanations, which illustrate the motivation and the design well.

**Compliance With Llm Reviewing Policy:**

Affirmed.

**Final Justification:**

The authors have addressed all my concerns. Accordingly, I have decided to raise my score to "5, accept".

**Key Questions For Authors:**

1. Although the starting point of this paper is to improve LDM-based methods, I wonder whether this method still works on pixel-space DMs. Can the authors provide a comparison of the effects with and without their method on at least one more pixel-space DM (rather than DPS), on one dataset and task?

**Limitations:**

1. The limitations of this paper are minor. Although the proposed method shows insufficient performance on specific methods, this does not affect the contribution of this paper to promoting the development of this direction.

**Strengths And Weaknesses:**

**Strengths**

1. This paper is clearly written.
2. The motivation of the proposed method is reasonable, and theoretical explanations are provided under reasonable assumptions.

**Weaknesses**

1. When applying the proposed MCLC method to LatentDAPS in Table 1, performance degradation in metrics is observed in many cases (here "many" is relative to the improvements brought by applying MCLC to other methods). Although the authors provide some explanations in the main text and Appendix B.1, this may weakens its applicability when applied to certain methods, like methods injecting random noise during the reverse process.

---

> ### Author Rebuttal · Authors · 2026-03-31
>
> We sincerely thank `Reviewer Q2Fy` for their willingness and recognizing our work. We are glad that the motivation and theoretical explanation were found reasonable and well-presented. We provide our responses below.
>
> > **Q1. Can the authors provide a comparison of the effects with and without their method on at least one more pixel-space DM (rather than DPS), on one dataset and task?**
>
> We thank the reviewer for the suggestion. As requested, we have included additional results SITCOM [`R1`], which is the recent pixel-space diffusion inverse solver that enforces step-wise triple backward consistency for stabilizing. We evaluate on the FFHQ-256 test set across motion deblurring, Gaussian deblurring, and non-linear deblurring tasks. We will incorporate these quantitative and qualitative results into the final version, and further extend to additional pixel-space solvers if possible.
>
> |  | Motion Deblur |  |  | Gaussian Deblur |  |  | Non-linear Deblur |  |  |
> | --- | --- | --- | --- | --- | --- | --- | --- | --- | --- |
> |  | PSNR (↑) | LPIPS (↓) | FID (↓) | PSNR (↑) | LPIPS (↓) | FID (↓) | PSNR (↑) | LPIPS (↓) | FID (↓) |
> | SITCOM | 27.51 | 0.178 | 91.67 | 26.79 | 0.242 | 109.13 | 26.96 | 0.107 | 56.21 |
> | SITCOM + Ours | 28.94 | 0.147 | 82.36 | 26.81 | 0.228 | 103.88 | 26.99 | 0.106 | 55.28 |
>
> We note that our method addresses instability from a distributional perspective rather than a geometric perspective (e.g., lens of the manifold hypothesis). As a result, the benefits of MCLC can become more evident in settings with complex data distributions, such as general image priors (e.g., SD1.5, 2.1), compared to domain-specific priors (e.g., FFHQ-PDM, FFHQ-LDM, ImageNet-PDM, ImageNet-LDM), making it a key step toward more robust zero-shot inverse problem solvers.
>
> We would also like to remind the reviewer that experiments of pixel-space diffusion inverse solver with DPS are already included in `Appendix C.3` (`Table 10` and `Fig. 16`).
>
> - [R1] Alkhouri et al., SITCOM: Step-wise Triple-Consistent Diffusion Sampling For Inverse Problems, ICML 25
>
> > **W1. Less compatibility with LatentDAPS and applicability of MCLC**
>
> Thanks for the comment. Although MCLC is less compatible with such a noise annealing scheme, as shown in `Tables 3` and `4`, MCLC shows consistent gains across a range of recent diffusion- and flow-based solvers.
>
> Furthermore, we highlight that the proposed perspective on instability has strong potential and a meaningful foundation for future extensions, enabling the development of stabilization schemes for a broader class of solvers.

---

> > ### Author Rebuttal · Reviewer_Q2Fy · 2026-04-02
> >
> > The reviewer thanks the authors for their response and believes that their rebuttal has fully resolved Q1. Regarding W1, although the authors provide some explanation, they do not fully address my concern as to why LatentDAPS, specifically the noise annealing scheme, fails to benefit from the proposed MCLC method on some tasks while proving useful on others. Nevertheless, I acknowledge that W1 may be a difficult question to answer, and the authors have already offered some discussion in the rebuttal. Therefore, I would like to maintain my score and recommend the paper as "4, weak accept".

---

> > > ### Author Response · Authors · 2026-04-03
> > >
> > > We sincerely thank Reviewer Q2Fy for the thoughtful acknowledgement and a positive assessment of our work. We also appreciate the reviewer's careful reading of our detailed discussion.
> > >
> > > Regarding `W1`, we would like to provide a more precise characterization of why LatentDAPS exhibits less compatibility with MCLC, complementing our prior discussion in `Appendix B.1`. The key distinction lies in whether the solver is designed to evolve along the sequence of time-marginal distributions $\{p_t\}$ that define the reverse diffusion dynamics.
> > >
> > > In standard reverse-dynamics based solvers (e.g., LDPS, PSLD, ReSample), the estimated score contributes to each transition $z_t \rightarrow \hat{z}\_{t-1}$ together with the measurement consistency step. Even though these solvers also predict $\hat{z}\_0$ and apply a measurement-consistency update, the subsequent DDPM/DDIM-based transition back to $z_{t-1}$ **maintains a dependency on** $z_t$. For example,
> > >
> > > $$
> > > z_{t-1} = \sqrt{\bar{\alpha}\_{t-1}}\hat{z}\_0(y) +\sqrt{1-\bar{\alpha}\_{t-1}}\frac{\mathbf{z}\_\mathbf{t}-\sqrt{\bar{\alpha}\_{t}}\hat{z}\_0}{\sqrt{1-\bar{\alpha}\_{t}}}.
> > > $$
> > >
> > > Thus, the trajectory remains anchored to the reverse dynamics, and consecutive states are designed to stay aligned with $\{p_t\}$.
> > >
> > > In contrast, LatentDAPS is fundamentally not designed to follow this trajectory. At each annealing step, it freshly predicts $\hat{z}\_0^{(t)}$ via an ODE solver from $z_t$, updates itself using the $\nabla_{\hat{z}\_0} \log p(\hat{z}\_0|y)$ to obtain $\hat{z}\_0^{(N)}$, and then injects annealed noise to produce the next state (fully decoupling consecutive transitions). **Importantly, in this scheme,** $z_t$ **is discarded after the ODE step, and the next state is determined entirely by** $\mathbf{\hat{z}}\_\mathbf{0}^\mathbf{N}$, **with no dependency on the previous trajectory:**
> > >
> > > $$
> > > z\_{t-1} \sim \mathcal{N}(\mathbf{\hat{z}}\_\mathbf{0}^\mathbf{N}, \sigma\_{t-1}^2 I).
> > > $$
> > >
> > > In other words, the annealing schedule (steps) is not designed to follow the reverse dynamics, but rather to explore a broader solution space outside of the reverse dynamics through iterative updates over independent $\hat{z}\_0^{(t)}$ estimates.
> > >
> > > Since MCLC is designed to reduce the KL divergence to $\{p_t\}$ along the reverse trajectory, its stabilizing effect can be less pronounced in such a decoupled annealing scheme where the underlying solver does not follow these dynamics.
> > >
> > > Nevertheless, as the reviewer also acknowledged, this does not diminish the value and contribution of our work: (1) the introduction of a novel perspective on instability, and (2) a theoretically grounded stabilization scheme that respects measurement consistency, which is the core objective of inverse problems, providing a meaningful foundation for future stabilization schemes.

---

### Official Review · Reviewer_boGA · 2026-03-13

**Soundness:** 2
**Presentation:** 2
**Significance:** 2
**Originality:** 2
**Overall Recommendation:** 4
**Confidence:** 3

**Summary:**

The paper proposes “Measurement-consistent Langevin Corrector”, a plug-and-play module for latent diffusion inverse problem solvers that mitigates the solver instability by taking corrector steps at each reverse diffusion step, restricted to the orthogonal complement of the measurement-consistency update direction, ensuring that measurement-consistency is not affected, unlike vanilla Langevin correction. This module improves the solver dynamics of existing inverse problem solvers, especially in the latent space, and various experiments across several inverse problems show improved performance over the base solvers, validating its effectiveness.

**Compliance With Llm Reviewing Policy:**

Affirmed.

**Final Justification:**

My concerns were fairly addressed in the rebuttal. I raised my score accordingly.

**Key Questions For Authors:**

>Can authors provide ablations on the number of correction steps and how it degrades/preserves measurement consistency (see points raised in the weaknesses)?

>More severe degradations, such as large-hole inpainting, where a large chunk or a blob is masked out, may be good for assessing MCLC more thoroughly, and also for ablations in Q1 above. Did the authors try this?

>I’m confused as to what exactly this “instability” issue is attributed to in the paper. Shouldn’t it be because existing inverse solvers almost use approximations for the measurement-consistency step? Even if the approximation is good enough, wouldn’t proper scaling for the measurement-fidelity and prior steps matter (since methods rely on hyperparameters such as learning rate, etc.)?

>Can the authors discuss whether MCLC would still be effective if the measurement-consistency approximation is poor, or cases where measurements are sparse, e.g., for severe degradation operators, such as in Q2?

**Limitations:**

Yes

**Strengths And Weaknesses:**

**Strengths:**

>The paper is generally well written and organized. The proposed module’s effectiveness is validated across several base inverse solvers and a variety of inverse tasks, with the results showing improved performance. The methodology seems sound, and the experiments are well designed, though not complete. The appendix provides extensive implementation details, providing a clearer picture of the proposed method in practice.

**Weaknesses:**

>Given that MCLC builds heavily on the Langevin corrector framework and restricts the usual corrector step to a subspace where measurement consistency is preserved, it seems more incremental. MCLC also appears moderately novel as its formulation can appear more straightforward from the optimization perspective of Langevin dynamics, where each update step involves a measurement-consistency and a prior-gradient step, with MCLC essentially taking sequential updates by restricting the prior gradients to the orthogonal complement of the measurement-fidelity gradient. The paper leaves the number of corrector steps as a tuning parameter. However, it may be a crucial parameter for the instability issue and warrants ablations, since MCLC could likely degrade measurement consistency if the number of correction steps is large. This also reduces the problem to finding the correct scaling of measurement-fidelity and prior-update steps: a typical issue in optimization-based inverse solvers. Furthermore, the empirical improvements appear marginal in most cases. More severe degradation operators may be required for a complete assessment of MCLC. The notations are not consistent with the standard diffusion literature, especially in Sec 2.

---

> ### Author Rebuttal · Authors · 2026-03-31
>
> We sincerely thank `Reviewer boGA` for the time and constructive comments. We address each point below. We believe our work is further improved by reflecting the suggestions. We will include the additional results and discussion in the final version.
>
> > **Q1. Ablation study on the number of corrector steps**
>
> We have included an ablation study on the number of corrector steps under a severe degradation setting (`92% random inpainting`), along with measurement consistency (y-PSNR, i.e., PSNR on measurements).
>
> As shown in Theorem 3.4, measurement consistency perturbation is bounded but may accumulate across correction steps, which may lead to a decrease in measurement consistency. In practice, however, this decrease is marginal: y-PSNR decreases only from `35.75` to `35.47` (0→7 steps).
>
> Meanwhile, more corrector steps consistently reduce the KL gap and improve performance. This suggests our key finding that MCLC maintains measurement consistency while improving stability via better alignment with diffusion time-marginals.
>
> | correction step | y-PSNR (↑) | PSNR (↑) | LPIPS (↓) |
> | --- | --- | --- | --- |
> | 0 (base) | 35.75 | 21.51 | 0.758 |
> | 1 | 35.69 | 22.18 | 0.719 |
> | 3 | 35.60 | 22.76 | 0.679 |
> | 5 | 35.51 | 23.08 | 0.655 |
> | 7 | 35.47 | 23.33 | 0.639 |
>
> > **Q2. More severe degradations for more complete assessments**
>
> We agree that evaluating under more severe degradations would further strengthen the assessment. Our submission already includes highly challenging settings, such as super-resolution (×12, ×16) in `Tables 3` and `4`, where we observe improvements.
>
> We have additionally included results under more severe degradations, including `92% random inpainting` (large-hole) and `super-resolution ×8` (sparse), and we still observe consistent improvements, demonstrating its effectiveness in highly ill-posed settings.
>
> We note that our response to Q1 was obtained under the same 92% inpainting setting.
>
> |  | Inpainting (92%) |  |  | Super-Resolution (x8) |  |  |
> | - | - | - | - | - | - | - |
> |  | PSNR (↑) | LPIPS (↓) | FID (↓) | PSNR (↑) | LPIPS (↓) | FID (↓) |
> | Base (Resample) | 21.51 | 0.758 | 177.81 | 23.13 | 0.563 | 153.71 |
> | Ours | **25.01** | **0.524** | **128.18** | **25.19** | **0.443** | **108.90** |
>
> > **Q3. Attribution of instability in inverse solvers**
>
> Instability in LDM-based inverse solvers arises from multiple factors. While inaccurate posterior approximations can contribute [`R1`], even with accurate $\nabla_{x_t} \log p(y|x_t)$, there is no guarantee that measurement-consistency updates remain on the desired diffusion manifold at $t$. Such deviations have been observed even when the measurement consistency step is performed without approximation (e.g., at $t=0$) [`R1-3`].
>
> Such deviations can accumulate over reverse sampling [`R4`] and are further exacerbated in LDMs due to the highly nonlinear and non-unique nature of the decoder [`R1, R5-6`], making the problem more complex and ill-posed.
>
> Regardless of its origin, these factors share a common manifestation: a gap between the solver dynamics and the learned diffusion dynamics. We characterize the instability through this gap, formalize it as a KL divergence, and directly reduce it.
>
> Regarding the reviewer’s point on balancing the measurement-consistency step, hyperparameter tuning can partially mitigate instability but exhibits a ceiling and a trade-off between data fidelity and stability (`Appendix B.6`). In contrast, MCLC improves both beyond this practical ceiling, achieving gains that cannot be obtained through step-size tuning alone.
>
> - [R1] Zirvi et al., Diffusion State-Guided Projected …, ICLR 25
> - [R2] He et al., Manifold Preserving Guided Diffusion, ICLR 24
> - [R3] Song et al., Solving Inverse Problems with Latent Diffusion …, ICLR 24
> - [R4] Chung et al., Improving Diffusion Models for Inverse Problems using Manifold Constraints, NIPS 22
> - [R5] Raphaeli et al., SILO: Solving Inverse Problems …, ICCV 25
> - [R6] Chung et al., Prompt-tuning Latent Diffusion …, ICML 24
>
> > **Q4. Is MCLC still effective if the measurement consistency gradient is poor?**
>
> We simulate the poor measurement consistency step by varying measurement noise levels under a severe setting (92% inpainting, highly sparse). Due to the space limit, please refer to the table in the response to `Q1` of `Reviewer Krky`. As shown there, MCLC remains effective even with poor measurement-consistency approximations and sparse measurements.
>
> > **W1. It seems incremental**
>
> We clarify that our contribution is not a mere extension of Langevin correction: we introduce a novel perspective by characterizing instability as a measurable reverse-dynamics gap, and we propose a theoretically justified, measurement-consistent correction scheme that respects the core objective of inverse problems.
>
> > **W2. Notation consistency**
>
> We will standardize the notation following common diffusion literature.

---

> > ### Author Rebuttal · Reviewer_boGA · 2026-04-04
> >
> > I appreciate the authors' comments. However, the rebuttal fails to address most of my concerns thoroughly.
> >
> > > I've noticed that random inpainting is a surprisingly easy task for diffusion models, even with 92% masked pixels. Perhaps because it preserves low-freq content, and diffusion models are good at generating fine details, i.e., high-freq content. Also, it is easier if considered on FFHQ/AFHQ datasets, which are simpler than ImageNet. That was the reason why **I suggested masking a chunk or a blob in my question**, which removes low-frequency content and is much harder for the diffusion models. So, a large-hole inpainting task on the ImageNet dataset would be a more appropriate setting than FFHQ or other easier datasets.
> >
> > > Also, why are the correction steps in ablations limited to 7? The point of my question was not to see if MCLC works. It does work, but the concern was that this is an important parameter that controls the scales of measurement consistency and prior gradients, and therefore, monotonically increasing it should not yield monotonically improved performance (again, a better choice would be to consider harder tasks, say masking half of the image or only observing a 128x128 center-crop on a 256x256 image etc on a harder dataset like ImageNet). I believe this should prove the point.
> >
> > >  Harder tasks like 12x SR, etc., when considered on easier datasets like FFHQ/AFHQ, have a mostly unimodal posterior and therefore the conclusions are not really plausible for general cases. I would recommend doing this on ImageNet to prove the point.
> >
> > > Regarding the attribution of instability in inverse problem solvers: I thank the authors for providing a clear explanation. As mentioned, there are a multitude of reasons for this instability that contribute to intermediate samples drifting away from the manifold. I'm unsure about the claim of the method since when the measurement-consistency gradient approximation is poor (which is often the case since many inverse solvers rely on approximations that are poor for multimodal posteriors e.g. harder tasks on ImageNet), it may need many correction steps, which could actually affect measurement consistency (in effect, the practical use of the bounds on this quantity, as shown in the paper are unclear)

---

> > > ### Author Response · Authors · 2026-04-08
> > >
> > > We appreciate the reviewer’s thoughtful follow-up questions. We address each point below.
> > >
> > > > **C1. Additional results on ImageNet large hole inpainting**
> > >
> > > Following the reviewer’s suggestion, we have included experiments on ImageNet (512×512) box inpainting (masking a random 256×256 box) and extreme inpainting (observing only a random 256×256 box). Even in this severe setting, MCLC still consistently improves over both base solver and comparison method, although the gains are more modest than in random inpainting.
> > >
> > > | |Box Inpainting| | |Extreme Inpainting| | |
> > > |-|-|-|-|-|-|-|
> > > ||LPIPS|FID|Patch-FID|LPIPS|FID|Patch-FID|
> > > |Base|0.316|159.08|95.28|0.601|239.30|125.75|
> > > |DiffStateGrad|0.319|161.14|95.51|0.605|238.59|125.92|
> > > |MCLC (Ours)|0.314|153.46|92.53|0.586|236.74|117.13|
> > >
> > > We agree that large hole inpainting, which removes low-frequency structure, is more challenging than random inpainting. Importantly, this difficulty is a shared challenge across diffusion-based inverse solvers, rather than being specific to MCLC. The relatively modest gains reflect the inherent difficulty of the task, where the overall performance of DM-based inverse solvers is generally lower than in other tasks. We acknowledge that recovering low-frequency structure remains a challenge for LDM-based inverse solvers, including ours. We will include this as a limitation and future direction in the paper.
> > >
> > > Despite this, MCLC continues to outperform the comparison. We believe this stems from our novel perspective on instability and stabilization scheme, which fundamentally advance prior approaches.
> > >
> > > > **C3. Additional results on ImageNet SR x12**
> > >
> > > We also report results of SR×12 on ImageNet to verify in general cases. MCLC consistently outperforms both base solver and prior stabilization method.
> > >
> > > | |PSNR|LPIPS|FID|Patch-FID|
> > > |-|-|-|-|-|
> > > |base|20.66|0.616|265.82|201.24|
> > > |DiffStateGrad|20.52|0.649|253.46|198.85|
> > > |MCLC (Ours)|22.76|0.606|222.05|161.81|
> > >
> > > We further note that all experiments are conducted using a general prior (e.g., SD v1.5) rather than domain-specific ones (e.g., FFHQ-LDM). Since such general priors induce a broader solution space, this constitutes a more challenging and general setting.
> > >
> > >
> > > > **C2. Ablation on corrector steps in harder tasks, showing non-monotonic behavior and cumulative error effect**
> > >
> > > We have conducted an ablation study on ImageNet extreme inpainting (Table R1) and extended the range of correction steps on FFHQ 92% random inpainting (Table R2; see also the Table in the response to `Q1`). FFHQ results are included to clearly show the same trend, as base solver already degrades substantially in this setting.
> > >
> > > |Table R1: |Extreme| Inpainting| | |
> > > |-|-|-|-|-|
> > > |Corrector step| y-PSNR|LPIPS|FID|Patch-FID|
> > > |0 (base)|32.89|0.601|239.30|125.75|
> > > |1|32.88|0.603|239.19|124.74|
> > > |3|32.85|0.598|238.31|120.44|
> > > |7|32.77|0.595|238.58|118.54|
> > > |15|32.61|0.588|237.73|118.17|
> > > |30|32.23|0.598|240.23|120.26|
> > >
> > > |Table R2:|Extended|ablation|results| |
> > > |-|-|-|-|-|
> > > |Corrector step|y-PSNR|PSNR|LPIPS|FID|
> > > |15|35.39|24.51|0.577|141.75|
> > > |30|35.25|25.05|0.524|129.36|
> > > |100|34.35|24.79|0.596|150.04|
> > >
> > > Across both settings, we demonstrate the non-monotonic behavior by empirically confirming cumulative error bound and show that while the number of corrector steps is a crucial parameter, MCLC provides stabilization benefits without significantly harming measurement consistency over a wide range.
> > >
> > > Specifically, as shown in both `Tables R1` and `R2`, performance continues to improve even at a large number of steps (e.g., 15 in Table R1, 30 in Table R2). Importantly, even with many correction steps, the degradation in measurement consistency remains within an acceptable range, suggesting that the cumulative error does not grow significantly.
> > >
> > > However, when the number of steps becomes excessive (e.g., 30 in Table R1, 100 in Table R2), measurement consistency degrades more (y-PSNR: 32.89 → 32.23 in Table R1, 35.75 → 34.35, in Table R2) due to the accumulation of the perturbation bound, which in turn reduces performance gains and exhibits non-monotonic behavior. We will include these results and discussions in the paper.
> > >
> > > > **C4. In harder tasks, many correction steps can be required, which may non-negligibly degrade measurement consistency, potentially limiting the practicality of the bound.**
> > >
> > > As discussed in our response to `C2`, even with many correction steps, measurement consistency remains within an acceptable range. Although cumulative error exists, measurement consistency gradually decreases. This gradual behavior is enabled by the per-step perturbation bound in Theorem 3.4. That is, each correction results in only a controlled perturbation, thereby preventing the cumulative error from increasing rapidly even over many steps.
> > >
> > > This suggests that cumulative error does not undermine practicality of our bound, which can serve as a guideline for selecting safe hyperparameters. We will include this in the paper.

---

### Decision · Program_Chairs · 2026-04-30

**Decision:**

Accept (regular)

**Comment:**

This paper presents a theoretically grounded module for stabilizing LDM-based inverse problem solvers by identifying instability as deviation from the diffusion model's learned time-marginal distribution. Reviewers initially were initially concerned about incrementalism and modest improvements (two weak accepts, score 4). However, the rebuttal comprehensively addressed core technical concerns, incluidng those related to step-size conditions providing empirical validation of robustness to noisy gradients, and demonstrating substantial gains on severe degradation tasks. Subsequent reviewers recognized the contribution's technical merit and recommended acceptance (score 5), reflecting the rebuttal's effectiveness in resolving legitimate technical questions.

The paper merits acceptance on three grounds. First, it offers a novel perspective on solver instability distinct from prior work's focus on off-manifold geometry or linearisation assumptions, reviewers acknowledged this insight even when skeptical of broader claims. Second, it is technically sound with clear theoretical motivation and KL-divergence guarantees. Third, MCLC is potentially applicable across multiple solvers and tasks. The work advances diffusion-based inverse problem solving, an area of growing interest with a principled contribution that is non-redundant and useful to the community.

The limitations of the work are reasonable directions for future research. The authors transparently acknowledged constraints and engaged thoughtfully with reviewer feedback. The combination of technical soundness, novel perspective, broad applicability, and the author serious engagement with the reviewer concerns all contributed to this decision,.